# Synchronised WindScanner Field Measurements of the Induction Zone Between Two Closely Spaced Wind Turbines

Anantha Padmanabhan Kidambi Sekar[*1], Paul Hulsman[*1], Marijn Floris van Dooren[1], and Martin Kühn[1]

[*]These authors contributed equally to this work.
[1]ForWind, Institute of Physics, University of Oldenburg, Küpkersweg 70, 26129 Oldenburg, Germany

**Correspondence:** Anantha Padmanabhan Kidambi Sekar (anantha.kidambi@uni-oldenburg.de)

**Abstract.** Field measurements of the flow interaction between the near-wake of an upstream wind turbine and the induction zone of a downstream turbine are scarce. Measuring and characterising these flow features in wind farms for various operational states can be used to evaluate flow models and design control systems at the windfarm level. In this paper, we present induction zone measurements of a utility-scale 3.5 MW turbine with a rotor diameter of 126 m in a two-turbine wind farm operating

under waked and un-waked conditions. The measurements were acquired by two synchronised continuous-wave WindScanner lidars that could resolve longitudinal and lateral velocities by dual-Doppler reconstruction. An error analysis was performed to quantify the uncertainty in measuring complex flow situations with two WindScanners by simulating the measurement setup, WindScanners sensing characteristics, and inflow conditions in a Large-Eddy Simulation. The flow evolution in the induction zone of the downstream turbine was characterised by performing horizontal planar dual-Doppler scans at the hub height for

four different inflow cases, varying from undisturbed inflow to full and partial wake scenarios. The measurements revealed the existence of a horizontal asymmetry in the induction zone possibly due to a combination of the rotor and terrain effects under undisturbed inflow. Evaluation of the engineering models of the undisturbed induction zone showed good agreement along the rotor axis. In the full wake case, the measurements indicated a deceleration of the upstream turbine wake due to the downstream turbine induction zone as a result of the very short turbine spacing. We observed the downstream turbine induction zone during

wake steering while the lateral movement of the deflected wake could be measured for the first time in the free field. The measurements further highlight the challenges in conducting field measurements and therefore the emphasis on the need for a thorough description of the measurement setup and an evaluation of the associated uncertainties to provide comprehensive traceability.

## 1   Introduction

During operation, wind turbines create a reduced velocity region upstream due to rotor thrust, i.e. the induction zone. To account for the induction zone, the IEC 61400-12-1 (International Electrotechnical Commission, 2022) standard recommends performing freestream velocity measurements more than 2 - 4 rotor diameters ($D$) upstream of the turbine. Wind turbines also create wakes, the main driver of unfavourable aerodynamic interactions in a wind farm where the downstream turbine extracts

less power and is subject to higher structural loads due to reduced wind speeds and high wake turbulence. The near wake of a turbine extends 2.0 $D$ - 4.0 $D$ downstream and is highly influenced by rotor aerodynamics (Göçmen et al., 2016). Therefore, for tightly packed wind farms, the induction zone of a downstream turbine can overlap with the near wake of an upstream turbine.

The upstream induction zone of a wind turbine has consequences for many wind power applications. The velocity deficit upstream of the turbine is responsible for the estimation bias in power curve measurements for isolated turbines (Slinger et al., 2020) and global blockage at the wind farm level (Schneemann et al., 2021). Moreover, the flow slowdown and expansion around the turbine also affect lidar-based feedforward controllers, which require precise information on the velocity magnitudes and arrival times at the rotor (Dunne et al., 2014). Several approaches have been previously followed to numerically (Medici et al., 2011; Branlard and Gaunaa, 2015; Troldborg and Meyer Forsting, 2017) and experimentally (Asimakopoulos et al., 2014; Simley et al., 2016; Mikkelsen et al., 2020) investigate the induction zone in free inflow conditions. The most detailed three-dimensional triple-synchronised lidar characterisation of the induction zone by Simley et al. (2016) was performed around a Vestas V27 turbine with a diameter of 27 m, which is not representative of modern utility-scale multi-MW turbines.

Wind turbines operating in the downstream rows of wind farms are not always exposed to undisturbed inflow. Depending on the farm layout, wind direction, and wake effects such as meandering (Trujillo et al., 2011) and wake deflection strategies (Jiménez et al., 2009), the downstream turbines operate under partial or fully waked inflow. High-resolution measurements of the induction zone in partial and fully waked inflows are still lacking.

Engineering models of the induction zone have been developed to accurately estimate the annual energy yield and implement flow control strategies. Medici et al. (2011) presented a 1-D model for the induction zone using a vortex sheet method. Branlard and Gaunaa (2015) developed a 2-D induction zone model based on a vortex cylinder implementation. Troldborg and Meyer Forsting (2017) presented a self-similar analytical 2-D induction zone model. Branlard and Meyer Forsting (2020) coupled these models with the wind farm simulation tool FLOw Redirection and Induction in Steady State (FLORIS) (NREL, 2023) to provide flow estimations for wind farm control purposes. Although the coupling was evaluated against actuator disk simulations, a comparison with full-field data has not yet been performed because of the lack of high-quality field measurements.

Field measurement campaigns using scanning lidars provide valuable data which can be used to characterise the induction zone behaviour for highly dynamic inflow conditions, atmospheric stabilities, and turbine interactions. However, conducting full-field measurements is challenging because of the complicated installation, highly dynamic inflow conditions, finite number of measurement sensors, and their associated limitations and uncertainties, which add complexity during post-processing. Multi-lidar systems such as WindScanners (Simley et al., 2016) can perform user-defined trajectories, whereby the laser beams are synchronised in space and time, enabling the resolution of two or three wind velocity components depending on the number of devices used. These devices have been used previously to map the induction zone (Simley et al., 2016), measure the flow around trees (Angelou et al., 2021), helicopter downwash (Sjöholm et al., 2014) and in the wind tunnel (van Dooren et al., 2017; Hulsman et al., 2022b; van Dooren et al., 2022). Depending on the orientation and scan pattern, detailed two- or three-dimensional flow retrievals are possible. However, owing to the lidar measurement principle and scanning limitations, such as

the volume averaging effect, assumptions of vertical velocity for dual-Doppler reconstruction, scanning speeds, beam pointing,
and intersection accuracies, a thorough error and uncertainty assessment is required before interpreting the measurements.

Several studies have been conducted to estimate the measurement accuracy of scanning lidar retrievals. van Dooren et al. (2017, 2022) presented an uncertainty analysis considering the lidar measurement uncertainty and the artificially added uncertainty of the dual-Doppler reconstruction for a two-lidar configuration. Giyanani et al. (2022) presented an uncertainty model for reconstructing a 3D wind vector considering the probe volume and the pointing accuracy for a three-lidar config-
uration. Emulating lidar measurement properties in high-fidelity CFD simulations provides a high-quality reference for error assessment and uncertainty quantification. Such approaches have been extensively utilised to understand long-range, pulsed scanning lidar measurements and their limitations (Lundquist et al., 2015; Bromm et al., 2018; Rahlves et al., 2022; Robey and Lundquist, 2022). For continuous-wave systems, Kelley et al. (2018); Debnath et al. (2019) used virtual-lidar in Large-Eddy Simulations (LES) approach to evaluate the accuracy of retrieving horizontal wind speeds for turbine-mounted wake scanning
lidars considering effects such as probe volume averaging, assumption of zero vertical velocity and atmospheric effects such as stability. Meyer Forsting et al. (2017) utilised a virtual lidar technique to understand the influence of measurement averaging on wake measurements. They reported that the differences between lidar and point measurements are greatest at wake edges where the measurement volume extends from the wake into the freestream reaching up to 30 % at 1 $D$ downstream up to 60 % at 3 $D$ downstream.

In this study, two synchronised ground-based continuous-wave WindScanner lidars were used to characterise the flow region between two 3.5 MW turbines spaced 2.7 $D$ apart. The very short spacing creates an interaction between the near wake of the upstream turbine and the induction zone of the downstream turbine. During the measurement campaign, we implemented an active wake steering control on the upstream turbine. The near wake-induction zone interaction is of interest for wake redirection control. Therefore, cases such as partial and full wake impingement with the induction zone are studied here.

Considering the measurement campaign, the main objectives of the paper include:

1. Demonstration of two-dimensional scanning of wind fields around utility-scale turbines with two synchronised Wind-Scanner lidars.

2. Identification and investigation of errors associated with performing ground-based synchronised scanning lidar measurements with two WindScanner in a controlled simulation environment.

3. Characterisation of the two-dimensional induction zone behaviour and interaction between two closely spaced turbines for unwaked, waked and partial wake scenarios and evaluation of induction zone models.

The remainder of this paper is organised as follows. Section 2 describes the measurement and LES simulation setup. We discuss results from the LES simulations and the full field measurements in Section 3. A discussion of the results and conclusions are presented in Section 4 and 5 respectively.

## 2  Methods

We describe the layout of the wind farm in Subsection 2.1 along with a description of the measurement setup. Subsection 2.2 contains information on WindScanners, programmed scan trajectories, and data processing methods. The datasets shown in this paper are collected in Subsection 2.4 while the setup and description of the Large-Eddy Simulation and lidar simulator are available in Subsection 2.5.

### 2.1  Test site description and inflow characterisation

The measurement campaign was conducted from November 2020 to June 2021 at a wind farm close to Kirch Mulsow in Northern Germany. The site has two eno126 turbines from eno energy systems GmbH with a rated wind speed of 11.4 m/s and power of 3.5 MW with a diameter of 126 m. The upstream and downstream turbines are abbreviated as WT1 and WT2 respectively.

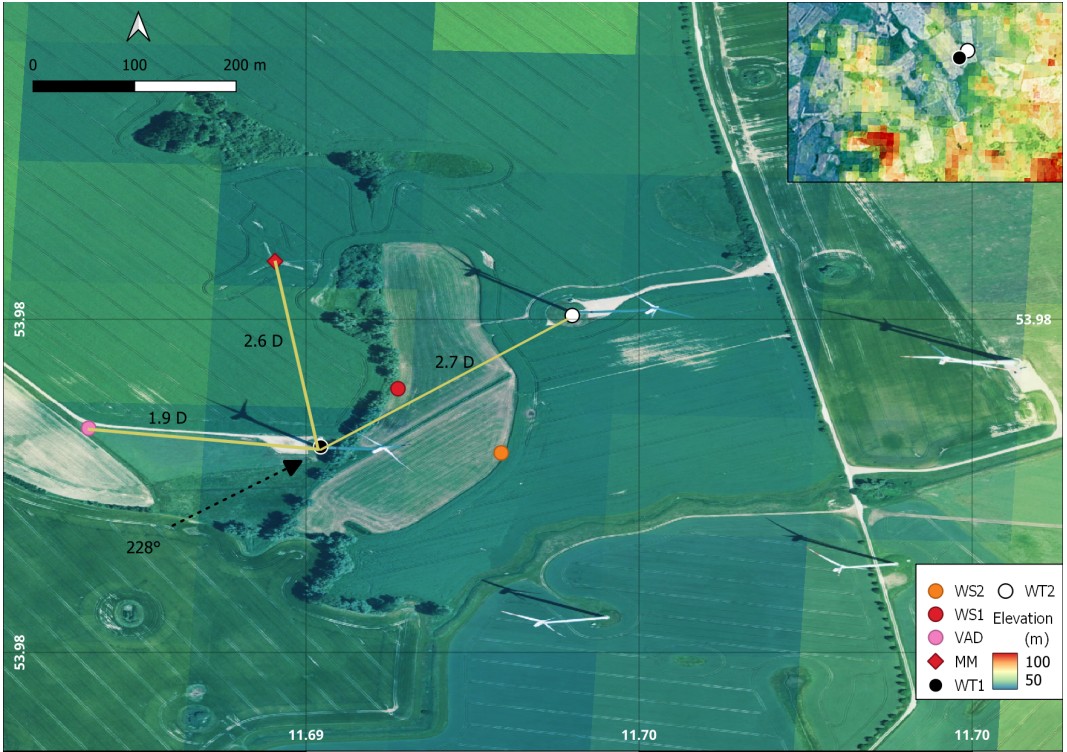

**Figure 1.** The wind park and measurement layout at Kirch Mulsow overlayed with elevation contours. A zoomed-out image of the site is shown in the top right corner illustrating the hills present upstream of the wind park. Here WT1 and WT2 refer to the upstream and downstream turbines, MM and VAD refer to the met mast and the inflow lidar while WS1 and WS2 refer to the two WindScanners. © Microsoft Bing.

The layout of the site is illustrated in Fig. 1. The hub height of the downstream turbine at 137 m is 20 m higher than that of the upstream turbine at 117 m. The site itself is characterised as farmland with moderately rolling hills. The elevation data presented in Fig. 1 was obtained with a resolution of 200 m maintained by the German Ministry of Cartography and Geodesy. While the elevations at the turbine locations are approximately 52 m, abrupt changes in elevation are seen upstream notably the presence of a small hill with an elevation of 105 m, 22 D upstream of WT1 along the predominant wind direction of 228°, creating a slope of 1.09° towards the two turbines. The village of Garvensdorf with its farmhouses was approximately 1200 m (9.5 D) upstream of WT1. Notably, a treeline exists between WT1 and WT2 extending towards the met mast with a height of approximately 15 m-20 m that can act as a windbreak (Counihan et al., 1974) while other tree lines and clumps of forested areas are present at various upstream positions along the 228° sector.

The prevailing wind direction during the measurement campaign was from the west-south-westerly sector. The two turbines are fully aligned for a wind direction of 228°. During the measurement campaign, wake steering tests were performed on the upstream turbine leading to partial wake scenarios at the downstream turbine. Additional information on the yaw controller is available in Hulsman et al. (2022a).

Inflow conditions were measured by a met mast placed 2.6 D north of WT1, equipped with two anemometers, Thies First Class Wind Transmitter anemometer of type 4.3352.00.400 at the lower tip of 54 m and close to the WT1 hub height of 116 m. A wind vane of type Thies First Class Wind Direction Transmitter of type 4.3151.00.212 is also installed at 112 m. All the instruments stored the data at a sampling rate of 50 Hz. To measure the atmospheric stability, an integrated $CO_2/H_2O$ open-path gas analyser and 3D sonic anemometer (Irgason, Campbell Scientific) were also installed on the mast at a height of 6m on a boom oriented towards 136°. More details on the derivation of the Obhukhov length from the Irgason are detailed in Bromm et al. (2018). The inflow measurements were further supported by a WindCube 200S lidar placed 1.9 D upstream of the WT1. The inflow lidar was performing VAD scans with an elevation angle of 75° and with range gates set from 50 m to 840 m with a spacing of 5 m and a pulse length of 25 m. The accumulation times and angular speeds were 0.5 s and 30° s$^{-1}$, respectively. The data from the VAD scans were binned into 10-minute averages from which the wind shear and veer profiles were estimated. The turbine heading of WT1 and WT2 during operation was precisely measured using a differential GPS System of the type 3 Trimble Zephyr™ model. All the measurement devices were synchronised to the UTC time.

## 2.2 WindScanners

The WindScanners are continuous-wave (cw) scanning lidars with a steerable scan head that users can program to perform any user-defined scan trajectory (Mikkelsen et al., 2017). The steerable scan head consists of two prisms connected to individual drives, which can be rotated independently, while a third motor is used to control the focal distance of the lidar. Each of the two prisms deflects the focused laser beam by ±30° to achieve a maximum measurement cone angle of 120°. In the present setup, the lidar can continuously sample line-of-sight speeds at a maximum sampling rate of 451.7 Hz. Two WindScanners were installed in the field in the region between the two turbines inside offshore containers for weather protection (Fig. 2 (a)).

Both WindScanners synchronously provide a Doppler velocity spectrum for every measurement sample calculated from a discrete Fourier transform of the backscattered light sampled at 120 MHz. Many individual Doppler spectra are averaged to

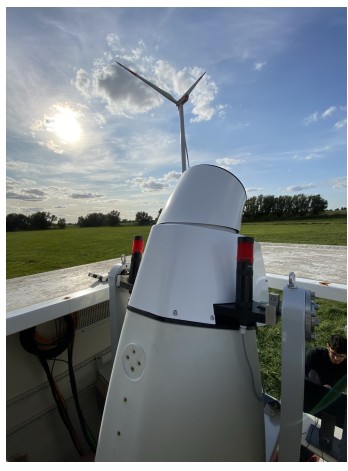 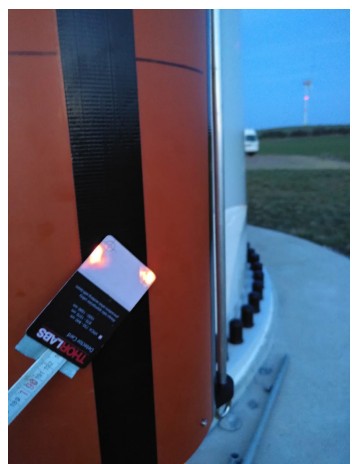

**Figure 2.** (a) The WindScanner WS2 installed in a weatherproof container while WT1 is seen in the background. The WindScanners were lifted through a hatch on the roof during operation using a hydraulic table. (b) The two laser beams from WS1 and WS2 simultaneously focused over a distance of approximately 200 m onto a 5.3 cm by 8.6 cm laser beam detector card, seen as white dots at the bottom of WT2 after the final steering calibration.

reduce noise, and the shot-noise-based mean background spectrum is removed to obtain the peak of the Doppler spectra. The
135 line-of-sight velocity is estimated by determining the spectral peak through the median peak-finding method for continuous-wave lidars, as it is less sensitive to spurious noise than the centroid and maximum methods (Angelou et al., 2012).

A single lidar can only estimate the line-of-sight speed along the laser beam direction that contains contributions from all three velocity components:

$$v_{\text{los}} = \cos(\chi)\cos(\delta)u + \sin(\chi)\cos(\delta)v + \sin(\delta)w. \tag{1}$$

where $u, v$ and $w$ are the longitudinal, lateral, and vertical wind velocity components, respectively, and $\chi$ and $\delta$ are the azimuth and elevation of the laser beam, respectively. By synchronising the two WindScanners in time and pointing at the same point, the WindScanners can estimate in the intersection point, the two dimensional wind speed component projected on the plane defined by the beams.

$$\begin{bmatrix} v_{\text{los},1} \\ v_{\text{los},2} \end{bmatrix} = \begin{bmatrix} \cos(\chi_1)\cos(\delta_1) & \sin(\chi_1)\cos(\delta_1) & \sin(\delta_1) \\ \cos(\chi_2)\cos(\delta_2) & \sin(\chi_2)\cos(\delta_2) & \sin(\delta_2) \end{bmatrix} \begin{bmatrix} u \\ v \\ w \end{bmatrix} \tag{2}$$

The longitudinal ($u$) and lateral wind components ($v$) can be resolved by an additional assumption of the vertical flow component and combining the two $v_{\text{los}}$ measurements by dual-Doppler wind field reconstruction by solving Eq. (2). Equation 2 can now be rewritten as:

$$u = \frac{\sin(\chi_2)\cos(\delta_2)(v_{\text{los},1} - \sin(\delta_1)w) - \sin(\chi_1)\cos(\delta_1)(v_{\text{los},2} - \sin(\delta_2)w)}{\cos(\delta_1)\cos(\delta_2)\sin(\chi_2 - \chi_1)} \tag{3}$$

$$v = \frac{\cos(\chi_1)\cos(\delta_1)(v_{\text{los},2} - \sin(\delta_2)w) - \cos(\chi_2)\cos(\delta_2)(v_{\text{los},1} - \sin(\delta_1)w)}{\cos(\delta_1)\cos(\delta_2)\sin(\chi_2 - \chi_1)} \tag{4}$$

In our measurements, the actual local value of the $w$ component is unknown. Without generalisation, we assume that the vertical flow component to vanish in our case. The uncertainty associated with measuring three-dimensional flow events with two synchronous lidars is discussed in Section 2.3. Another important lidar measurement property is volume averaging, that is, the $v_{\text{los}}$ measurements contain weighted contributions along a volume extending on either side of the focus point along the laser beam direction. The measured line-of-sight velocities of a cw lidar at the position $\boldsymbol{x} = (x, y, z)$, $v_{\text{los}}(\boldsymbol{x})$ can be mathematically expressed as the convolution of the wind vector $\mathbf{u}(\boldsymbol{x})$ projected along the laser beam direction and the volume averaging function:

$$v_{\text{los}}(\boldsymbol{x}) = \int\limits_{-\infty}^{\infty} \phi(s)\boldsymbol{n} \cdot \mathbf{u}(s\boldsymbol{n} + \boldsymbol{x})\,\mathrm{d}s. \tag{5}$$

Here, $\boldsymbol{n}$ is the unit vector along the line-of-sight direction and $\phi(s)$ is the spatial volume averaging function following Sonnenschein and Horrigan (1971) for cw lidars approximated as a Lorentzian function where $s$ is the distance from the focal point along the laser beam. For cw lidars, the range weighting of line-of-sight speeds that occur along the laser beam direction at a point located at a distance $f$ away from the lidar can be expressed as the Full Width at Half Maximum (FWHM) of the focused laser beam $\Gamma = 2\frac{\lambda f^2}{\pi a}$ where $\lambda = 1.56\,\mu\text{m}$ and $a = 56$ mm are the laser wavelength and effective radius of the lidar's 6" aperture telescope, respectively. As the length of the measurement volume is related to $f^2$, the measurement volume is quite large at large distances, and hence turbulent structures smaller than the measurement volume will be low-pass filtered by the lidar.

### 2.2.1 Scanning patterns

As our region of interest is the inflow of the downstream turbine, the WindScanners are programmed to perform spatially and temporally synchronised horizontal plane scans upstream of WT2. The measurement plane is at hub height and centred around the alignment of WT1 and WT2 at 228°. The WindScanners were not perfectly symmetrical to WT2 because of a tree line which prohibited symmetrical placement of WS1 with WS2 and WT2. The measurements are visualised in a global fixed reference frame centred at the bottom of WT2, where the $x$-axis is the connecting line between the two turbines and the $y-, z-$ axes are positive in the downstream and upward directions, respectively. The scan pattern was composed of a sinusoidal variation of the $x-, y-$ coordinates of the focal point:

$$x(t) = A_{\text{x}}\sin\left(\frac{2\pi t}{T}\right) + x_0 \qquad\qquad y(t) = A_{\text{y}}\sin\left(\frac{20 \cdot 2\pi t}{T}\right) + y_0 \qquad\qquad z(t) = z_0. \tag{6}$$

Here $A_{\text{x}} = 0.60\,D$, $A_{\text{y}} = 0.59\,D$ are the amplitudes while $x_0 = $ -0.20 $D$, $y_0 = 0\,D$, $z_0 = 137.0$ m are the offsets and $T$ is the time period to complete each trajectory with each scan taking 29.6 s to complete. The horizontal scan plane at the hub height of WT2 extends from $0.8\,D$ upstream of the turbine to $0.4\,D$ downstream, with a width of $1.18\,D$ as shown in Fig. 3. The offsets due to the terrain-induced height differences and the vertical offset of the WindScanners inside the container mounting are included in Eq. (6), and are tabulated in Table 1.

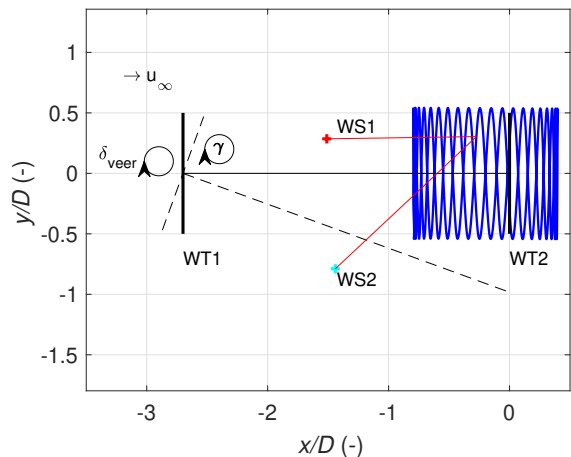

**Figure 3.** Illustration of the horizontal scanning pattern performed by the WindScanners, indicating the relative position of the two turbines and the two WindScanner with and without an intentional misalignment. The coordinate system is centred at the bottom of WT2.

**Table 1.** Relative and normalised distances from the bottom of WT2 and WS1, WS2 and WT1. The height offsets for WS1 and WS2 are calculated from the middle of the outer prism at the highest jacked-up position of the hydraulic table (Fig. 2 (a)).

|  | x (m) | y (m) | z (m) |
|---|---|---|---|
| WS1 | -157.82 (1.25 $D$) | -112.37 (0.89 $D$) | -1.25 |
| WS2 | -54.04 (0.43 $D$) | -199.76 (1.59 $D$) | -0.60 |
| WT1 | -240.98 (1.91 $D$) | -234.04 (1.86 $D$) | -2.06 |
| WT2 | 0 | 0 | 0 |

With a temporal sampling rate set at 451.7 Hz, each complete scan had approximately 13079 measurement points. In this sector, active wake steering was performed by toggling between two unique wake steering controllers and one greedy controller where no wake steering is performed, each operational for 35 minutes. The measurement campaign regarding the active wake steering is described in detail in Hulsman et al. (2022a). The WindScanner measurements are then subdivided into 35 min blocks, each representing a different operating state of the upstream turbine. All horizontal plane scans are grouped and averaged to obtain averaged profiles of the measured longitudinal and lateral velocities. For visualisation, the longitudinal and lateral velocities are interpolated using a cubic interpolation scheme onto a uniform grid with a spacing of 10 m. We rotated all measurements in the global reference frame into the main wind direction at the met mast hub height.

## 2.3 WindScanner measurement errors and uncertainties

While performing synchronised WindScanner measurements, several errors affecting the measurement accuracy can be broadly divided into single- and dual-lidar errors. For this particular site and measurement setup, the various lidar errors, their impact

and their analysis methodology are tabulated in Table 2.

**Table 2.** Summary of Dual-Doppler lidar measurement errors. Here, LES and SUP refers to Large Eddy Simulations and Standard Uncertainty Propagation methods that are described in the following sections.

| Error | Source | Impact | LES | SUP |
|---|---|---|---|---|
| Single-Lidar | | | | |
| $v_{\text{los}}$ accuracy | Inaccuracy in estimation of radial wind speeds | Low | | ✓ |
| Probe volume averaging | Measurement volume variation during scanning | Medium | ✓ | |
| Dual-Lidar | | | | |
| Pointing accuracy | Imprecise pointing angles inherent to the lidar systems | Low | | ✓ |
| Dual-Doppler reconstruction error | Amplification of single-Doppler uncertainty due to dual-Doppler reconstruction | High | ✓ | ✓ |
| Statistical uncertainty | Flow turbulence combined with slow scanning times requires multiple scans | Medium | ✓ | |
| Assumption of $w = 0$ m/s | Assumption for a dual-lidar setup | High | ✓ | ✓ |

### 2.3.1 Single-lidar errors and uncertainties

First, we discuss the sources of the errors associated with single-lidar systems. For WindScanners, the absolute measurement uncertainty of the lidar radial velocity estimation was experimentally determined by Pedersen and Courtney (2021) to be less than 0.1% under nearly ideal conditions.

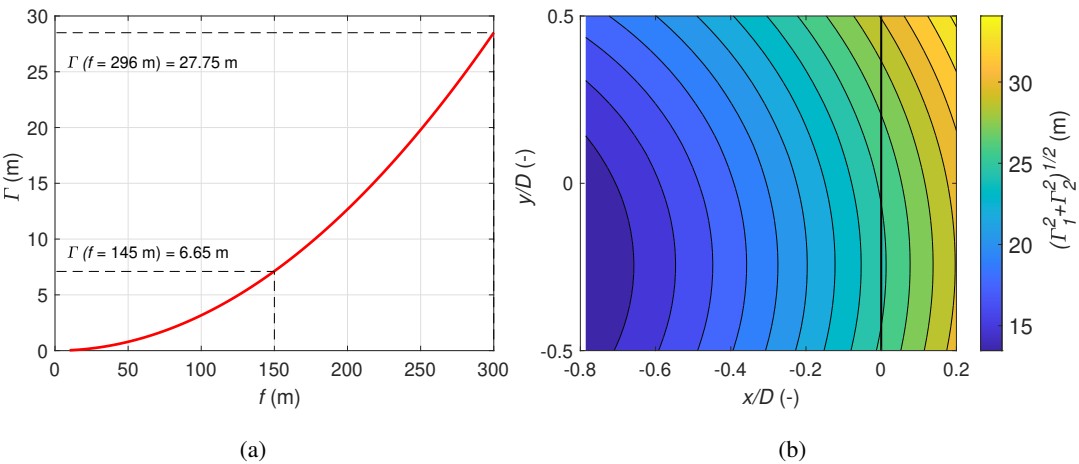

(a)                                                    (b)

**Figure 4.** (a) The variation of the lidar measurement volume with focus distance. The dashed lines indicate the minimum and maximum measurement ranges, and (b) the variation in the effective lidar measurement volume within the scanning area.

While performing scanning cw measurements, a variable measurement volume exists throughout the scan area. For the horizontal scans, the WindScanners measured at distances from 145 m to 296 m, corresponding to a probe volume ranging from 6.65 m (0.05 $D$) to 27.75 m (0.21 $D$), as shown in Fig. 4 (a, b). The WindScanners with their larger 6" aperture and shorter focus rods enabled the probe volume to remain below 30 m (0.24$D$) even at the maximum 300 m range in comparison to the previously used 3" WindScanners with smaller aperture (van Dooren et al., 2017). The probe volume averaging effect is a significant source of uncertainty, especially at considerable focus distances, as it can lead to a measured wind speed bias in a sheared flow. This effect concerns our study as is most severe for measurements at the wake edges, as the measurement volume extends from inside the wake to the freestream, and for measurements very close to the downstream turbine WT2, as the measurement volume would extend partially into the turbine wake. Due to range weighting, velocity measurements are subject to spatial filtering that attenuates the high-frequency wind information, which makes estimates of small-scale turbulence challenging at large focal distances.

### 2.3.2 Dual-lidar errors and uncertainties

Next, we discuss dual-Doppler pointing accuracy, which concerns the ability to steer the focused laser beam to a predefined point in space. To enable dual-Doppler wind field reconstruction, the laser beams from the two WindScanners must focus and spatially and temporally synchronised with each other. The scanner orientation and levelling were thoroughly checked in a controlled laboratory. The final calibration of the steering motors was performed using the turbine tower and a rotating setup as hard targets, and by locating the laser beams using an infrared sensor card (Fig. 2 (b)). A pointing accuracy of 0.1° was determined in the field from the commanded and actual positions of the motors steering the prism. The temporal synchronisation of WindScanners was validated in a previous wind tunnel campaign by van Dooren et al. (2022) and in the field by Giyanani et al. (2022). Giyanani et al. (2022) also estimated similar ranges for the pointing accuracy and calculated the effective intersection diameter at the intersection volume of laser beams to be in the order of 2 m to 5 m.

Due to the spatial and temporal variation in turbulence and the scanning strategy that requires a finite amount of time to complete each scan, the dominant flow features in the induction zone would not be revealed until a multiple scans are collected and averaged. The chosen averaging period must allow the mean velocity measurements to converge while maintaining similar flow conditions throughout the scan duration. Simley et al. (2016) showed that for their measurements where each longitudinal scan took 10 seconds to complete, the dominant flow features were revealed after averaging for at least 3 minutes (18 scans) while the results were presented as 10-minute (60 scans) averages. In our setup, owing to the active toggling of the yaw controller on WT1, the inflow into WT2 changed every 35 minutes; hence, a maximum of only 71 complete scans were available for ensemble averaging over 35 minutes. The ability of WindScanners to capture salient flow features in the induction zone is further investigated in Section 3.1 through statistical uncertainty analysis.

The error in the dual-Doppler reconstruction is dependent on the relative alignment $R_{\text{int}}$ of the laser beams to each other, which depends on the lidar position and measurement trajectory (Stawiarski et al., 2013; Peña and Mann, 2019). If the two laser beams are aligned with each other and with the main wind direction, the longitudinal wind component can be estimated accurately, whereas the orthogonal wind speed component cannot be accurately reconstructed. In other words, when the intersection angle

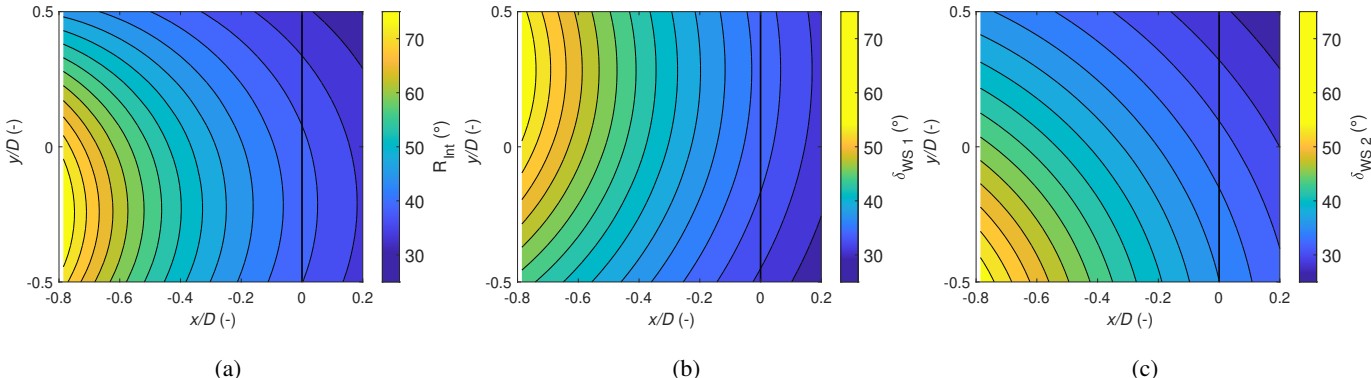

**Figure 5.** The variation of the beam intersection angle $R_{\mathrm{int}}$ (a) and the elevation angles, $\delta_{\mathrm{WS1}}$ (b), $\delta_{\mathrm{WS1}}$ (c) in the scanning area. WT2 is represented by a black vertical line at $x/D = 0$.

tends towards 0° or 180°, the lateral component cannot be resolved. Figure 5 (a) illustrates the variation of $R_{\mathrm{int}}$ in the scan plane which decreases from 68° at $x/D$ = -0.8 to 34° at $x/D = 0.2$.

The rotation in the wake of the upstream turbine induces a non-negligible vertical component in the flow. Therefore, the
$w = 0$ m/s assumption to obtain Eq. (2) contributes to an error in dual-Doppler reconstruction. As the WindScanners are programmed to scan at the WT2 hub height (137 m), the corresponding elevation angles for WS1 (28° to 55°) and WS2 (25° to 56°) introduce a directional bias (Fig. 5 b,c). Hence in Eq. (1), the spatial variation of the non-zero vertical component and the corresponding $\sin(\delta)$ terms are a major error source, especially at measurement points with large elevation angles. While installing the lidars closer to WT1 would reduce the required elevation angles, the lidar position was dictated and limited by
the maximum achievable 300 m of range and available installation area.

Furthermore, we calculate the total uncertainty in the estimation of the longitudinal ($e_u$) and lateral ($e_v$) wind components by applying the Standard Uncertainty Propagation (SUP) method (Stawiarski et al., 2013; van Dooren et al., 2017) on Eqns. 3 and 4. Assuming small errors and zero correlation between them, this method considers the total propagated uncertainty in the dual-Doppler reconstruction due to beam intersection angles, pointing errors, and line-of-sight estimation errors due to
neglecting the vertical flow component, and is described by the following equations:

$$
\begin{aligned}
e_u = \Bigg[ \left( \frac{\partial u}{\partial v_{\mathrm{los},1}} e_{\mathrm{v_{los},1}} \right)^2 &+ \left( \frac{\partial u}{\partial v_{\mathrm{los},1}} \sin(\delta_1) e_{\mathrm{w}} \right)^2 + \left( \frac{\partial u}{\partial v_{\mathrm{los},2}} e_{\mathrm{v_{los},2}} \right)^2 + \left( \frac{\partial u}{\partial v_{\mathrm{los},2}} \sin(\delta_2) e_{\mathrm{w}} \right)^2 + \left( \frac{\partial u}{\partial \chi_1} e_{\chi_1} \right)^2 + \\
&\left( \frac{\partial u}{\partial \chi_2} e_{\chi_2} \right)^2 + \left( \frac{\partial u}{\partial \delta_1} e_{\delta_1} \right)^2 + \left( \frac{\partial u}{\partial \delta_2} e_{\delta_2} \right)^2 \Bigg]^{1/2}
\end{aligned} \tag{7}
$$

$$
\begin{aligned}
e_v = \Bigg[ \left( \frac{\partial v}{\partial v_{\mathrm{los},1}} e_{\mathrm{v_{los},1}} \right)^2 &+ \left( \frac{\partial v}{\partial v_{\mathrm{los},1}} \sin(\delta_1) e_{\mathrm{w}} \right)^2 + \left( \frac{\partial v}{\partial v_{\mathrm{los},2}} e_{\mathrm{v_{los},2}} \right)^2 + \left( \frac{\partial v}{\partial v_{\mathrm{los},2}} \sin(\delta_2) e_{\mathrm{w}} \right)^2 + \left( \frac{\partial v}{\partial \chi_1} e_{\chi_1} \right)^2 + \\
&\left( \frac{\partial v}{\partial \chi_2} e_{\chi_2} \right)^2 + \left( \frac{\partial v}{\partial \delta_1} e_{\delta_1} \right)^2 + \left( \frac{\partial v}{\partial \delta_2} e_{\delta_2} \right)^2 \Bigg]^{1/2}
\end{aligned} \tag{8}
$$

where $e_{v_{\mathrm{los},1}}, e_{v_{\mathrm{los},2}}$ are line-of-sight errors, $e_{\mathrm{w}}$ is the error due to assumption of zero vertical velocity, i.e., the true value of $w$ and $e_{\chi_1}, e_{\chi_2}, e_{\delta_1}$ and $e_{\delta_2}$ are lidar pointing errors. While SUP can be used to understand the influence of different aspects concerning measurement accuracy, not all errors can be studied in detail due to lack of references. Therefore, we also used additional lidar simulations to understand and quantify the different errors affecting the dual-Doppler reconstruction. The impact of the measurement volume, averaging times, lidar placement, and trajectory on the measurements was qualitatively investigated in Section 3.1 using a virtual lidar within LES.

## 2.4 Measurements

As the region of interest was the zone between the two turbines, measurements were only performed when the turbines were aligned, i.e. when the wind direction was approximately 228°. We noticed that many measurements were also affected by unfavourable conditions such as rainfall and lower availability of aerosols to backscatter the laser beam. For operational safety reasons, the WindScanners were operated only with on-site personnel supervision. The measurements were further influenced due to the presence of the wind turbine nacelle and the rotating blades that would systematically reduce data availabilites in the scan region. We present exemplary measurements of four cases made during February 2021, which are summarised and tabulated in Table 3.

In Case 1, WT1 was switched off while WT2 was operational; hence, an undisturbed induction zone upstream of the turbine WT2 was expected. In Case 2, the two turbines were active and aligned, creating a full-wake inflow scenario for WT2. Cases 3 and 4 are measurements conducted while the wake steering control was active on the upstream turbine with averaged measured yaw offsets of 12.8° and -8.95°, respectively, creating a partial wake scenario at WT2.

**Table 3.** Summary of the measurement cases. Each case is characterised by its freestream wind speed $u_\infty$, turbulence intensity (TI), mean wind direction ($\theta_{\mathrm{wdir}}$), stability parameter ($z/L$), stability, wind veer ($\gamma$), vertical wind shear ($\alpha_{\mathrm{shear}}$) and the yaw offset of the turbines ($\gamma_{\mathrm{WT}}$).

| Case | Time (UTC) | $u_\infty$ (m/s) | TI (%) | $\theta_{\mathrm{wdir}}$ (°) | $z/L$ (-) | Stability | $\gamma$ (°) | $\alpha_{\mathrm{shear}}$ (-) | WT1 | WT2 | $\gamma_{\mathrm{WT1}}$ (°) | $\gamma_{\mathrm{WT2}}$ (°) |
|------|-----------|------|------|--------|-------|-----------|------|------|-----|-----|------|------|
| 1 | 24.02 14:14 - 15:07 | 8.51 | 9.30 | $225.35 \pm 9.05$ | 0.040 | Weak Stable | 2.11 | 0.21 | Off | On | 3.73 | 0.95 |
| 2 | 25.02 08:33 - 09:08 | 9.60 | 7.68 | $225.35 \pm 7.11$ | 0.003 | Near Neutral | 19.84 | 0.38 | On | On | -1.25 | 0.60 |
| 3 | 25.02 09:43 - 10:15 | 8.11 | 7.80 | $217.10 \pm 9.78$ | -0.034 | Near Neutral | 13.39 | 0.23 | On | On | 12.80 | 2.22 |
| 4 | 24.02 17:36 - 18:02 | 8.68 | 5.39 | $227.17 \pm 4.44$ | 1.307 | Strong Stable | 19.41 | 0.49 | On | On | -8.95 | -1.10 |
| LES | 35 mins | 7.77 | 6.7 | $228 \pm 4.44$ | - | Strong Stable | 20.7 | 0.44 | On | On | 0 | 0 |

The freestream wind speed $u_\infty$, turbulence intensity TI, wind direction $\theta_{\mathrm{wdir}}$ and its standard deviation were calculated using the anemometer and wind vane at the hub height of WT1 placed on the met mast. The atmospheric stability of the boundary layer can be characterised well by the Monin-Obhukov similarity theory (Monin and Obukhov, 1954; Barthelmie, 1999). The Obhukov parameter ($z/L$) was measured by the eddy covariance station at a height of 6 m above the ground. The Monin-

Obhukov length was calculated as:

$$L = \frac{-u_*^3 \theta_s}{kg(\overline{w'\theta_s'})}, \tag{9}$$

where $u_*$ denotes the friction velocity, $k = 0.4$ denotes the von Kármán constant, $g$ denotes the acceleration due to gravity, $\theta_s$ denotes the sonic temperature, and $\overline{w'\theta_s'}$ denotes the buoyancy flux. The friction velocity is estimated as $u_* = (\overline{u'w'}^2 + \overline{u'v'}^2)^{\frac{1}{4}}$. The stability classification of the Obhukov parameter $z/L$ is performed for 30-minute averages based on Wyngaard (2010), where negative values indicate the presence of unstable conditions ($z/L \leq -0.04$), positive values ($z/L \geq 0.4$) correspond to stable conditions, and values close to zero ($-0.04 \leq z/L \leq 0.4$) are related to neutral conditions.

The wind shear profile was also estimated from the VAD lidar by fitting a shear exponent $\alpha_{\mathrm{shear}}$ based on the power law between the top and bottom blade tips. The test site experienced larger than expected values of wind shear with an average value of 0.3 throughout the measurement campaign (Sengers et al., 2023). The wind veer $\gamma$ was calculated from the VAD lidar as the difference in wind direction between the top and bottom blade tips, and was clockwise positive. The actual yaw offset $\gamma_{\mathrm{WT}}$ was calculated by subtracting the GPS measured WT1 heading from the wind direction at the hub height measured from the met mast as follows:

$$\gamma_{\mathrm{WT1}} = \theta_{\mathrm{GPS},1} - \theta_{\mathrm{wdir}} \qquad \text{and} \qquad \gamma_{\mathrm{WT2}} = \theta_{\mathrm{GPS},2} - \theta_{\mathrm{wdir}}. \tag{10}$$

A positive yaw misalignment was identified when the turbine was rotated clockwise looking from the top (Fig. 3).

## 2.5 Numerical simulations of the experimental site

Before interpreting results, it is necessary to quantify the effect of lidar measurement error and uncertainties discussed in Section 2.3. To this end, we modelled the wind farm and inflow conditions in a simulation environment. The wind data are obtained from high-fidelity LES runs where the performance of two virtual WindScanners were assessed. The wind field was created using high-resolution LES performed with the Parallelised Large-Eddy Simulation Model (PALM). The PALM code is widely used for atmospheric boundary layer studies and works by solving the filtered, incompressible, non-hydrostatic Navier-Stokes equations. Further details of the model are available in Maronga et al. (2015). A single stably stratified LES run was performed and the two eno126 turbines are simulated with the actuator sector method using the Fatigue, Aerodynamics, Structures and Turbulence code (FAST) v8 (Jonkman et al., 2005), by the National Renewable Energy Laboratory (NREL) that is directly coupled with the LES (Krüger et al., 2022) allowing for the transfer of forces and velocities between the two simulations. The turbine FAST model was built using the aerodynamic properties, tower properties, and turbine controller provided by the farm operator. The eigenfrequencies of the FAST model of the two turbines are further tuned based on load data measured during the experiments. The WindScanners were simulated using the integrated lidar simulator (LiXim) developed by Trabucchi (2020) which can simulate lidar kinematic and optical properties. LiXim simulates the volume averaging property by discretising Eq. (5) in the LES while the uncertainty in beam pointing and environmental factors are not modelled. An atmospheric boundary layer of stable stratification was simulated in a domain of dimensions 81 $D$ x 20 $D$ x 3.8 $D$ with a uniform grid spacing of 5 m. Turbulence recycling (Lund et al., 1998) was applied at a distance of 15 $D$ from the inlet, where

the instantaneous wind fields of the precursor simulation are introduced into the main simulation. The potential temperature at the ground was set to 280 K. A temperature gradient of 1 K/100 m was prescribed from 100 m above the ground while the simulation was performed for 4800 s sampled at 5 Hz. For the analysis, the first 600 s of the simulation were removed to avoid transient effects, and only the final 35 simulation minutes were utilised to correspond with the field measurements. The terrain was modelled by prescribing a ground roughness length of 0.1 m. The simulated wind field has a mean wind speed at hub height $u_\infty$ = 7.77 m/s and a TI = 6.7%. The stable atmospheric boundary layer (ABL) is characterised by a strong shear exponent $\alpha_{\mathrm{shear}}$ = 0.44 and a wind veer of 20.7° between the top and bottom rotor tips. The virtual WindScanners are programmed to perform horizontal plane scans similar to the experimental setup following Eq. (6). The two operational turbines aligned in the prevailing wind direction in the LES resembled a full-wake scenario at WT2.

## 3 Results

This section is divided into two parts. In the first section, we show the results of the virtual WindScanner simulations in the LES and estimate the uncertainty associated with the dual-Doppler reconstruction. The results from the field measurements are presented in the second section. As the measurement plane extends 0.4 $D$ downstream of WT2, laser beam blockage due to blade rotation was expected. During post-processing, it was discovered that the data quality for the measurements at $0.2 \leq x/D \leq 0.4$ was poor and hence was discarded for both LES and field measurements. Data filtering for the field measurements was performed using a kernel density-based filter based on Beck and Kühn (2017) to identify and remove low-quality measurements. A comparison against engineering models of the induction zone is shown only for the undisturbed induction case.

### 3.1 Virtual WindScanner Evaluation in LES

Line-of-sight simulations of the two WindScanners are performed using LiXim and the LES flowfield, after which a dual-Doppler reconstruction is applied to resolve the longitudinal and lateral wind fields. In Fig. 6, reconstructions of the Wind-Scanner estimated 35-minute averaged longitudinal, lateral wind profiles are presented alongside the reference LES. A good qualitative agreement between the LES and the virtual WindScanner resolved $u, v$ profiles are noted at most parts of the scanning area. The simulations reveal that the WindScanners can capture the spatial features in the flow such as the wake rotation and flow expansion at the rotor tips. For the $u$ profiles, the velocity profiles show deviations from the LES reference, presumably because of the directional bias induced by the large elevation angles and the probe volume extending through the shear layer and from the wake into the freestream. The lateral velocity profiles illustrating the wake rotation and flow expansion are captured well by the WindScanners. The profiles also indicate that the dominant flow structures in the induction zone are captured well for an average duration of 35 minutes when similar wind conditions are maintained for the scan duration.

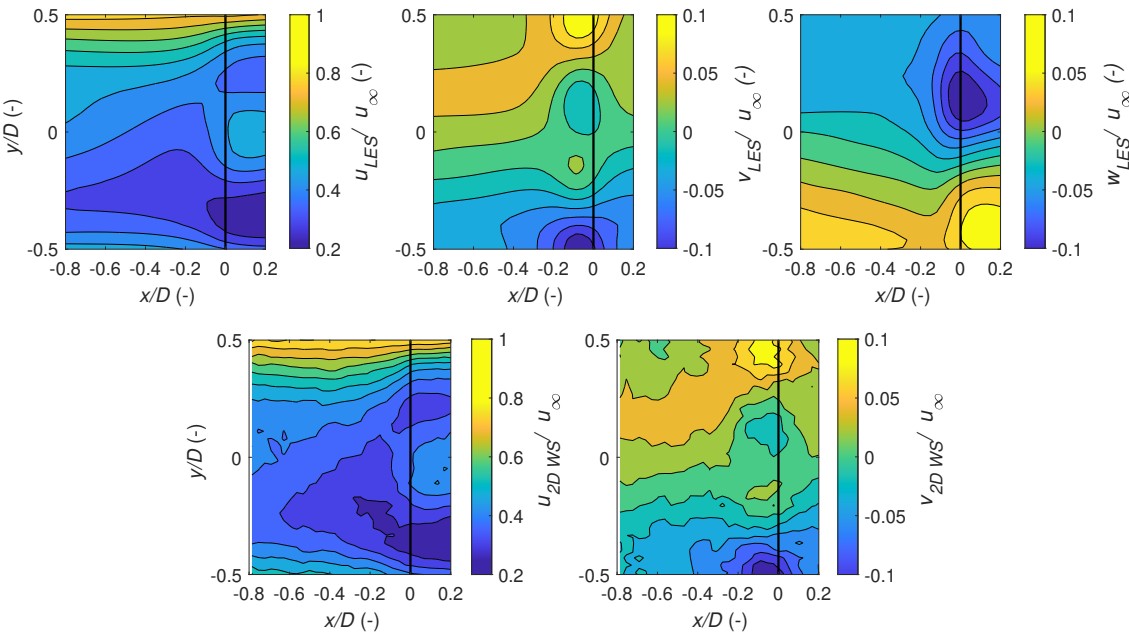

**Figure 6.** Longitudinal ($u$), lateral ($v$) and vertical ($w$) velocities on the horizontal plane from the LES (1$^{\text{st}}$ row) and the results of 2D WindScanner reconstruction inside the LES (2$^{\text{nd}}$ row) both averaged for the last 35 minutes of the simulation. The black vertical line at $x/D = 0$ is the rotor of WT2.

### 3.1.1 Statistical Uncertainty

First, we discuss statistical uncertainty. While the total propagated uncertainty regards the propagation of uncertainty of single variables through the dual-Doppler reconstruction, the statistical uncertainty quantifies the precision of the results from different scans with a higher number of scans typically reducing measurement noise from the statistical error. To quantify the statistical uncertainty, we use the margin of error estimated in the scanning area. The margin of error was calculated as $e_{\text{u,stat}} = \frac{z_\gamma \sigma_{\text{u}}}{\sqrt{N_{\text{s}}}}$ and $e_{\text{v,stat}} = \frac{z_\gamma \sigma_{\text{v}}}{\sqrt{N_{\text{s}}}}$. Here $z_\gamma$, the confidence level, is set to 1.96, denoting the 95 % confidence interval, $\sigma_{\text{u}}, \sigma_{\text{v}}$ are

the standard deviations of the longitudinal and lateral velocity components in the scan plane obtained from the WindScanner simulations and $N$ is the number of samples.

Figure 7 shows the variation of the margin of error in the scanning area for the two reconstructed components normalised with the mean longitudinal wind speed. The margin of error for the longitudinal component varies in the scan area between 2 % to 5 % depending on the turbulence intensity in the wake. Similarly, for the $v$- component, the margin of error varies between

345 1.3 % and 2.5 %. The higher errors at scan edges could be attributed to the low amount of data points in these locations as a consequence of the scanning patterns. In the field, we expect that the margin of error would be slightly higher than in the idealised LES due to the filtering procedure reducing data availability in each scan.

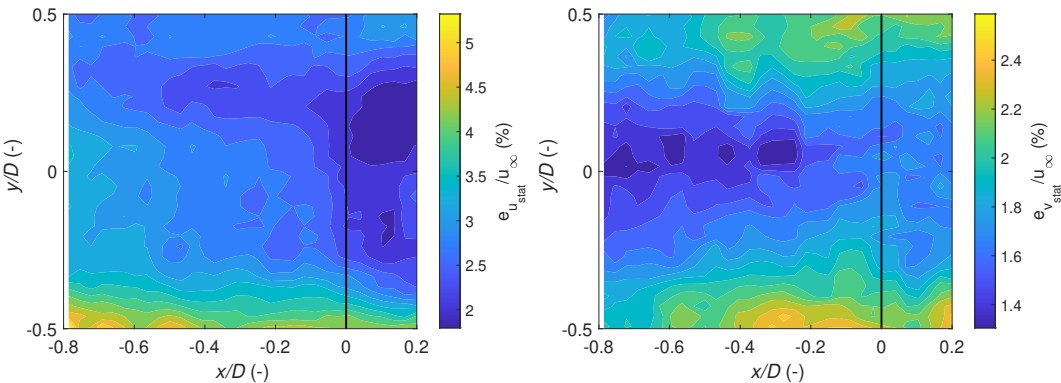

**Figure 7.** Statistical uncertainty estimated through the normalised margin of error for the $u, v$ components for the evaluated flow field.

### 3.1.2 Dual-Doppler Propagated Uncertainty in LES

Secondly, we discuss the total propagated uncertainties estimated through the SUP method. The total propagated errors in the estimation of the longitudinal and lateral wind speed components are performed based on Eqs. 7 and 8 and illustrated in Fig. 8. To study the influence of spatial velocity variation in the wake, the actual LES $w$ component in Fig. 6 is used. The normalised $u$ component estimation error $e_u$ varied between 2 % and 15 % of $u_\infty$. As expected, $e_u$ is large at the WT2 rotor plane for the locations exhibiting higher $w$ velocities. $e_u$ is highest at the scan location closest to WS1 with the highest elevation angles

whereby the lidars could only measure a small projection of the longitudinal wind speed. Similar behaviour is seen for the $e_v$ as well ranging from 5 % to 14 % of $u_\infty$ in the scanning area with the highest values seen where larger $w$ velocities are present and at the scan area where the beam intersection angles are the lowest (Fig. 5).

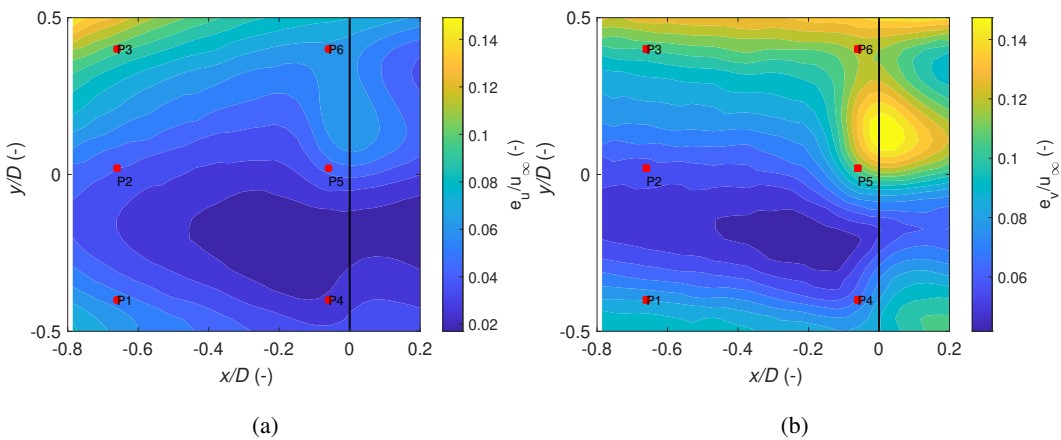

**Figure 8.** Dual-Doppler reconstruction error for the (a) longitudinal ($e_u$) and (b) lateral ($e_v$) components for the evaluated flow field. The markers P1 - P6 indicate regions of interest.

Figure 9 illustrates the ratio of contributions of the different terms in Eq. 7 and Eq. 8 and the total error for the respective
flow component visualised for six locations (P1, P2,... P6) as marked in Fig. 8.

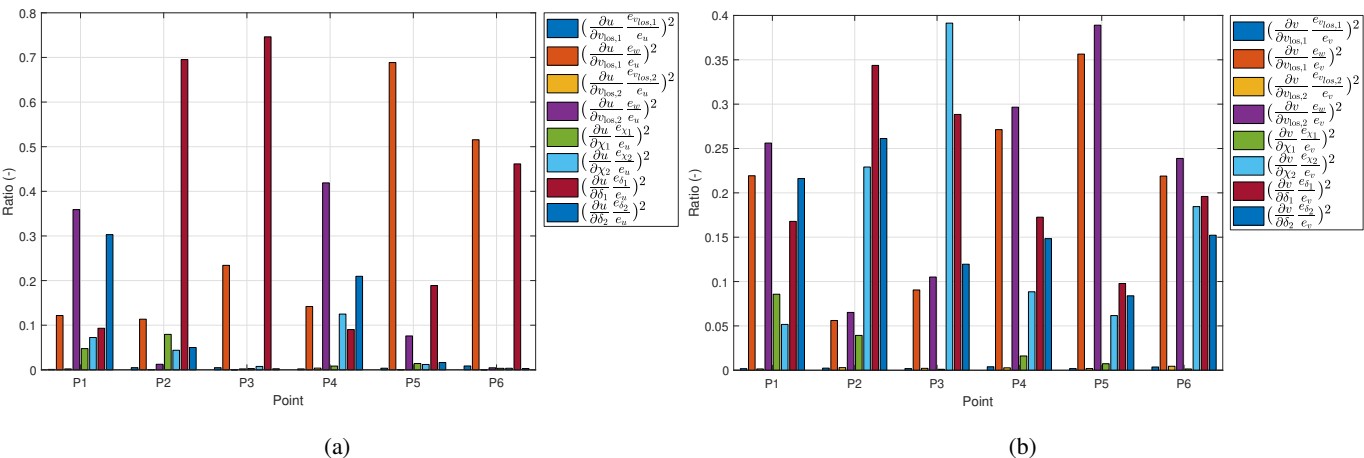

**Figure 9.** Dual-Doppler reconstruction error for the (a) longitudinal ($e_v$) and (b) lateral ($e_v$) components for the evaluated flow field. The
markers P1 - P6 indicate regions of interest.

To analyse the contribution of certain measurement errors, the standard uncertainty propagation is evaluated for an error in
$v_{\text{los}}$ of 0.1 %, a pointing accuracy of 0.1° while the error introduced due to neglecting the vertical flow component is obtained
from the local LES $w$ component. The magnitude of the individual error contributions is normalised by the total error ($e_{\text{u}}$,
$e_{\text{v}}$) to obtain the contribution of each term to the total error. For $e_{\text{u}}$, the following trends are noticed. The line-of-sight error
$e_{\text{vlos,i}}$ contribution is almost negligible for all 6 points. The error due to the $w$ component assumption $e_{\text{w,i}}$ has a significant
contribution to $e_{\text{u}}$, especially at P4, P5 and P6 due to the large local $w$ at these locations. At P1, P2 and P3, $e_{\delta_i}$ has a large
contribution due to the severe elevation angles required to scan at these points and the positive correlation between $\frac{\partial u}{\partial \delta}$ and $\delta$.
The varying contributions of $e_{\text{w,i}}$ at the points of interest can be explained by the relative alignment of the lidar with the wind
direction. For a non-zero $w$ component, an aligned lidar will contain a larger contribution of the $w$ component projected onto
its line-of-sight compared to the un-aligned case. At P1 and P4, the contribution of $e_{\text{w,2}}$ is the largest as WS2 is more aligned
with the longitudinal wind component in comparison to WS1. Similarly, at P3, P5 and P6, WS1 is approximately aligned with
the longitudinal wind speed component. So the errors at these points are dominated by the $e_{\text{w,1}}$, which is highest at P5 due to
the large local $w$ velocity in the LES field. For $e_{\text{v}}$, it is clear that the errors are preliminarily driven by the $e_{\text{w}}$ while $e_{\text{vlos,i}}$ is
almost negligible. However, the contributions of $e_{\chi,i}$ and $e_{\delta,i}$ are larger compared to that of $e_{\text{u}}$ highlighting the sensitivity of
the pointing angles for the lateral component reconstruction.

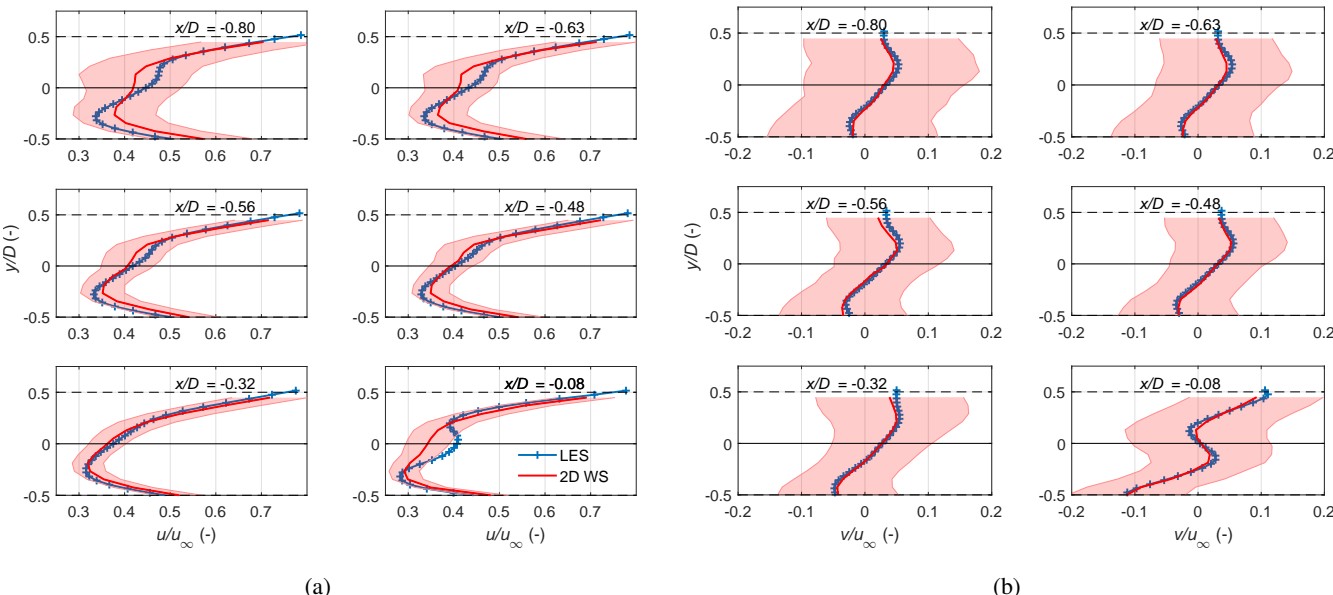

**Figure 10.** WindScanner estimated velocity profiles (red) of longitudinal (a) and lateral (b) velocities at six upstream cross sections compared against the reference LES (blue). The red shaded area indicate the uncertainties estimated through Eqn. 7 and Eqn. 8 using the local $w$ velocity component.

To visualise the reconstruction accuracy, horizontal velocity profiles are extracted at six upstream streamwise cross sections and compared against LES in Fig. 10. The shaded region illustrates the total propagated uncertainty for the dual-Doppler reconstruction. The longitudinal velocity profiles measured by the virtual WindScanners at $x/D < -0.63$ exhibit deviations from the LES due to the large elevation angles required for scanning, while the estimation errors reduce towards WT2. Similarly, very close to the downstream turbine at $x/D = -0.16$, the WindScanner measured $u$ profile is lower compared to the LES close to the rotor axis as the measurement volume extends behind the rotor while scanning very close to the rotor plane. At $x/D = -0.8$, the maximum $u$ error is 11.6 %, while the error reduces in the downstream direction with a maximum error of 9.7 % at $x/D = -0.63$. The WindScanner-measured $u$ velocity profiles at $x/D = -0.56$, $x/D = -0.48$, and $x/D = -0.32$ agree well with the LES. Moving further downstream, the difference in the intersection angles of the two lidars decreases. Therefore, the $u$ component is estimated better on the scan's downstream side as the laser beams align with the prevailing wind direction with reducing elevation angles. While the intersection angle reduces towards WT2, the velocity profile at $x/D = -0.16$ shows slightly larger error bars due to a large vertical wind speed component resulting from local aerodynamic effects close to the rotor plane of WT2. The lateral velocity component profiles show a good agreement with the LES, with minor differences seen at the scan edges. The error bars around the $v$ component profiles are larger than the differences in the LES and WindScanner resolved profiles due to the inclusion of multiple error terms in the SUP. This indicates that the WindScanners can resolve the 2D velocity profiles with the current setup. While using the local $w$ component in the SUP, it is seen that the observed velocity reconstruction errors are dependent on both the scanning strategy and the flow dynamics.

### 3.1.3 Dual-Doppler Propagated Uncertainty in Field Measurements

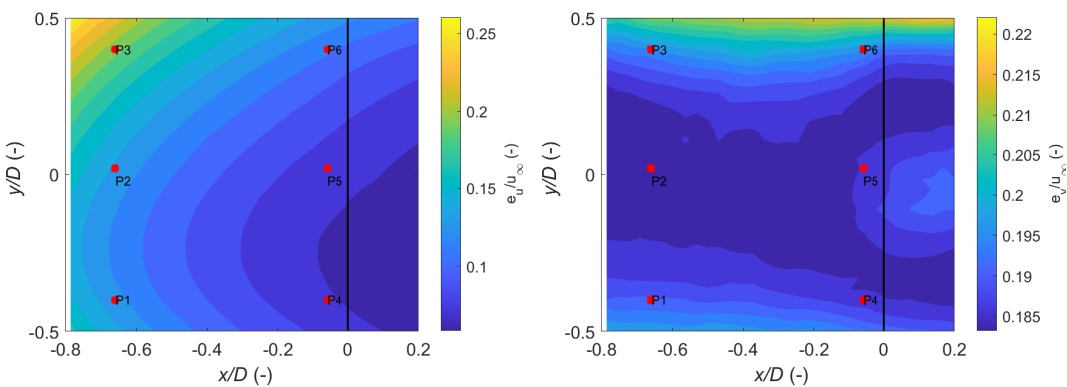

**Figure 11.** Variation of $e_u$ and $e_v$ in the scanning area using the assumption of $w = 1$ m/s in the SUP.

In the presented analysis, the reference LES fields provided an accurate value of the local $w$ component variation that is subsequently used to calculate the propagated uncertainties allowing to investigate the lidar error inside the LES. However, in the free field, no measurements of the local $w$ component would be available. Therefore, an assumption of a constant $w$ is required to estimate the propagated uncertainties. For the full wake and partial wake cases, a value of $w = 1$ m/s is taken, similar to the model turbine in wind tunnel experiments of van Dooren et al. (2017). Figure 11 illustrates the variation of $e_u$ and $e_v$ in the scanning area using the assumption of $w = 1$ m/s. The comparison of $e_u$ and $e_v$ using the local LES $w$ velocity (Fig. 8) and the assumed vertical velocity (Fig. 11) reveals two significant differences. Firstly, assuming constant $w$ on the scanning area masks the velocity reconstruction error that is dependent on the flow dynamics, especially close to the rotor. Secondly, the magnitude of $e_u$ and $e_v$ for the constant $w$ velocity case is substantially larger than the errors estimated using the local $w$ velocities. For both approaches, the statistical uncertainty is significantly lower in comparison to the propagated uncertainties.

## 3.2 Measurement Results

This section illustrates and discusses the field measurements for the four measurement cases covering scenarios from undisturbed inflow to full and partial wake scenarios as described in Table 3.

### 3.2.1 The undisturbed induction zone

Figure 12 shows the averaged longitudinal and lateral wind velocities extracted from the WindScanner measurements of Case 1 in Tab. 3 with a mean wind speed of 8.51 m/s and a weakly stable stratification. The non-operating upstream turbine had an average yaw misalignment of 3.73°, whereas the downstream turbine had an average misalignment of 0.95° during the measurement period. The extent of the induction zone can be visualised by the $u$-component deceleration and is very strong within $-0.6 \leq x/D \leq 0$ upstream of WT2. This strong velocity deficit can be attributed to high axial induction and weakly

stable stratification during the measurement period inhibiting vertical turbulent mixing. The induction effect is strongest at the inboard blade stations and decreases towards the blade tips. The induction zone also exhibits a slightly asymmetrical distribution between the left ($y/D > 0$) and right sides ($y/D < 0$) of the rotor.

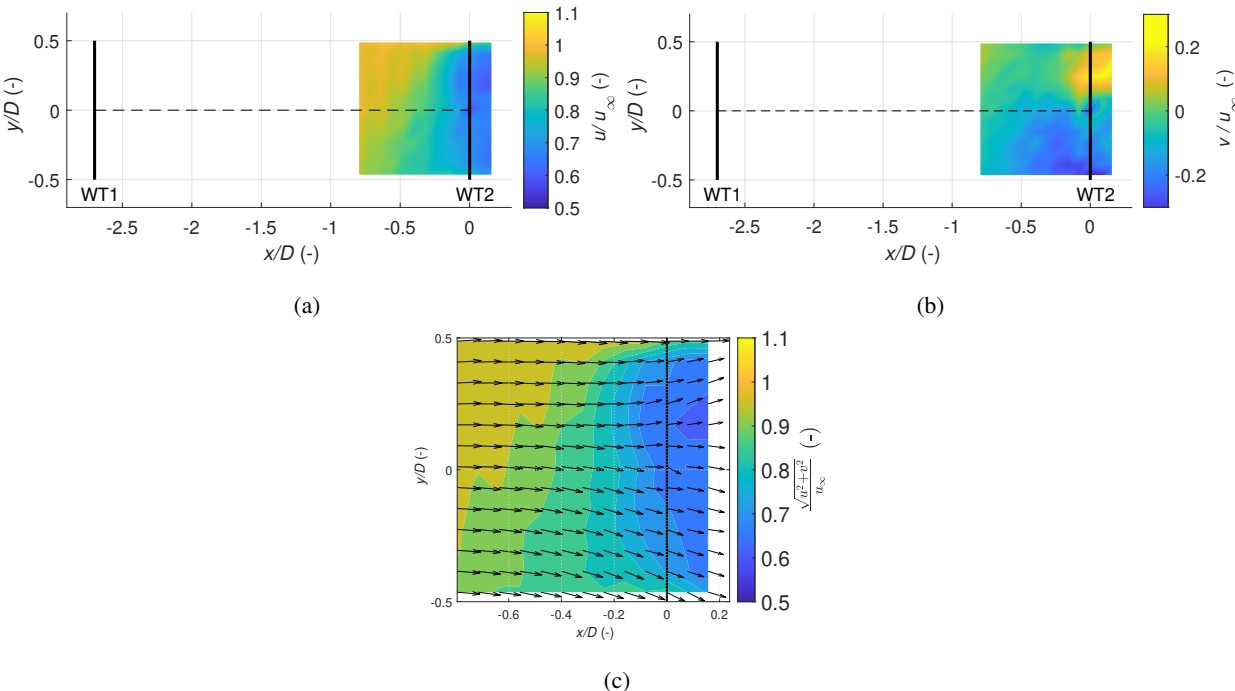

(a)
(b)
(c)

**Figure 12.** Case 1: Longitudinal (a) and lateral (b) velocities measured while WT1 was not operating and WT2 was operational. In (c), the quiver plot is based on the measured horizontal velocities.

Looking downwind, this slight asymmetry could be attributed to the presence of a tall treeline in-between WT1 and WT2 per-
turbing the flow by acting as a windbreak (Counihan et al., 1974; Tobin et al., 2017) and the strong vertical shear $\alpha_{\text{shear}} = 0.21$ that causes a vertical wind speed gradient varying the relative wind speed and the angle of attack of the blades during a rotation. Additionally, the induced velocities at the rotor plane are influenced by the counter-rotating wake creating a momentum transfer between the lower and upper rotor regions leading to a difference in flow magnitude between $y/D > 0$ and $y/D < 0$ (Madsen et al., 2012). Hence, the blade sections would experience varying blade forces that vary the local thrust coefficient,
and the corresponding deceleration. The lateral velocity component is non-zero close to the blade tips, indicating a flow expansion around the rotor. The large lateral velocities present close to the rotor plane can be attributed to the lower data availability due to blade passage, improper tracking of wind direction by WT2 influencing the induced velocities that the WindScanners measuring complex aerodynamic phenomenon with a large probe volume and reconstructed by neglecting the vertical wind component. In Figure 12 [c], the $u, v$ wind components within the scanning plane are combined to illustrate the wind direction
behaviour in the scan plane, exhibiting an induction zone asymmetry and flow expansion around the WT2 rotor.

In Fig. 13, horizontal inflow profiles at five upstream distances moving towards WT2 are plotted. The shaded regions indicate the propagated uncertainty bounds calculated for the dual-Doppler reconstruction using Eq. (7), and Eq. (8). Here, a constant vertical component $w = 0.2$ m/s is assumed, as no wakes are propagating from the non-operational upstream turbine and no direct measurements of the $w$ component were available in the scanned area. The $u$ component uncertainty due to the dual-Doppler reconstruction decreases moving toward the rotor. The horizontal profiles at $-0.8 \leq x/D \leq -0.31$ exhibit asymmetrical behaviour with an approximately 5.8 % difference between the left ($y/D > 0$) and right ($y/D < 0$) blade tips, whereas at $x/D = -0.16$, the asymmetry disappears. The lateral velocity profiles show a large magnitude very close to the rotor tips due to the flow expansion.

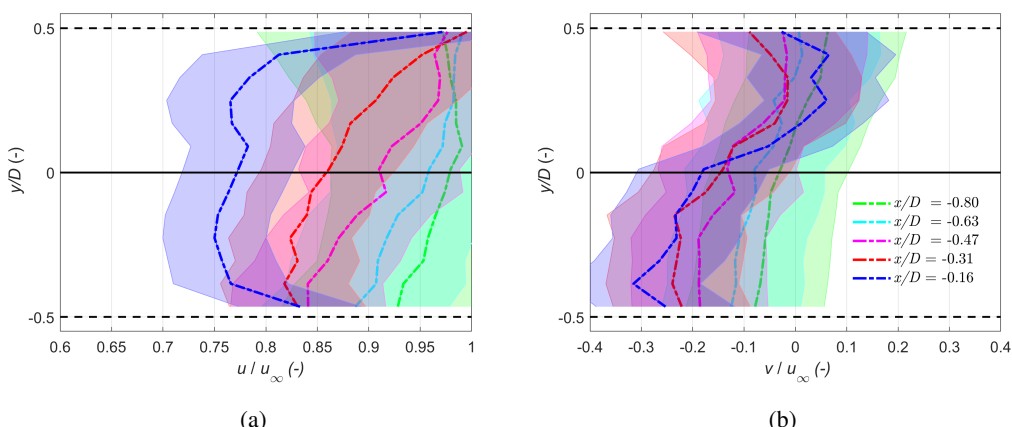

**Figure 13.** Case 1: Inflow longitudinal (a) and lateral (b) profiles extracted at various positions upstream of WT2. The shaded area represents the bounds calculated from the standard uncertainty propagation method ($e_u$) with $w = 0.2$ m/s, as WT1 was not operational.

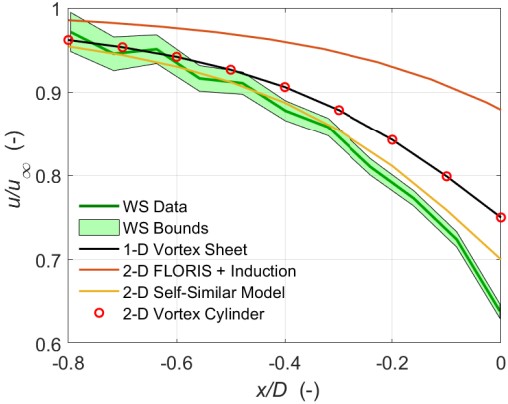

**Figure 14.** Case 1: Comparison of the velocity deceleration along the rotor axis against the predictions from different induction zone models. The green shaded area represents the upper and lower bounds of the measurement.

We compared the different induction zone models against the measurements along the rotor axis, as illustrated in Fig. 14. The upper and lower bounds of the WindScanner represent the propagated uncertainty bounds. Data availability between $0 \leq x/D \leq 0.2$ is reduced due to the presence of the nacelle and therefore excluded from the fit. Also plotted is the velocity deceleration predicted by the 1D vortex sheet theory (Medici et al., 2011) using the freestream velocity measured at the met mast extrapolated to WT2 hub height using the shear exponent and the axial induction factor estimated from the turbine thrust
curve. The model-predicted velocity deceleration falls within the WindScanner bounds till $x/D < -0.4$ while the slowdown is under-predicted close to the rotor plane. Simley et al. (2016) also noted similar bias, the reasons for which were the model does not consider atmospheric stability nor the presence of the tower and nacelle-induced deceleration.
       The velocity deceleration predicted by the Vortex Cylinder model (VC model) (Branlard and Gaunaa, 2015), FLORIS coupled with the Induction model (FLORIS+Induction) (Branlard and Meyer Forsting, 2020) and the self-similar model (Troldborg
and Meyer Forsting, 2017) is also illustrated in Fig. 14 using the inflow conditions in Table. 3 as input parameters. Although these models can predict the upstream velocity deceleration in the horizontal plane, they do not consider the vertical shear. Therefore, only the deceleration along the rotor axis is displayed. The FLORIS+Induction model also utilises said VC method to predict the induction deceleration coupled with the Gaussian wake model in FLORIS accessed from Branlard (2019). As expected, the VC model shows excellent agreement with the 1D vortex sheet results but exhibits an under-prediction of the
velocity decrease compared to the measurements. A similar under-prediction of the velocity decrease by the VC model was noted in Meyer Forsting et al. (2021) as no wake expansion is considered to affect the momentum balance between upstream and downstream of the rotor, which increases with increasing thrust coefficients, is included. Also shown in Fig. 14 are the results of the self-similar model proposed by Troldborg and Meyer Forsting (2017). Along the rotor axis, the model is similar to the VC model but contains an additional thrust-dependant scaling term to correct for the systematically underestimated
axial induction. Applying their thrust correction factor, a better agreement with the WindScanner measurements is obtained until $x/D < -0.4$. The FLORIS+Induction model consistently under-predicts the magnitude of the velocity decrease along the centre line, with the effect becoming more severe towards the rotor. The axial induction and, therefore, the deceleration obtained from the FLORIS model were lower compared to the field measurements.

### 3.2.2   The fully waked induction zone

This section presents the results of Case 2, with $u_\infty = 9.60$ m/s, $\theta_{\mathrm{wdir}} = 225.35° \pm 7.1°$ and wind veer $\gamma = 19.84°$ in a near-neutral stratification. During the measurement period, the upstream turbine was operated by a greedy controller that introduced an average yaw misalignment of $-1.25°$, while WT2 was misaligned with an average of $0.60°$ with the prevailing wind direction. Hence, a full-wake scenario at WT2 is occurring. The WindScanners were programmed to perform horizontal scans at the hub height of WT2. This resulted in scans capturing the WT1 wake on a horizontal plane $0.16\,D$ above the hub height of WT1
owing to the hub height difference.
       Due to the downstream turbine operation, an induction zone deceleration is observed inside the wake between $-0.5 \leq x/D \leq 0$ upstream of the rotor, as shown in Fig. 15. The lateral velocity component is dominated by a lateral flow towards the left side ($y/D > 0$) of the rotor looking downstream. The flow expansion around the downstream turbine can be observed with

stronger lateral velocities on the left side of the rotor ($y/D > 0$) looking downwind. In the region, $-0.8 \leq x/D \leq 0$ and $-0.1 \leq y/D \leq 0.3$, a strong cross-wind component is introduced to the wake that rotates in the opposite direction to that of the clockwise rotating rotor. By combining the $u$ and $v$ velocities, the local wind vector in the horizontal scan plane can be estimated. Plotted in Fig. 15 [c] is the total velocity magnitude $U$ superimposed with streamlines. A clear induction zone is visible centred around the rotor axis in the region $-0.5 \leq x/D \leq 0$ while the wake is expanded around the strong induction. Due to the proximity between the two turbines, an interaction between the induction zone of the downstream turbine with the wake of the upstream turbine is observed, while the wake deficit is further increased as the induction zone blocks and expands the flow around it.

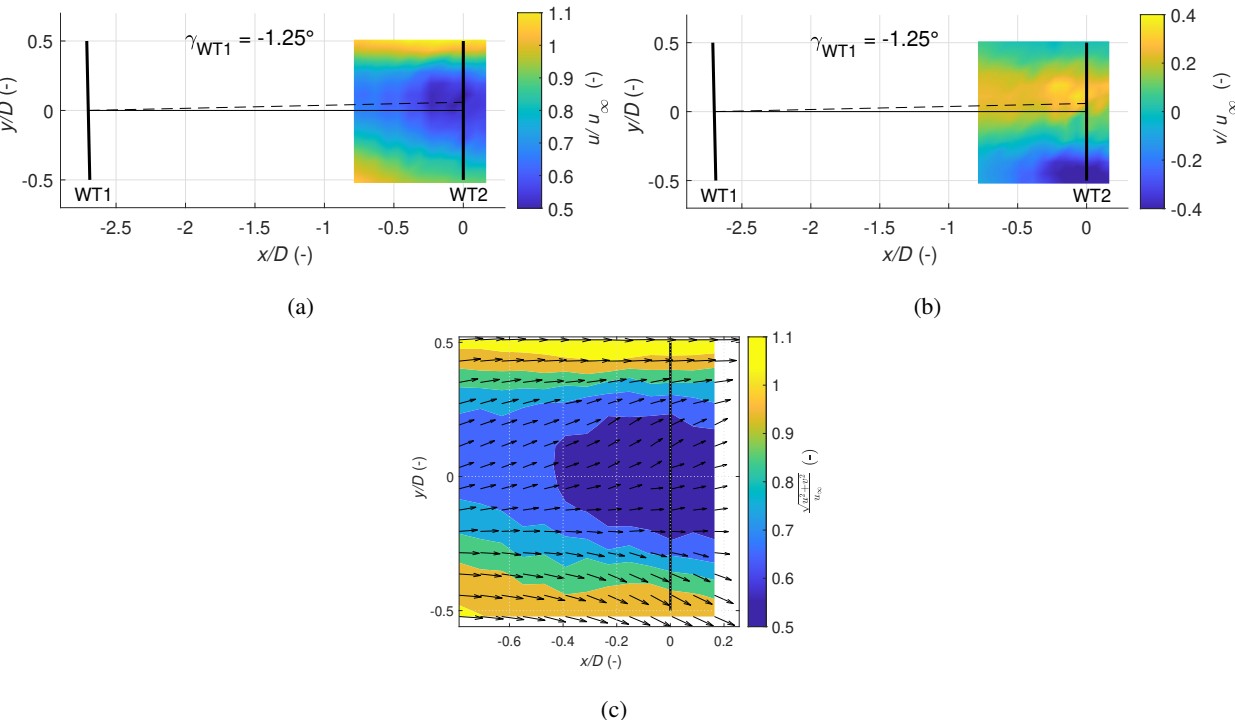

**Figure 15.** Case 2: Longitudinal (a) and lateral (b) velocities measured for the full wake case. The contour of the magnitude of the horizontal velocity and its vector field is plotted in (c).

The horizontal flow profiles were plotted at five locations upstream of WT2, as shown in Fig. 16 to investigate the effect of WT2 induction on the wake profiles. As no measurements of the $w$ component were available, a conservative value of 1 m/s has been utilised to estimate the upper and lower bounds of the profiles. Instead of recovering, the longitudinal velocity profiles show a deceleration towards WT2. The effect of induction is strongest close to the rotor axis between $-0.2 \leq y/D \leq 0.2$ where a velocity reduction of 27 % is observed between $x/D = -0.8$ and $x/D = -0.15$. The lateral velocity component shows a non-zero component between $-0.2 \leq y/D \leq 0.2$, indicating that the flow is pushed towards the blade tips and around the induction

zone. The lateral velocity variations at the blade tips are due to the reduced data availability close to the blades ($x/D \geq -0.15$) due to blade passage and the yaw error of the downstream turbine. The turbine does not follow the wind direction perfectly; hence, time-varying yaw errors can be introduced, which would induce movement of the rotor within the scan area leading to erroneous estimates in the measurements. Interestingly, the spanwise velocity profiles at various upstream positions exhibit a slight asymmetry. This finding might correspond to Sezer-Uzol and Uzol (2013), where asymmetric wake profiles under vertical shear were reported and attributed to the different wake convection speeds between the upper and lower rotor halves. Moreover, the combination of the strong shear layer and wake rotation results in an asymmetric velocity distribution as the wake rotation enhances mixing between the low and high momentum regions of the rotor (Xie and Archer, 2017; Abkar et al., 2018). At all five upstream positions, the wake at $y/D < 0$ exhibited stronger velocity reductions than the wake at $y/D > 0$.

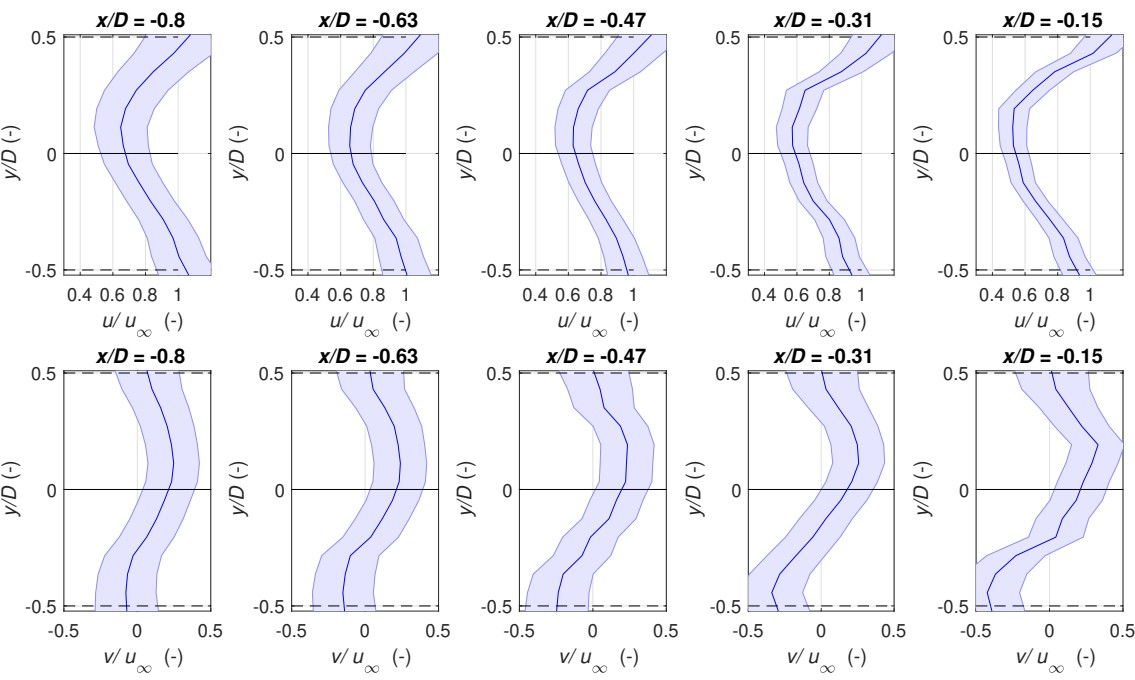

**Figure 16.** Case 2: Longitudinal and lateral velocities measured for the full wake case. The shaded area represents the bounds calculated from the standard uncertainty propagation method ($e_u$) with $w$ = 1 m/s, as WT1 was operational.

### 3.2.3   The partially waked induction zone

Finally, we present the measurements of the induction zone upstream of WT2 during a partially waked condition shown in Fig. 17.

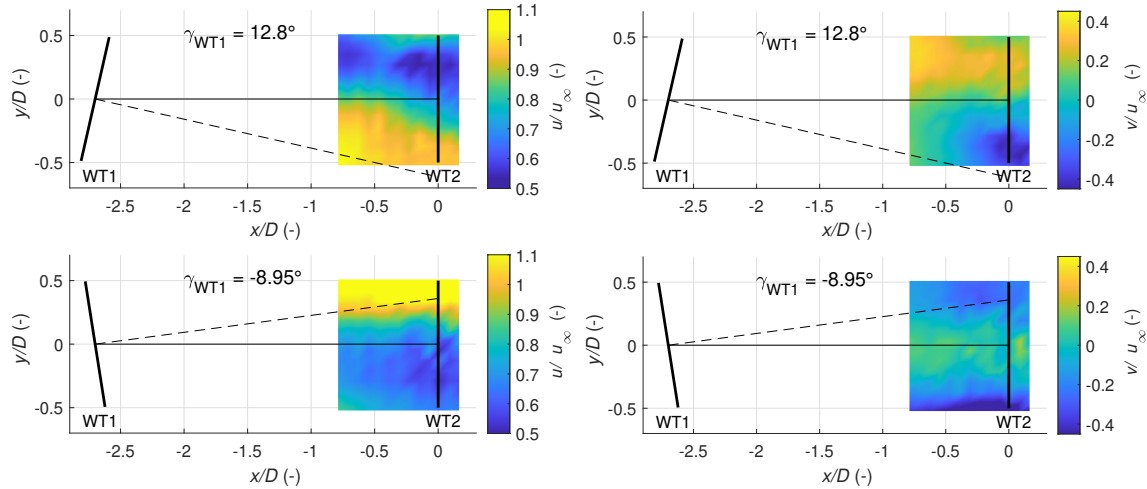

**Figure 17.** Cases 3 and 4: Longitudinal and lateral velocities measured for positive yaw offset (Case 3, 1[st] row) and negative yaw offset (Case 4, 2[nd] row).

The results for both positive and negative yaw offsets of WT1 are illustrated in Fig. 17. For a positive offset (Case 3: $\gamma_{WT1} = 12.8°$), a wake deflection towards the left of the rotor ($y/D > 0$) is observed in the $u$ component looking downstream, while the wake deflects to the right of the rotor ($y/D < 0$) for the negative offset (Case 4: $\gamma_{WT1} = -8.9°$). For Case 3, the lateral velocity component is characterised by a flow towards the left side of the rotor ($y/D > 0$) due to a combination of the counter-clockwise wake rotation and the lateral force applied on the flow due to the intentional yawing of the turbine. The opposite effect is observed for Case 4, where a lateral flow towards the right side of the rotor plane is seen. The findings correspond to Fleming et al. (2018), where a stronger wake deflection for the positive yaw case is seen due to the aggregated effect of the wake rotation and counter-rotating vortices in comparison to the negative deflection case. In both cases, the maximum magnitude of the longitudinal velocity inside the deflected wake is approximately $0.2\ u_\infty$ to $0.25\ u_\infty$ with the positive yaw offset case exhibiting a comparatively more substantial lateral flow component compared to the negative yaw offset. As the measurements are in the near wake region of WT1, the lateral velocity would be additionally influenced by the aerodynamic effects of the rotor while the effect of yaw steering on the lateral component would be dominant further downstream. In both cases, the lateral component increases towards the blade tips to account for the flow expansion close to the downstream turbine. It is noted that for the positive offset case, the spatial distribution of the $u$ component seems to move near the rotor axis instead of deflecting towards $y/D > 0$. This could be potentially attributed to the 10° misalignment between the wind direction and the turbine orientation direction in addition to the large variability of the wind direction from 208° to 223° which was the highest of all investigated cases.

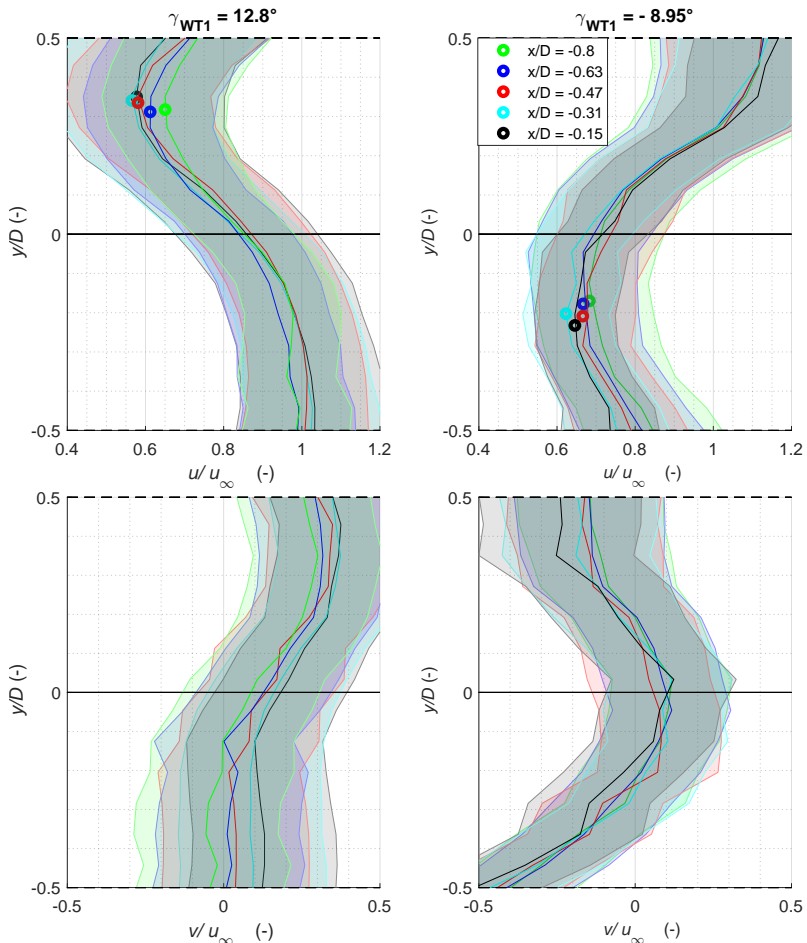

**Figure 18.** Cases 3 and 4: Wake profiles of the normalised longitudinal component (upper row) and the lateral velocity (lower row) extracted at various positions upstream of WT2 during active wake steering at WT1. The dots correspond to the wake centre at each location. The shaded area represents the bounds calculated using the SUP method with $w = 1$ m/s.

Figure 18 illustrates the horizontal wake profiles of the longitudinal and lateral velocities at five distances upstream of WT2 for the two wake deflection cases. The dots correspond to the wake centre position determined by fitting a Gaussian through the measured wake profiles (Hulsman et al., 2022b). For the positive offset $\gamma_{WT1} = 12.8°$, the wake centre deflects further to the left of the rotor to approximately $y/D = 0.32$. Similarly, the wake centre is deflected to the right to approximately $y/D = -0.2$ for negative offset $\gamma_{WT1} = -8.95°$. In both cases, the wake centre in the horizontal profiles does not exhibit significant lateral

movement evidenced by the clustered wake centre locations in Fig 18. In both cases, a yaw-induced lateral flow is observed inside the deflected wake with a magnitude depending on the yaw offset and is present at the location of the maximum velocity deficit.

Because of the partial wake inflow at WT2, the wind field across the rotor is inhomogeneous; hence, the induction across

the rotor plane varies. This is illustrated in Fig. 19, where the $u$ velocity deceleration towards WT2 is plotted for six lateral
positions for the positive and negative yaw offset case.

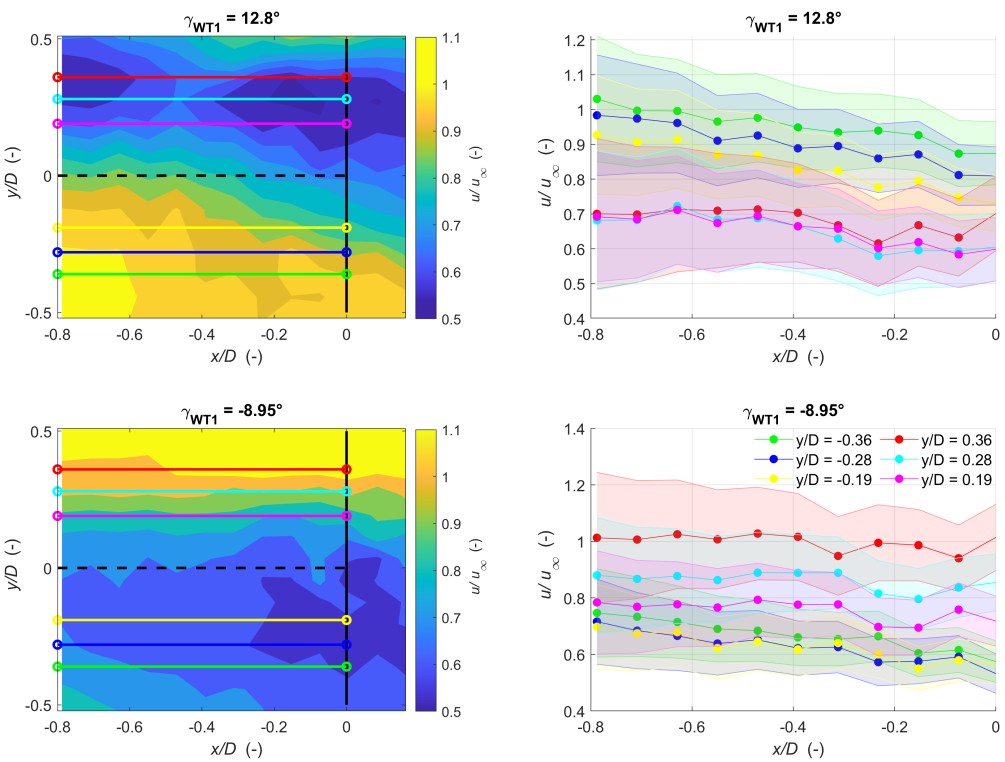

**Figure 19.** Cases 3 and 4: Variation of $u$ component velocity profiles extracted at six lateral locations for the negative (top row) and positive
offsets (bottom row).

The induction effect is the strongest close to either side of the rotor axis and reduces towards the blade tips. The induction
strength between the left and right sides of the rotor is quantified using the ratio of the longitudinal wind speed far upstream of
the scan at 0.8 $D$ and the rotor plane at 0 $D$ as $v_{\mathrm{ratio}} = \frac{v_{0.8\mathrm{D}}}{v_{0\mathrm{D}}}$. For the positive yaw case, the value of $v_{\mathrm{ratio}}$ was 1.2 and 1.14
at $y/D = -0.19$ and $y/D = 0.19$ respectively. Similarly for the negative yaw case, the value of $v_{\mathrm{ratio}}$ was 1.23 and 1.04 at
$y/D = -0.19$ and $y/D = 0.19$ respectively with similar trends observed at other spanwise positions. In both cases, the right
side of the rotor ($y/D < 0$) experiences a stronger induction than the left side ($y/D > 0$) looking downstream. For the positive
offset case, the right part of the rotor at $y/D < 0$ is exposed to the freestream and exhibits typical wind speed deceleration.
The region where the upstream wake impinges on WT2 ($y/D > 0$) shows a small recovery of wake deficit recovery up to
$x/D < -0.4$, after which the wind field is decelerated. For the negative offset case, similar effects of the induction are seen
where the wake at $y/D < 0$ decelerated faster compared to the freestream at $y/D > 0$. At $y/D < 0$, a strong deceleration

of the wake deficit is visible, with the deceleration reducing towards the rotor tips. At $y/D = 0.19$ and $y/D = 0.28$, the $u$ component deceleration is noticeable from $x/D > -0.32$ whereas, at $y/D = 0.36$, the $u$ component even slightly increases towards the rotor. The measurements indicate that the asymmetrically distributed induction zone of the downstream turbine exhibits a noticeable interaction with the deflected wake.

## 4 Discussion

We characterise the interaction of the near-wake and the induction zone between two closely spaced turbines with two synchronised scanning lidars. During the measurement campaign, yaw control is implemented on the upstream turbine. Hence, 2D characterisation of the induction zone of the downstream turbine is achieved for unwaked, waked and partial wake conditions. Measurement campaigns require a comprehensive description of the measurement setup and appropriate uncertainty quantification to interpret results. As only two lidars were available for the experiment, an assumption on the vertical flow component, e.g. $w = 0$ m/s for the dual-Doppler reconstruction is necessary to extract two-dimensional horizontal flow fields. The location of the lidar and scanning trajectory have a significant impact on the measured velocity profiles. Therefore, to quantify the effect of the measurement setup and WindScanner limitations, we simulate the measurement scenario using a high-fidelity LES and a lidar simulator. Although such simulations might not completely capture the spatio-temporal dynamics observed in the field, they provide a complementary methodology for performance and quality assessment.

A comparison of the LES and simulated lidar velocity profiles indicates that the WindScanners could capture the horizontal flow profiles despite the $w = 0$ m/s assumption and inherent measurement principle limitations, such as directional bias and probe volume averaging. Analysis of the statistical and propagated uncertainties indicated that the former was typically smaller while the latter was dominant with the error in the longitudinal component having a maximum of 5% vs 15 % of the mean wind speed. A deeper analysis into the propagated uncertainties indicated that the error in the $u, v$ component estimation was primarily because of the volume averaging effect, beam-intersection angles and beam-pointing errors while the $w = 0$ m/s assumption was the most dominant source of error.

The simulations also show that the spatial error variation depends on the local vertical velocity distribution in the scanning area, meaning that the reconstruction accuracy is not only lidar-dependent but also influenced by the measured flow field itself. This is especially true for our measurements because of the significantly high elevation angles required to scan the hub height of the downstream turbine. Because no local vertical velocity measurements were available in the scanning area, a local vertical assumption was required for the SUP. Therefore, the approach of van Dooren et al. (2016) was followed by assuming a conservative vertical velocity of 1 m/s over the scanning area. This assumption limits the direct transferability of the results of the LES analysis to the field measurements because the influence of flow dynamics on the propagated errors is masked and the magnitude of the propagated errors is much larger. Owing to this limitation, it is challenging to derive the significance of the observed flow features from field measurements, especially in waked cases. Further measurements, preferably with a third synchronised lidar, would be beneficial as they would negate the assumption of vanishing vertical velocity and provide detailed insights into the flow development in the induction zone.

This investigation presents the experimental challenges of performing full-scale experiments using synchronised lidar systems. While accurate spatial and temporal synchronisation could be achieved with careful calibration indoors and in the field, the inherent uncertainties of scanning lidar measurements need to be further evaluated. We note that the error in $v_{\text{los}}$ of 0.1 % might be low compared to our measurements. The work of Pedersen and Courtney (2021) suggested a 0.1% error in a highly controlled environment. van Dooren et al. (2022), used the same WindScanner lidars in a wind tunnel study and quantified the error against a hot wire anemometer with a mean average error metric less than 2 % in their set-up. However, the probe lengths in this study were of the order of 13 cm. For our field measurements, where probe lengths were in the order of 6.75 m to 27.75 m, an increase in the error is expected. Further measurements are required, for example, by focusing the lidars against a sonic anemometer with representative probe lengths into the lidar line-of-sight to obtain a representative $e_{\text{vlos}}$. An important aspect to consider in scanning measurements is the trade-off between spatial and temporal resolution owing to the time required to perform spatially extensive scans. With slower scan speeds, the measurements cannot capture the fluctuating behaviour of the flow but only a fingerprint of the highly turbulent near wake. Moreover, the variable probe length during the scan causes a focal distance-dependent bias and a variable low-pass filtering effect throughout the scan. Correcting for this effect is a challenging task and requires precise knowledge of the filtered and unfiltered spectra to construct a transfer function or model (Angelou et al., 2012; van Dooren et al., 2022) and was not performed in the current measurements.

As expected, the measurements indicate that the wake of the upstream turbine influences the downstream turbine induction zone. A longitudinal speed reduction towards the rotor plane is seen in the free inflow case. The lateral component shows a non-zero speed component towards the edges of the rotor, which is consistent with the flow expansion. An asymmetrical longitudinal velocity distribution at hub height was also observed due to the vertical shear altering the blade angle of attack. Furthermore, an asymmetrical longitudinal velocity deceleration at hub height was observed presumably due to a combination of the site layout and the rotor aerodynamics. The asymmetrical induction zone was attributed to vertical shear in the wind tunnel measurements of Bastankhah and Porte-Agel (2017) while Madsen et al. (2012) noted that the induction at the rotor plane could be influenced by the counter-rotating wake creating a momentum transfer between the different rotor areas. Furthermore, the presence of the long and staggered treeline between the two turbines acts as a wind break to the flow perturbing in the region between the two turbines (Counihan et al., 1974; Tobin et al., 2017). Previous studies at the site (Hulsman et al. (2022a)) have indicated the vertical profile of the flow is perturbed by the treeline through comparison of the met mast and the inflow lidar even at 100 m elevation. The treeline-induced perturbations would not only affect the flow behaviour by diverting the flow but also impact the lidar measurements, in particular, the assumption of $w = 0$ m/s necessary for the dual-Doppler reconstruction. The magnitude of these terrain effects on the flow and the lidar measurements could not be quantified in our study as no measurements were available when both the turbines were non-operational, which would have provided insights into the flow behaviour due to terrain. A high-resolution LES study with a terrain map could be used to isolate the flow behaviour due to terrain and the rotor influence. However, the LES runs in our study were intended to study the lidar measurement accuracy and therefore were initialised with a roughness length to simulate terrain effects.

By evaluating various induction zone models, we find that the induction deceleration is captured quite well with the self-similar model (Troldborg and Meyer Forsting, 2017), which is in good agreement with our data. More measurements covering more

extensive operating and stability regimes would benefit further investigation. When both turbines were operational and aligned with the prevailing wind direction, a clear overlap of the upstream turbine wake and the downstream turbine induction zone was observed because of the very close spacing creating a fully waked situation at the downstream turbine. The presence of the induction zone further increases the wake deficit, while an asymmetrical wake velocity profile is observed. While vertical plane scans would have revealed the vertical shear and veer interaction with the wake, this was not performed in our current setup. The measurements during yaw steering show partial wakes impinging on either side of the rotor. Both partial wake cases showed a stronger induction at the right side of the rotor and weaker induction on the left side looking downstream assumed to be related to the asymmetric loading on the downstream turbine. Additional measurements during various wake steering conditions would be beneficial, especially with a third synchronised lidar, are needed to determine if this trend is significant and to assess its impact on wake steering in general. While the hub height difference of 20 m between the two turbines would influence the measured induction zone and wake interaction, the effect could not be characterised as the presented study is specific to this two-turbine layout.

While a comprehensive error analysis was performed to understand and interpret the measurements, the results of this study are based only on four short datasets collected from horizontal scans performed at the hub height of the downstream turbine. Future investigations should include measurements conducted with turbines of the same hub height, vertical planes or full rotor plane measurements to study the influence of atmospheric effects such as shear and veer and multiple vertical plane measurements to study the evolution of the flow field between the two turbines. A longer measurement campaign covering a range of shear and veer conditions including negligible veer and shear could be used to strengthen the evidence of induction asymmetry and to investigate any possible correlations between atmospheric effects and the behaviour of the induction zone.

## 5  Conclusions

In this paper, we present the results of a measurement campaign using two synchronised WindScanner lidars which were used to measure the flow in the region between two 3.5 MW turbines separated by 2.7 turbine diameters ($D = 126$ m). The lidar measurements were further supported by a VAD lidar, met mast and an eddy covariance station to accurately characterise the inflow. The narrow turbine spacing and active wake steering on the upstream turbine allowed the characterisation of the induction zone flow behaviour for free inflow and full and partial wake scenarios by dual-Doppler reconstruction. We performed a careful error analysis by recreating a measurement in a Large-Eddy Simulation where two virtual WindScanner were simulated to evaluate the dual-Doppler reconstruction accuracy. In addition to the limitations of the measurement and reconstruction principle, we show that the reconstruction accuracy is influenced by the spatial variability of the measured flow. The high-quality measurements reveal flow features around large turbines, such as an asymmetrical induction zone presumably due to a combination of terrain and rotor effects and the first measurements of the lateral flow component exerted onto the near wake during intentional yaw misalignment. For a fully waked inflow, an interaction between the upstream turbine near-wake and downstream turbine induction zone was observed, whereby the induction of the downstream turbine increased the wake deficit. The measurements also revealed the interaction of the asymmetrical induction zone with the deflected wake.

The study further highlights the challenges in conducting field measurements, the additional considerations needed to characterise the induction zone behaviour and therefore how one should analyse observed flow fields. As field data is accepted as the ground truth and demanded for validating numerical models, we have strived to provide a thorough characterisation of the site, lidars, measurements and their associated uncertainties to ensure comprehensive traceability of the measurements. Further measurements, covering a larger range of inflow scenarios, preferably with a third synchronised lidar to remove the vertical velocity assumption in the dual-Doppler reconstruction, in conjunction with high-resolution modelling would be beneficial to obtain a deeper understanding of the induction zone behaviour for various operational states of the turbine.

*Data availability.*   The turbine operational data cannot be made public due to an existing non-disclosure agreement with the operator. The lidar and inflow data can be made available on request by directly contacting the authors.

*Author contributions.*   APK and PH prepared and installed the WindScanner setup of the experiment and jointly measured the data. APK focused on the measurements in a horizontal plane of the induction zone while PH mainly dealt with investigating the vertical plane(not shown in this paper). APK and PH analysed the measurement data obtained by the WindScanner, the met mast and the two turbines. APK performed the virtual WindScanner simulations and wrote the manuscript, while PH calibrated the turbine model and executed the LES runs. APK interpreted the WindScanner results with regard to the induction zone. PH analysed and prepared the VAD data for further use. MvD assisted during the design and execution of the measurement campaign, helped interpret the data and provided the uncertainty quantification methodology. MK was involved in the design of the measurement campaign, provide extensive reviews and had a supervisory role. All authors contributed to fruitful discussions and reviewed the manuscript.

*Competing interests.*   The authors declare no competing interests.

*Acknowledgements.*   This work was partly funded by the Federal Ministry for Economic Affairs and Energy according to a resolution by the German Federal Parliament in the scope of research project DFWind (Ref. Nr. 0325936C) and CompactWind II (Ref. Nr. 0325492H). We acknowledge eno energy systems GmbH for their support during the measurements. Special mention to Stephan Stone for his major support during the measurement campaign and field work to keep the campaign and the various measurement devices running. We also acknowledge Torben Mikkelsen, Mikael Sjoholm and Claus Brian Munk Pedersen from the Technical University of Denmark for sharing their expertise with the WindScanner systems and support in solving technical issues.

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
