# Peer review of "Synchronised WindScanner Field Measurements of the Induction Zone Between Two Closely Spaced Wind Turbines"

_Wind Energy Science, 2023_

## Referee Comment (RC2)

**Comments to wes-2023-114**

**General comments**

This paper presents some unique results of dual-Doppler retrieval of wind field in the induction zone of a turbine subject to different wake conditions. This work is novel and quite relevant, but some important changes, especially to the physical explanations provided to the observed phenomena, are necessary.

The error analysis is very accurate but some results of it may be misrepresented. SUP and LES do not include all the sources of error, as correctly indicated in Table 2. This should be reiterated when commenting, for instance, Fig. 9 to make sure the reader does not interpret the error bands based on SUP as an estimate for the limits of the difference of the LES validation. For instance, the error band around v is significantly larger than the difference shown by LES but simply due to the lack of pointing accuracy in LES.

Also, not including statistical uncertainty in the following plot may be misleading as this last contribution can easily dominate the overall uncertainty in real experimental campaigns. Statistical uncertainty and convergence are discussed in Section 2.3.2 but the simple comparison with the work of Simley et el. 2016 is not sufficient to justify the current results. The statistical uncertainty is a function of the specific flow conditions (mostly turbulence intensity and integral timescale) as well as number of samples. Please add at least an estimate of the error on the mean to make sure it is ok to neglect it compared to the other uncertainties.

A major drawback is also discussing differences in the measurements that are much smaller than the associated uncertainty (e.g. Fig. 16 and 17). The claimed "asymmetries" in the yaw steering cases are not significant enough to be considered a physical feature. The uncertainty quantification must indeed be used to flag significance of the results, otherwise it becomes just a theoretical exercise. Please either mention that the observed small differences between cases 3 and 4 are not relevant (so everything is really speculative) or remove at all Fig. 16 and 17 and the associated discussion.

A major issue with the interpretation of the results concerns the observer asymmetry of the induction zone. First, a mild asymmetry seems to show in the LES results, but on the other side of the rotor (Fig 6) compared to the lidar observation. Please explain why this is the case. Second and foremost, the physical explanation provided for this asymmetry, namely the "different angles of attacks" between the left and right side of the rotor induced by "wind shear" is not clear. If we consider the same height AGL, then the incoming wind speed is the same on both sides of the rotor. Being the rotational speed the same, this results in identical local inflow, relative speed and angle of attack. Furthermore, it was not found any evidence in the cited references supposedly reporting such effect. The cited papers do indeed show symmetrical induction zones even with shear, as it should be. If there's a fundamental mechanism creating this asymmetry, please provide a schematic with the angles of attacks as a function of the blade position.

The Reviewer's opinion is that major revisions addressing the fundamental points listed above are necessary before acceptance.

**Specific comments**

L 18, "…due to the extraction of kinetic energy by the rotor": this may be a subtlety but technically induction zones are present also around bodies that do not necessarily extract significant kinetic energy from the flow (e.g. in front of an airfoil). It is suggested rephrasing as "…due to the rotor thrust".

L 57: please clarify what "assumptions of the global flow field" refers to.

Table 2: A few improvements are suggested:

- The source of the dual-Doppler error is not necessarily "Non-ideal lidar placement impacts the beam-intersection angles" as an ideal placement leading to 0 error does not exists. It is indeed an "Amplification of single-Doppler uncertainty due to dual-Doppler reconstruction".
- The "averaging period error" could be renamed "statistical uncertainty".
- It would be better to remove the "unavoidable" word in the description of the error due to neglecting $w$. In fact, one could use for instance continuity or other techniques to estimate $w$. Also, please explain SUP in the caption.

L 213: Please add reference for the dual-Doppler error (e.g. Stawiarski 2013). Also, the $\Delta x$ which indicates the angle between the intersecting beams which is the driving factor for the error is not the difference of the azimuth angles $\chi_1 - \chi_2$. Please clarify this point and possibly use a symbol other that $\Delta\chi$ to describe the intersection angle as $\chi$ was already used for azimuth.

L 225-239: The main assumptions of SUP, namely the small errors and the 0 correlation between errors, should be clarified.

Table 3: A few improvements are suggested:

- Please explain all symbols in the caption.
- The stability classes indicated here do not match the three given later (stable, neutral, unstable)
- Add $z/L$ in a new column

L 254-259: Please add justification or reference for the choice the stability classes based on $L$.

L 325-327: The details of the SUP calculation should be moved at L 314 when first introducing the figures.

L 331-332: Please expand further why a high elevation leads to a high uncertainty associated with elevation. Is this just due to the structure of $\frac{\partial u}{\partial \delta}$ being proportional to $\delta$?

L 332-333: It is also not immediately clear why the error due to neglecting $w$ is larger for the lidar more aligned with the wind. Please clarify.

L 333-334: Not only P3 and P6, but also P5 has a preponderant error $\epsilon_{w,1}$ which should be explained.

L 341: The location $\frac{x}{D} = -0.16$ is not shown in Fig. 9.

L360-361: There are several ways stable stratification could impact the induction strength, please explain why it is enhancing the velocity decrease in this case.

L367-368: It is not clear why the blades would experience different relative inflow on the left and right side of the rotor. Considering hub-height for simplicity, the inflow velocity $U_{hub}$ is the same on both sides of

the rotor if vertical shear only is present. Being the rotational component $\omega R$ necessarily the same, the velocity experienced by the blades is identical. Also, the cited reference by Meyer-Forsting shows a symmetrical behavior of the induction in their LES results (below) and is therefore inadequate.

[Figure]

[Figure]

**Figure 1.** Time-averaged normal/axial velocity at the rotor disc for $\alpha = 0.5$. (LES simulation)

**Figure 2.** Time-averaged locally induction factor relative to rotor disc average ($\bar{a} = 0.270$) for $\alpha = 0.5$ at the turbine. Here $a(\mathbf{x}) = 1 - u(\mathbf{x})/V_\infty(\mathbf{x})$. (LES simulation)

The explanation based on the interaction of wake rotation and shear is sounder [Madsen et al., 2014] at least in the near wake. Please clarify this point.

L 380: Is the vertical velocity used only to compute uncertainty or also for the dual-Doppler reconstruction? Please explain.

L 388: Please clarify what the $1.96\sigma$ bounds mean. Is this the uncertainty from SUP? Why is the error bar not centered on the data?

Figure 12: A deeper analysis on why FLORIS is underpredicting so drastically the induction is needed. The fact that even at the rotor plane the estimated induction is 1/3 that of other models sounds concerning. Please also provide the Ct for this case to allow other researchers to replicate the results.

L 431: Same as comment on L 380.

L 440: It is unclear what "profiles at $\frac{y}{D} \pm 0.5$" means as those specific spanwise locations correspond to a point value in the velocity deficit, not "profiles".

L 442: The concept of "asymmetric induction" here is repeated and not explained. Again, the cited references do not mention any asymmetric induction but focus on the effects of stability and veer on the wake morphology.

L 453-456: You could cite Fleming et al, 2018 to support the fact that wake rotation and counter-rotating vortices sum up in the positive yaw case and cancel out in the negative one.

L 457: Which velocity? The lateral one?

Figures 16 and 17: please swap the plots to show first the positive yaw as in Figure 15.

L 474-475: It is hard to see the claimed stronger induction for negative y. Maybe provide a quantitative parameter like the velocity ratio between inflow and location closer to rotor plane.

L 477-478: The wake recovery followed by the deceleration is only visible for the positive yaw case.

L 496: why is the probe averaging not included as possible source of errors?

L 511: the explanation of the angle of attack as the cause of non-symmetrical induction is not sound (see above). The cited Bastankhah paper shows a symmetrical induction for 0-yaw misalignment (below):

[Figure]

FIG. 6. (a) Vectors of mean normal and tangential velocities induced by the turbine with $\gamma = 0°$ and $\lambda = \lambda_o$ in the upwind region. The background shows the contours of the normalized mean streamwise velocity $\bar{u}/\bar{u}_h$. *White lines* are iso-velocity contours, and *dashed lines* represent the side tips of the rotor. (b) Variation of the mean normalized streamwise velocity in the upwind region along the rotor axis for $\gamma = 0°$ and $\lambda = \lambda_o$.

L 524-525: the non-symmetrical induction one is hard to see (see above).

L 550: the explanation of the shear as the cause of non-symmetrical induction is not sound (see above).

**References**

[1] Madsen, H.A., Riziotis, V., Zahle, F., Hansen, M.O.L., Snel, H., Grasso, F., Larsen, T.J., Politis, E. and Rasmussen, F. (2012), Blade element momentum modeling of inflow with shear in comparison with advanced model results. Wind Energ., 15: 63-81.

[2] Fleming, Paul, et al. "A simulation study demonstrating the importance of large-scale trailing vortices in wake steering." Wind Energy Science 3.1 (2018): 243-255.

---

## Author Comment (AC1)

**Synchronised WindScanner Field Measurements of the Induction Zone Between Two Closely Spaced Wind Turbines**

**Anantha Padmanabhan Kidambi Sekar, Paul Hulsman, Marijn Floris van Dooren and Martin Kühn**

December 18, 2023
* * *
**Reviewer 1**

The authors present measurements of the induction zone of a wind turbine. The measurements were acquired by two Doppler lidars, that scanned in a synchronous mode a horizontal plane, at the hub height of a 3.5 MW wind turbine, that extended up to 0.8 D upstream. The conduction of field campaigns using multiple Doppler lidar is a challenging task. The authors provide a thorough description of the installation and the alignment of the wind lidars, as well as an investigation of the magnitude of the errors in the measuring setup. Using the acquired wind measurements, the spatial characteristics of the induction zone are examined during four periods when the inflow of the wind turbine examined was characterised by free flow, fully waked or partially waked conditions. The overall manuscript is well written and well structured. However, I think that there are some parts that should be clarified and explained better, before the article is ready for publication. I am highlighting these points in the list below.

*We thank the reviewer for their critical assessment of our work. In the following we address their concerns point by point. We hope these changes will positively benefit the manuscript. Comments to the reviewer points are made in* blue *while modifications to the manuscript are shown in* red.

Based on the comments of Reviewers 1, 2 we have summarised the major changes in the revised manuscript below:

- Expanded the site characterisation section with the topography and the presence of flow blockage (treeline) in between the two turbines and the discussion section to include the effects of the topography and treeline as a possible explanation for our measured flow features and a discussion on how to decouple the terrain effects from the rotor aerodynamic effects.

- Updated the LES results section with an analysis of the statistical uncertainty of the measurements.

- Expanded the LES Results section to include the rationale behind our assumption of the vertical velocity on the Standard Uncertainty Propagation (SUP) and its impact on the width of error bars. The discussion section has been updated to include the significance of our results based on the SUP assumptions and suggestions for future measurements with a third synchronised lidar.

- The conclusion section has been reworked substantially to include a description of our measurements, measured flow features, a summary of the uncertainty analysis and summary

of challenges in conducting field measurements for obtaining validation data for numerical models.
* * *
**Main Comments**

**Comment 1**: The authors present four data sets of induction zone measurements, which correspond to cases where the inflow of a utility-scale wind turbine is characterised by free, waked and partially waked conditions. There is a thorough description of the observed spatial features of the flow, but there is a lack of scientific conclusions. The interaction of the induction zone with the wake flow is something to be expected. My recommendation to the authors is to present what do we learn from the results of this study. In the last sentence of the Conclusions sections they write "A preliminary evaluation of the engineering models of the induction zone indicates that the models do not completely capture the complex flow behaviour and turbine interactions". I could not find a discussion of this topic in the manuscript for the case of the waked inflow. Figure 12 shows the induction zone for a free flow, but when the authors write "turbine interactions" I understand they did an evaluation of engineering models of the induction zone in waked conditions.

**Reply**:
    The objective of the work was first to use the unique fast scanning capabilities of the WindScanner systems and quantify the measurement errors due to the limitations of our experimental setup. We were able to calculate the magnitude of errors using a virtual lidar in LES approach and successfully characterised the induction zone for multiple inflow scenarios. The main conclusion that we could draw from our study is the complexity of setting up synchronised lidar measurements, difficulty in quantifying the effects that are seen in field measurements due to limitations in our measurement setup (no third lidar), the additional considerations needed to characterise the induction zone behaviour based on error analysis and therefore how one should interpret the observed flow fields. We believe this to be an important contribution as field data is usually considered as "ground truth" and is demanded as a reference for qualitative and quantitative validations of numerical flow development models. Therefore, we have strived to provide a thorough description of the site, lidars, measurements and their associated uncertainties for comprehensive traceability of our work in addition to making the lidar dataset available on request for future studies and validation attempts.
Indeed the observed spatial flow features such as induction - wake interaction and the induction zone behaviour during wake steering are expected phenomena. However, the experiments are, to the author's knowledge, the first high-resolution measurements of the induction zone - wake interaction for utility-scale turbines.
On the topic of turbine interactions, we had evaluated the coupled FLORIS induction zone model for the waked and partially waked cases but had not presented them in the paper due to the inability of the model to capture the flow field behaviour that was expected after the poor model performance in the undisturbed induction zone. We will therefore remove our statement in the conclusions and expand the discussion section that an analysis was performed but not shown in the paper.

We have updated the conclusion section based on our answer above.

**Comment 2**:  Figure 1 presents a photograph of the landscape where the measurement campaign took place. There we can see that the topography of the area between the two wind turbines is not homogeneous. There is a tree fence which is located in the vicinity of the WT1 wind turbine and extending towards the meteorological mast, and furthermore a crop field between the wind turbines WT1 and WT2. The authors don't describe in detail the physical characteristics of these features and don't discuss what if there is, or not, an impact of these features on the flow. I think that this is important, since it can partially explain the features of flow presented in Figs 6, 10, 13 and 15. For example, in Fig. 10 (c) the induction zone is seen to be symmetric right in-front of the rotor plane, but it gets asymmetric as the upwind distance increases. Why should this happen? Have the authors acquired any measurements or have they performed an LES study of the flow (where the variations of the terrain features were taken into consideration) while both wind turbines were not operating?

**Reply**:  We acknowledge that we did not discuss the site layout in detail. The induction zone of onshore wind turbines can be incorrectly estimated using observations if the effects of nonuniform terrain on the flow are not carefully considered. This non-uniformity could be a result of the changing elevation upstream of the turbine or as pointed out, through the presence of a high tree line. The impact of terrain on the induction zone have been studied through simulations [12] and experiments [9]. For instance, Mikkelsen et al., [9] in their dual-WindScanner lidar measurements at the DTU Riso site measured a vertical velocity of 1 m/s in the induction zone which was attributed to the sloping terrain upstream of the turbine.

[Figure]

Figure 1: The wind park and measurement layout at Kirch Mulsow with elevation contours. A zoomed out image of the site is shown in the top right corner illustrating the hills present upstream of the wind park. Here WT1, and WT2 refer to the upstream and downstream turbines, MM and VAD are met mast and the VAD lidar while WS1, WS2 refer to the WindScanners.

We provide a more detailed description of the site, as illustrated in Fig. 1. The elevation data was obtained with a resolution of 200m maintained by the German Ministry of Cartography and Geodesy [2]. While the elevations at the locations of the two turbines WT1, WT2 are approximately 52 m, abrupt elevation changes are seen upstream notably the presence of a small hill with an elevation of 105 m 22 D upstream of WT1 along the predominant wind direction of 228°, creating a slope of 1.09° towards the two turbines. The village of Garvensdorf with its farmhouses was approximately 1200 m upstream of WT1.

We also note the presence of tree lines and small clumps of forested terrain. As rightly pointed out, a treeline exists between WT1 and WT2 extending towards the met mast with a height of approximately 15 m-20 m estimated from pictures made during installation while other tree lines and clumps of forested area are present at various upstream positions along the 228° sector. Further analysis of the measurements from the same site was done by Hulsman et al. [8] who showed a non-negligible effect of the tall tree line by comparing the met mast and the VAD lidar data at 100 m elevation. However, the comparison is not directly transferable in our sector of interest due to the orientation of the VAD lidar and the met mast but provides enough evidence of the perturbation of the flow by the same treeline. This is consistent with literature where treelines acting as windbreaks have been shown to perturb the vertical flow profile high above the treeline [5, 14].

Therefore, terrain effects, in particular the treeline could have potentially influenced the induction zone flow. Furthermore, these could have also impacted the WindScanner measurements, in particular, the assumption of $w = 0$ m/s for the dual-Doppler reconstruction which could have influenced the measurement results shown in this paper.

However, we could not quantify the magnitude of these terrain effects on the flow and the lidar measurements. We could not acquire synchronised measurements when both the turbines were non-operational, which would have provided insights into the flow behaviour due to terrain. We acknowledge that a high-resolution LES study with a terrain map could have been used to isolate the flow behaviour due to terrain and the turbine influence. However, the LES runs in our study were intended to study the lidar measurement accuracy and therefore were initialised with a roughness length as a proxy for the terrain complexity.

We have updated the test site characterisation section, and also updated the corresponding results discussion section with the potential impact of the site orthography on the measured flow features.

**Comment 3**:   Following the comment above, in the abstract and the conclusions it is stated that the measurements presented in this article reveal a horizontal asymmetry of the induction zone which the authors claim that it is due to the vertical shear. This statement is based on the observed characteristics of one case (Case 1), with a shear exponent equal to 0.21. On which basis the authors support that this shear exponent is strong enough in order to induce a horizontal asymmetry? And how can they decouple the observed horizontal asymmetry from potential spatial variations of the horizontal flow due the heterogeneity of the terrain?

**Reply**: Ideally, we would have liked to show at least two undisturbed inflow cases, with a range of shear (or at the least strong and weak), with similar inflow and turbine operational conditions and measured with the same instrumentation. This was impossible in the limited amount of time we could perform measurements. This is the limitation of the study as the outcomes were based on small datasets which is just a small portion of the operational states of the wind turbine.

Bastankah et al, [1], in their wind tunnel experiments noted a "slight asymmetry with respect to rotor

axis" with a shear exponent of 0.17 and attributed the reason to the sheared inflow. Our argumentation followed this work and the stronger shear exponent (0.21) that was present during the measurements. However, as discussed earlier, we had discounted the effect of terrain on the flow and the measurements. Performing multiple LES runs with a flat terrain and the map of the terrain with increasing values of vertical shear can potentially decouple the effects of the terrain and the shear on the flow and identify the driving factors behind the asymmetry and provide a strong explanation for our measurements.

We have expanded the discussion of our results mentioned above to include the potential impact of the terrain characterisation on our measurements.

**Comment 4**:   I think that the error values that are used in Sect 3.1, concerning the line-of-sight (0.1 %) and the pointing accuracy (0.1 deg) are rather low. Regarding the line-of-sight the authors use as a reference the work of Pedersen and Courtney 2021 to support the choice of the 0.1 % value. First, the value reported in that study concerns a cw Doppler lidar, but not the Doppler lidars used here. And second, I guess that the line-of-sight error is dependent also on the probe length. Regarding the pointing accuracy, this will be dependent on the scanning speed. I think that the authors should address these points in the "Discussion" section.

**Reply**:  We acknowledge the points made by the referee regarding the line-of-sight and pointing errors. For continuous-wave lidars, the error will be dependent on the probe length which scales quadratically with the focus distance. The work of Pedersen and Courtney [10] suggests a 0.1% error, but as pointed out was with a different lidar system in a highly controlled environment. van Dooren et al ([16]), used the same WindScanner lidars in a wind tunnel study and quantified the error against a hot wire anemometer with a mean average error metric less than 2 %, dependant on the turbulence intensity of the inflow in their set up. The probe length in this study was in the order of 13 cm and 13.9 cm for the two WindScanners respectively. For our field measurements, where probe lengths were in the order of 6.75 m to 27.75 m, an increase in the error will be expected. One way to estimate this error would have been to set up a field measurement, focus the WindScanners with similar focus distances next to a sonic anemometer to obtain representative probe lengths, project the sonic measurements into the line-of-sight and compare the accuracy of the two systems. However, such a campaign was not executed before this study was conducted.

Indeed, the pointing accuracy will be dependent on scanning speed. We had chosen a sinusoidal motion pattern as this could be achieved by continuously rotating the two prisms in one direction only. Sinusoidal scanning therefore reduced the acceleration of the scan head reducing induced vibrations and eliminating the scan reset time. The two prisms in the WindScanner are controlled through a programmable multi-axis motion controller (PMAC), whose input is the angular position of the prisms calculated beforehand for a particular scan trajectory. Depending on the scanning trajectory, speed and the limitations of the system, it might be the case that the prism might not reach its intended angular angular position before performing a movement to the next measurement location. The WindScanner data stream captures the actual angular position along with the commanded motion positions. Figure 2 illustrates this difference $(\Delta\theta_{M1}, \Delta\theta_{M2})$ for the two steering motors for three consecutive horizontal plane scans from the presented measurement setup. For both motors, the values of $\Delta\theta_{M1}, \Delta\theta_{M2}$ did not exceed 0.1°, except for 16 points (0.12%) in each full measurement. These points were located on the very edges of the measurement plane and could not be fully eliminated. As more than 99 % of the measurement points showed an average pointing error of less than 0.1°, we report this number.

[Figure]

Figure 2: Angular difference between the commanded and actual motor positions for the two steering motors M1 and M2.

We have updated the discussion section with the explanation:
We note that the error in $v_\text{los}$ of 0.1 % might be low compared to our measurements. The work of [10] suggested a 0.1% error in a highly controlled environment. [16], used the same WindScanner lidars in a wind tunnel study and quantified the error against a hot wire anemometer with a mean average error metric less than 2 % in their set-up. However, the probe lengths in this study were of the order of 13 cm. For our field measurements, where probe lengths were in the order of 6.75 m - 27.75 m, an increase in the error is expected. Further measurements are required, for example, by focusing the lidars against a sonic anemometer with representative probe lengths into the lidar line-of-sight to obtain a representative $e_\text{vlos}$.

**Comment 5**: Section 3 presents the results of the virtual Windscanner evaluation using LES. According to the LES the largest values of the "w" component are equal to +/- 0.8 m/s (by taking into consideration the contours of Fig.6 and the inflow free wind speed at the hub height (7.7 m/s). However, for the estimation of the propagated uncertainty a constant vertical component is assumed, with a varying magnitude for each of the cases. Specifically, in Sect. 3.2.1 "w" is equal to 0.2 m/s while in Sects. 3.2.2 and 3.2.3 "w" is equal to 1.0 m/s. Can the authors comment on the selection of those values?

**Reply**: We acknowledge that the reasoning behind the choice of $w$ velocities for the propagated uncertainty calculations was not detailed in the paper. In the LES analysis, the simulated data provided an accurate value of the local $w$ component variation inside the scanning area. This local $w$ velocity was subsequently used to calculate the propagated uncertainties providing a methodology to investigate the lidar error inside the LES. However, in the free field, we did not have any measurements of the local $w$ component. Therefore, an assumption of a constant $w$ was required for the Standard Uncertainty Propagation (SUP) methodology. For the full wake and partial wake cases, a value of $w = 1$ m/s was assumed, similar to the wind tunnel experiments of van Dooren et al [15]. For the undisturbed inflow

case, the vertical velocity would be only dominated by the temperature flux between the ground and the air, and therefore a conservative value of $w = 0.2$ m/s was chosen based on the weakly stable conditions during the measurements.

The assumption of constant $w$ velocities in the measurement plane for SUP calculations will have a significant impact on our results. Due to this assumption, the errors would be overestimated at locations where the local $w$ component would be low, for instance, the most upstream part of the scan where the aerodynamic influence of the rotor on the flow would not be felt by the wind field. To illustrate this we plot the variation of $e_u$ and $e_v$ in the scanning area for the LES case with an assumption of $w = 1$ m/s in Figure 3.

[Figure]

Figure 3: Variation of $e_u$ and $e_v$ in the scanning area using the local $w$ component from the LES (first row) and the assumption of $w = 1$ m/s (second row).

The comparison of $e_u$ and $e_v$ for the two methods indicates a couple of things. Firstly, assuming constant $w$ velocities on the scanning area masks the velocity reconstruction error that is dependent on the flow dynamics, especially close to the rotor. Secondly, the magnitude of $e_u$ and $e_v$ for the constant $w$ velocity case is substantially larger than the errors estimated using the local $w$ velocities. The consequence of this is that the presented error bars in Figures 11, 14, 16 and 17 of the original manuscript are conservative and as pointed out by Reviewer 2, leads to difficulties in the interpretation and significance of the results. A possible workaround would have been to use a third synchronised lidar that would eliminate the assumption of vanishing vertical velocity, however, a third system was not available for measurements. We acknowledge this unavoidable limitation of our measurement setup and have made the following changes in the manuscript.

We have expanded the LES section with the results presented in Figure 3 for the constant $w$ velocity assumption and extended the discussion section, highlighting the assumption and its impact on the significance and interpretation of the measurements.

**Specific Comments**
* * *
**Comment 1** — Lines 10 – 12: The authors write: "The measurements revealed more evidence of horizontal asymmetry of the induction zone owing to vertical wind shear under undisturbed inflow conditions". Why do the authors write "revealed more evidence" here, since it is the first time where they mention the horizontal asymmetry of the induction zone?

**Reply**: We acknowledge that the phrasing of this sentence was not clear in the abstract. Based on the comments of both reviewers, we have rephrased this sentence to the following:
The measurements revealed the existence of a horizontal asymmetry in the induction zone possibly due to a combination of the rotor and terrain effects under undisturbed inflow.

**Comment 2** — Lines 14 – 15: The authors write: "We observed that the downstream turbine induction zone during wake steering depended on the direction of the wake steering...". Isn't this an obvious statement?

**Reply**: Yes, this is an obvious statement. At the end of the sentence, we mentioned that the "lateral movement of the deflected wake could be measured". To our knowledge, this was the first time, the lateral velocities in the deflected near wake were measured in field experiments. Therefore, we have modified this sentence to the following:
We observed the downstream turbine induction zone during wake steering while the lateral movement of the deflected wake could be measured for the first time in the free field.

**Comment 3** — Lines 105 – 106: What was the purpose of equipping the mast with a gas analyser?

**Reply**: The Irgason device is an eddy covariance system which combines both an infrared gas analyzer (IRGA) and a 3D sonic (SON) anemometer combined into a single sensor. For characterising the atmospheric stability, the Obhukov length was used. Therefore, flux measurements were carried out at the met mast at three heights, 2 m, 6 m and 60 m. Additional temperature and relative humidity probes were installed to calculate the mean air density for the moisture correction of sensible and latent heat fluxes. A more detailed description of the method for estimating stability from the Irgason is detailed in Bromm et al., [4]. We have added the following to the text:
More details on the derivation of the Obhukhov length from the Irgason are detailed in [4].

**Comment 4** — Line 110: What was the spatial resolution of the inflow lidar?

**Reply**: The pulse length of the inflow lidar was 25 m. This information has been updated in the revised manuscript.
The inflow lidar was performing VAD scans with a elevation angle of 75° and with range gates set from 50 m to 840 m with a spacing of 5 m and a pulse length of 25 m.

**Comment 5** — Line 146, Eq.5: Please describe what is denoted by "x".

**Reply**:  The term **x** denotes the spatial position vector of the measurement point in space. We have updated the text to the following:
The measured line-of-sight velocities of a cw lidar at the position $\mathbf{x} = (x, y, z)$, $v_{\text{los}}(\mathbf{x})$ can be mathematically expressed as the convolution of the wind vector $\mathbf{u(x)}$ projected along the laser beam direction and the volume averaging function:

**Comment 6** — Line 151: The effective radius is an important parameter for the operation of a cw Doppler lidar since it determines the spatial resolution of the lidar. How is the value stated here (56 mm) determined?

**Reply**:  We did not conduct any experiments to calculate the effective radius. The effective radius of the WindScanners were reported by DTU Wind Energy who developed the WindScanner systems.

**Comment 7** — Line 169: What is meant with the term "greedy controller"?

**Reply**:   The term "Greedy Controller" is used here to describe the control strategy when both the turbines were operated to extract maximum power, i.e., no wake steering was performed when this controller was active. We have rephrased this sentence to the following:
In this sector, active wake steering was performed by toggling between two unique wake steering controllers and one greedy controller where no wake steering is performed, each operational for 35 minutes.

**Comment 8** — Line 172 – 173: How were the group of horizontal planes averaged? Where the measurements grouped based on their position in a grid?

**Reply**:  We have described the post-processing of the measurement data in the beginning of Section 3, Results but have moved the text to Section 2.2.1.
For visualisation, the longitudinal and lateral velocities are interpolated using a cubic interpolation scheme onto a uniform grid with a spacing of 10 m. We rotated all measurements in the global reference frame into the main wind direction at the met mast hub height.

**Comment 9** — Line 175, Table 2: The authors present in Table 2 a list of different types of errors along with a qualitative characterisation of their impact. I think that this qualitative characterisation rather than general, it depends a lot on the measuring configuration and features of the measured flow. Therefore, I suggest that the authors should either discuss why the impact of these errors are general or mention that this characterisation concerns the specific measuring campaign. For example, the assumption of zero vertical component is probably not high over offshore areas. Furthermore, maybe this table is more part of the "Results" than of the "Methods".

**Reply**:  Agreed that the impact and magnitude of the errors from different sources are highly site-specific. However, we prefer to keep the table at the beginning as it would allow the reader to have an overview of the different errors before reading through the section. To make our table specific for our particular setup, we have modified the following text to:
For this particular measurement setup, the various lidar errors, their impact and their analysis methodology are tabulated in Table 2.

**Comment 10** — Line 186: Why is the maximum distance of the cw lidars used equal to 300 m?

**Reply**: Continuous-wave systems have an inherent maximum range (on the order of a few hundred meters), beyond which it is impossible to focus the beam, because of diffraction [6]. Furthermore, the maximum range of the cw lidars has been specified as 300 m as the measurement volume extends beyond 30 m after this range. This definition of maximum range follows the reasoning of the smaller 3″ WindScanners first developed by DTU, whose maximum range was 150 m after which the probe volume extended beyond 30 m.

**Comment 11** — Line 189: "This effect is most severe for measurements at the wake edges, . . ." Please add that this statement concerns this study.

**Reply**: We agree with this statement. The text has been amended to:
This effect concerns our study as is most severe for measurements at the wake edges, as the measurement volume extends from inside the wake to the freestream, and for measurements very close to the downstream turbine WT2, as the measurement volume would extend partially into the turbine wake.

**Comment 12** — Lines 202 – 203: What is meant with the term "effective intersection diameter"?

**Reply**: Giyanani et al [7] define effective intersection diameter for a synchronised WindScanner system as the diameter of a sphere circumscribing the location of the laser beams at the focal point. The effective diameter therefore describes the sphere within which the laser beams are expected to intersect at a particular measurement point.

**Comment 13** — Lines 213 – 218: I think that the authors should refer here to already published articles that have investigated the impact of the measuring position to measuring errors in a dual lidar measuring configuration, such as:
Peña A, Mann J. Turbulence Measurements with Dual-Doppler Scanning Lidars. Remote Sensing. 2019; 11(20):2444. https://doi.org/10.3390/rs11202444 Please note that I am neither the author nor the co-author of the above publication.

**Reply**: We have added relevant references to the paper of Pena and Mann [11] along with the work of Stawiarski et al [13] and van Dooren et al [15] to this section.

**Comment 14** — Lines 231 – 234: Eqns. 7 and 8 assume that the uncertainty terms are uncorrelated. Please add this assumption.

**Reply**: Agreed. We have added the assumption of small errors and the zero correlation between errors for SUP to the text.

**Comment 15** — Line 244. What are the unfavourable conditions that the authors refer to? Please elaborate. And what was the impact of the lower availability on the spatial distribution of the measurements? Were they certain areas with systematically lower data availability values than others?

**Reply**: The majority of the unfavourable conditions were due to periods of rain which impacted our measurements. The spatial availability distribution was dominated by the presence of the wind turbine nacelle and the spinning blades that systematically reduced data availabilities close to and behind the rotor. We have modified the text to the following:

We noticed that many measurements were also affected by unfavourable conditions such as rainfall and lower availability of aerosols to backscatter the laser beam. For operational safety reasons, the WindScanners were operated only with on-site personnel supervision. The measurements were further influenced by the presence of the wind turbine nacelle and the rotating blades that would systematically reduce data availability in the scan region.

**Comment 16** — Line 261. What was the agreement between the power law model and the measurements?

**Reply**: Figure 4 illustrates the averaged vertical wind speed distribution along with the power law fit for the full wake case. In general, a good fit is observed between the measurements and the fit with minor discrepancies noted at the lower and upper parts of the rotor due to the nature of the terrain. Also note that for case 2,3,4, the VAD lidar measurements are influenced by the induction zone of WT1 due to the small separation of 1.9 $D$.

[Figure]

Figure 4: Averaged measurements of the VAD lidar during the occurrence of the full wake case illustrated along with the power law fit.

Furthermore, no flow events such as low level jets which could have impacted our measurements were detected during the investigated measurement periods.

**Comment 17** — Line 306, Figure 6 (top row, right), why does the distribution of the "w" component is tilted in respect to the wind turbine rotor?

**Reply**: The tilted / shifted distribution of the $w$ component could be attributed to the 20 m hub height difference between the upstream and downstream turbines, similar to the field campaign operating in highly veered wind flow that is displacing the wake.

**Comment 18** — Line 310: The authors state that there is "an excellent agreement between the LES and the virtual WindScanner..." I think that this is subjective statement. The author should

explain why they think that there is a good agreement between the two. Especially when in the next sentence it is stated that there are deviations between the virtual lidars and the LES.

**Reply**: We agree that the comparison between the LES and virtual WindScanner profiles is subjective. Here, we wanted to describe that the spatial plots of the 2D velocity reconstructions between LES and WindScanner simulations are similar in a qualitative way, showing that the WindScanner could capture the dominant flow structures with the reference LES wind field such as the flow expansion at the rotor tips. Therefore, we have rephrased the text to:
A good qualitative agreement between the LES and the virtual WindScanner resolved $u, v$ profiles are noted at most parts of the scanning area. The simulations reveal that the WindScanners can capture the spatial features in the flow such as the wake rotation and flow expansion at the rotor tips.

**Comment 19** — Lines 316 – 335. The authors state that Fig 7. shows that error in "u" is at larger the WT2 rotor plane. However, from the contour plot this is not obvious. For example, the error in P2 is similar to the one at P5 and higher than the P6. Furthermore, the maximum error in the "v" component when $y/D > 0$ looks that is higher than 14%.

**Reply**: The u component error was stated to be large at the rotor plane at locations where high local $w$ velocities were present. This can be seen in the seen in Fig.06 top right where the spatial variation of the vertical velocity is shown. The error in P2 and P5 are similar (4.2 % and 4.6 %) due to the similar $w$ velocities while at P6, the u component error is 6.7 % owing to the larger local w component present here.
For comparative purposes, we presented the colour axes for both $e_u$ and $e_v$ up to 0.2. We have modified the colour limits of this plot in the revised manuscript where it is clear that the maximum value of $e_v$ at $y/D > 0$ is 15%.

**Comment 20** — Figure 8. It is very difficult from the colour of the bars to identify the contribution of each error. Can you please choose better colours?

**Reply**: We have updated the figure and assigned a separate colour for the eight error terms for easier visualisation.

**Comment 21** — Line Figure 12 presents the velocity deceleration of the longitudinal wind speed along centre of the rotor. The figure presents that the model based on the 1D Vortex sheet theory fails to reproduce the observed wind characteristics. I am wondering to which extent this is observed due to the model or due to a non-optimum selection of the free wind speed and of the induction zone factor. Have the authors tried to estimate the free wind speed and the induction zone from the Windscanner measurements?

**Reply**: The authors agree with the reviewer that an improper selection of induction factor will have a substantial influence on the modelled flow deceleration. We had indeed investigated estimating $u_\infty$ and the axial induction factor from the measurements following the work of Borracino et al [3] with the 1-D vortex sheet theory. A negligible difference was seen between the measured and modelled rotor axis velocity deceleration and hence was not reported in the paper.
The most likely reason for the discrepancy is a bug in the coupling between the induction zone models and the wake models in FLORIS. This reasoning is supported by the good agreement between the standalone induction zone models that are used in the coupling and the field measurements.

[Figure]

Figure 5: Wind direction from the met mast and WT1 heading on 25.02.2021. The period from 09:45AM to 10:15AM UTC was when the 12.8° yaw offset case was recorded.

**Comment 22** — Line 450. Figure 15 presents the longitudinal component of the wind measured for two different yaw misalignment angles. I can understand the spatial distribution of the "u" component, where the trace of the wake propagates mainly at the region where $y/D < 0$. One the other hand when the yaw misalignment of the WT1 is equal to 12.8 degrees then wake tract is in the region where $y/D > 0$. However the direction of the propagation of the wake seems strange. I would expect that the wake should move upwards, but from the measurements it looks that it is moving downwards. Can the authors comment on this?

**Reply**: For the 12.8° yaw offset case, the spatial distribution of the longitudinal component seems to indicate that the wake was deflected towards $y/D \geq 0$, but closer to the rotor deflects downwards. This could be possibly because of how the measurements were chosen for analysis. The WindScanner was performing measurements when the wind direction was aligned with the two turbines. The mean wind direction for the 12.8° case was 217° already creating an offset as both the turbines are not completely aligned, as in the other three cases. Furthermore, the standard deviation of wind direction at approximately 10°, was the highest amongst all the investigated cases. We show the wind direction at the metmast and the WT1 turbine heading in Figure 5. Here the toggler value of 1 indicates when the measurements were recorded for analysis. During the time of measurements, the wind direction ranged from 208° to 223°. This comparatively large variability in wind direction during measurements could be the most possible explanation influencing the spatial wind field distribution over the individual scans, and therefore the averaged results. We have added the following text to the modified manuscript:

It is noted that for the positive offset case, the spatial distribution of the $u$ component seems to move near the rotor axis instead of deflecting towards $y/D > 0$. This could be potentially attributed to the 10° misalignment between the wind direction and the turbine orientation direction in addition to the large variability of the wind direction from 208° to 223° which was the highest of all investigated cases.

**Minor Comments**

**Comment 1** — Line 5. Please change the "The measurements were conducted with..." with

"The measurements were acquired by..."

**Reply**: We have changed this in the revised manuscript.

**Comment 2** — Lines 96 – 97: Please add in this sentence that WT1 is the upstream wind turbine and WT2 is the downstream.

**Reply**: We have changed this in the revised manuscript.
The upstream and downstream turbines are abbreviated as WT1 and WT2 respectively..

**Comment 3** — Line 103: Please replace the verb "outfit" with the verb "equip"

**Reply**: We have changed this in the revised manuscript.

**Comment 4** — Lines 104-105: Please add the names of the Theis products.

**Reply**: We have added the following to the text:
Inflow conditions were measured by a met mast placed 2.6 $D$ north of WT1, equipped with two anemometers,Thies First Class Wind Transmitter anemometer of type 4.3352.00.400 from at the lower tip of 54 m and close to the WT1 hub height of 116m. A wind vane of type Thies First Class Wind Direction Transmitter of type 4.3151.00.212 is also installed at 112 m.

**Comment 5** — Line 140: What is meant with the "Without generalisation ..."? Please clarify.

**Reply**: We have used the terminology without generalisation to indicate that our assumption of a vanishing vertical component might not be the only way to reconstruct two-dimensional wind fields from dual lidar measurements.

**Comment 6** — Line 167: Remove the dot after "Table"

**Reply**: Removed.

**Comment 7** — Line 168: Table 1. Label: Is it Fig. 2(b) or Fig. 2(a)?

**Reply**: The reference is for Fig. 2(a). We have made the change in the revised manuscript.

**Comment 8** — Line 183: Figure 4. Replace "The doted lines..." with "The dashed lines..."

**Reply**: Replaced.

**Comment 9** — Line 187: Please delete the second "averaging"

**Reply**: The repetition has been deleted.

**Comment 10** — Line 252, Table 3. Please add a description of each column in the label of the table.

**Reply**: We have added the following to the caption:
Summary of the measurement cases. Each case is characterised by its freestream wind speed $u_\infty$, turbulence intensity (TI), mean wind direction ($\theta_{\mathrm{wdir}}$), stability parameter ($z/L$), stability, wind veer ($\gamma$), vertical wind shear ($\alpha_{\mathrm{shear}}$) and the yaw offset of the turbines ($\gamma_{\mathrm{WT}}$).

**Comment 11** — Line 317. Replace Figure 7 with Fig. 7

**Reply**: Replaced.

**Comment 12** — Line 325. Replace Figure 7 with Fig. 7

**Reply**: Replaced.

**References**

[1] M. Bastankhah and F. Porte-Agel. Wind tunnel study of the wind turbine interaction with a boundary-layer flow: Upwind region, turbine performance, and wake region. *Physics of Fluids*, 29(6), 2017.

[2] BKG. Digitales Geländemodell Gitterweite 200 m, 2013.

[3] A. Borraccino, D. Schlipf, F. Haizmann, and R. Wagner. Wind field reconstruction from nacelle-mounted lidar short-range measurements. *Wind Energy Science*, 2(1):269–283, 5 2017.

[4] M. Bromm, A. Rott, H. Beck, L. Vollmer, G. Steinfeld, and M. Kühn. Field investigation on the influence of yaw misalignment on the propagation of wind turbine wakes. *Wind Energy*, 21(11):1011–1028, 11 2018.

[5] J. Counihan, J. C. Hunt, and P. S. Jackson. Wakes behind two-dimensional surface obstacles in turbulent boundary layers. *Journal of Fluid Mechanics*, 64(3):529–564, 1974.

[6] R. Frehlich. Effects of wind turbulence on coherent Doppler lidar performance. *Journal of Atmospheric and Oceanic Technology*, 14(1):54–75, 1997.

[7] A. Giyanani, M. Sjöholm, G. Rolighed Thorsen, J. Schuhmacher, and J. Gottschall. Wind speed reconstruction from three synchronized short-range WindScanner lidars in a large wind turbine inflow field campaign and the associated uncertainties. *Journal of Physics: Conference Series*, 2265(2):022032, 5 2022.

[8] P. Hulsman, C. Sucameli, V. Petrović, A. Rott, A. Gerds, and M. Kühn. Turbine power loss during yaw-misaligned free field tests at different atmospheric conditions. *Journal of Physics: Conference Series*, 2265(3):032074, 5 2022.

[9] T. Mikkelsen, M. Sjöholm, P. Astrup, A. Peña, G. Larsen, M. F. van Dooren, and A. P. Kidambi Sekar. Lidar Scanning of Induction Zone Wind Fields over Sloping Terrain. *Journal of Physics: Conference Series*, 1452(1):012081, 1 2020.

[10] A. T. Pedersen and M. Courtney. Flywheel calibration of a continuous-wave coherent Doppler wind lidar. *Atmospheric Measurement Techniques*, 14(2):889–903, 2 2021.

[11] A. Peña and J. Mann. Turbulence Measurements with Dual-Doppler Scanning Lidars. *Remote Sensing*, 11(20):2444, 10 2019.

[12] M. Sanchez Gomez, J. K. Lundquist, J. D. Mirocha, R. S. Arthur, D. Muñoz-Esparza, and R. Robey. Can lidars assess wind plant blockage in simple terrain? A WRF-LES study. *Journal of Renewable and Sustainable Energy*, 14(6):63303, 11 2022.

[13] C. Stawiarski, K. Traumner, C. Knigge, and R. Calhoun. Scopes and challenges of dual-doppler lidar wind measurements-an error analysis. *Journal of Atmospheric and Oceanic Technology*, 30(9):2044–2062, 2013.

[14] N. Tobin, A. M. Hamed, and L. P. Chamorro. Fractional Flow Speed-Up from Porous Windbreaks for Enhanced Wind-Turbine Power. *Boundary-Layer Meteorology*, 163(2):253–271, 5 2017.

[15] M. F. van Dooren, F. Campagnolo, M. Sjöholm, N. Angelou, T. Mikkelsen, and M. Kühn. Demonstration and uncertainty analysis of synchronised scanning lidar measurements of 2-D velocity fields in a boundary-layer wind tunnel. *Wind Energy Science*, 2(1):329–341, 6 2017.

[16] M. F. van Dooren, A. P. Kidambi Sekar, L. Neuhaus, T. Mikkelsen, M. Hölling, and M. Kühn. Modelling the spectral shape of continuous-wave lidar measurements in a turbulent wind tunnel. *Atmospheric Measurement Techniques*, 15(5):1355–1372, 3 2022.

---

## Author Comment (AC2)

**Synchronised WindScanner Field Measurements of the Induction Zone Between Two Closely Spaced Wind Turbines**

**Anantha Padmanabhan Kidambi Sekar, Paul Hulsman, Marijn Floris van Dooren and Martin Kühn**

December 18, 2023

**Reviewer 2**

This paper presents some unique results of dual-Doppler retrieval of wind field in the induction zone of a turbine subject to different wake conditions. This work is novel and quite relevant, but some important changes, especially to the physical explanations provided to the observed phenomena, are necessary.

*We thank the reviewer for their critical assessment of our work. In the following, we address the concerns point by point. We hope these changes will positively benefit the manuscript. Comments to the reviewer points are made in* blue *while modifications to the manuscript are made in* red.

Based on the comments of Reviewers 1, 2 we have summarised the major changes in the revised manuscript below:

- Expanded the site characterisation section with the topography and the presence of flow blockage (treeline) in between the two turbines and the discussion section to include the effects of the topography and treeline as a possible explanation for our measured flow features and a discussion on how to decouple the terrain effects from the rotor aerodynamic effects.

- Updated the LES results section with an analysis of the statistical uncertainty of the measurements.

- Expanded the LES Results section to include the rationale behind our assumption of the vertical velocity on the Standard Uncertainty Propagation (SUP) and its impact on the width of error bars. The discussion section has been updated to include the significance of our results based on the SUP assumptions and suggestions for future measurements with a third synchronised lidar.

- The conclusion section has been reworked substantially to include a description of our measurements, measured flow features, a summary of the uncertainty analysis and summary of challenges in conducting field measurements for obtaining validation data for numerical models.

**General Comments**

**Comment 1**: The error analysis is very accurate but some results of it may be misrepresented. SUP and LES do not include all the sources of error, as correctly indicated in Table 2. This should

be reiterated when commenting, for instance, Fig. 9 to make sure the reader does not interpret the error bands based on SUP as an estimate for the limits of the difference of the LES validation. For instance, the error band around v is significantly larger than the difference shown by LES but simply due to the lack of pointing accuracy in LES.

**Reply**: Indeed, SUP and LES does not contain all sources of errors, which is why we had used both methods for our analysis. In the revised manuscript, we have clearly specified the method through with the error bars were obtained and expanded the discussion accordingly.

We have added the similar text to the revised manuscript in Section 3.1.2 to the measurement cases. The error bars around the $v$ component profiles are larger than the differences in the LES and WindScanner resolved profiles due to the inclusion of multiple error terms in the SUP.

**Comment 2**: Also, not including statistical uncertainty in the following plot may be misleading as this last contribution can easily dominate the overall uncertainty in real experimental campaigns. Statistical uncertainty and convergence are discussed in Section 2.3.2 but the simple comparison with the work of Simley et el. 2016 is not sufficient to justify the current results. The statistical uncertainty is a function of the specific flow conditions (mostly turbulence intensity and integral timescale) as well as number of samples. Please add at least an estimate of the error on the mean to make sure it is ok to neglect it compared to the other uncertainties.

**Reply**: We agree with the reviewer that the statistical uncertainty needs to be presented. While the total propagated uncertainty regards the accuracy of single input variables, the statistical uncertainty quantifies the precision of the results from different scans. A higher number of scans typically reduces measurement noise from the statistical error. To quantify the statistical uncertainty, we use the margin of error estimated in the scanning area, for the LES case with two aligned and operational turbines. The margin of error was calculated as $e_{\mathrm{u,stat}} = \frac{z_\gamma \sigma_{\mathrm{u}}}{\sqrt{N_{\mathrm{s}}}}$ and $e_{\mathrm{v,stat}} = \frac{z_\gamma \sigma_{\mathrm{v}}}{\sqrt{N_{\mathrm{s}}}}$. Here $z_\gamma$, the confidence level, is set to 1.96, denoting the 95 % confidence interval, $\sigma_{\mathrm{u}}, \sigma_{\mathrm{v}}$ are the standard deviations of the longitudinal and lateral velocity components in the scan plane obtained from the WindScanner simulations and $N$ is the number of samples.

[Figure]

Figure 1: Statistical Uncertainty estimated for the $u, v$ components.

Figure 1 shows the variation of the margin of error in the scanning area for the two reconstructed components normalised with the mean longitudinal wind speed. The margin of error for the longitudinal component varies in the scan area between 2 % to 5 % depending on the turbulence intensity in the wake. Similarly, for the $v$- component, the margin of error varies between 1.3 % and 2.5 %. The higher errors at scan edges could be attributed to the low amount of data points in these locations as a consequence of the scanning patterns. This indicates that our set-up can capture the averaged velocities in the scan plane with a statistical uncertainty that is lower than the dual-Doppler propagated uncertainties. In the field data, we can expect that the margin of error would be slightly higher than in the idealised LES due to the filtering procedure reducing data availability in each scan.

We have added the presented analysis of the statistical uncertainty in Section 3.1 of the revised manuscript.

**Comment 3**: A major drawback is also discussing differences in the measurements that are much smaller than the associated uncertainty (e.g. Fig. 16 and 17). The claimed "asymmetries" in the yaw steering cases are not significant enough to be considered a physical feature. The uncertainty quantification must indeed be used to flag significance of the results, otherwise it becomes just a theoretical exercise. Please either mention that the observed small differences between cases 3 and 4 are not relevant (so everything is really speculative) or remove at all Fig. 16 and 17 and the associated discussion.

**Reply**: We acknowledge that we have not discussed the significance of the results in detail. We first discuss the methodology by which the error bars were drawn in Fig 16, and 17 before suggesting a way linking the uncertainty quantification exercise and the measurements.

The error bars in Figs. 14, 16, and 17 were estimated using a $w$ component value of 1 m/s. We acknowledge that the reasoning behind the choice of $w$ velocities for the SUP method was not detailed in the paper. In the LES analysis, the reference wind fields provided an accurate value of the local $w$ component variation inside the scanning area. This local $w$ velocity was subsequently used to calculate the propagated uncertainties providing a methodology to investigate the lidar error inside the LES. However, in the free field, we did not have any measurements of the local $w$ component. Therefore, an assumption of a constant $w$ was required for the Standard Uncertainty Propagation (SUP) methodology. For the full wake and partial wake cases, a value of $w = 1$ m/s was assumed, similar to the wind tunnel experiments of van Dooren et al [15]. For the undisturbed inflow case, the vertical velocity would be only dominated by the temperature flux between the ground and the air, and therefore a conservative value of $w = 0.2$ m/s was chosen based on the weakly stable conditions during the measurements.

The assumption of constant $w$ velocities in the measurement plane for SUP calculations will have a significant impact on our results. Therefore, the errors would be overestimated at locations where the local $w$ component would be low, for instance, the most upstream part of the scan where the aerodynamic influence of the rotor on the flow would not be felt by the wind field. To illustrate this we plot the variation of $e_u$ and $e_v$ in the scanning area for the LES case with an assumption of $w = 1$ m/s in Figure 2.

The comparison of $e_u$ and $e_v$ for the two methods indicates two things. Firstly, assuming constant $w$ on the scanning area masks the velocity reconstruction error that is dependent on the flow dynamics, especially close to the rotor. Secondly, the magnitude of $e_u$ and $e_v$ for the constant $w$ velocity case is substantially larger than the errors estimated using the local $w$ velocities. The consequence of this is that the presented error bars in Figures 11, 14, 16 and 17 of the original manuscript are conservative, leading to difficulties in the interpretation and significance of the results. Had the $w$ component been low, the

[Figure]

Figure 2: Variation of $e_u$ and $e_v$ in the scanning area using the assumption of $w = 1$ m/s (bottom row) and the local $w$ component from the LES (top row).

error bars could have been smaller and we could have made definitive statements on the significance of the results. In the revised version of the paper, we have shown and discussed the spatial variation of the propagated error using the actual and assumed vertical velocities to bridge the LES and the field measurements. We have extended the discussion section, highlighting the assumption and its impact on the significance and interpretation of the measurements. A possible workaround would have been to use a third synchronised lidar that would eliminate the assumption of vanishing vertical velocity, however, a third system was not available for measurements. We acknowledge this unavoidable limitation of our measurement setup and have made the following changes in the manuscript.

We have made the following changes in the revised manuscript.

- Expanded the LES Section 3.1 with the SUP analysis based on the assumptions of $w$ velocity instead of the true LES velocities to link the simulations and experiments.

- Extended the discussion section, highlighting the assumption and its impact on the significance and interpretation of the measurements.

**Comment 4**: A major issue with the interpretation of the results concerns the observer asymmetry of the induction zone. First, a mild asymmetry seems to show in the LES results, but on the other side of the rotor (Fig 6) compared to the lidar observation. Please explain why this is

the case. Second and foremost, the physical explanation provided for this asymmetry, namely the "different angles of attacks" between the left and right side of the rotor induced by "wind shear" is not clear. If we consider the same height AGL, then the incoming wind speed is the same on both sides of the rotor. Being the rotational speed the same, this results in identical local inflow, relative speed and angle of attack. Furthermore, it was not found any evidence in the cited references supposedly reporting such effect. The cited papers do indeed show symmetrical induction zones even with shear, as it should be. If there's a fundamental mechanism creating this asymmetry, please provide a schematic with the angles of attacks as a function of the blade position.

**Reply**:

In our attempt to explain the measurements, we first will discuss the quality of the collected data and then proceed with an explanation for the measurement results. We are confident in the ability to achieve the scanning trajectories with high accuracy through extensive wind tunnel and field testing. Therefore, we could eliminate any mechanical/optical issues with the lidars that would contaminate measurements.

We referred to the work of Bastankah et al [2] who conducted wind tunnel measurements of a model wind turbine under sheared conditions ($\alpha = 0.17$). They noted "a slight lateral asymmetry with respect to the rotor axis" visualised through iso-velocity contours at far upstream positions. Similar to our measurements, the asymmetry also disappears closer to the rotor plane. Bastankah et al provided a possible explanation of the slight asymmetry due to rotor blades experiencing different angles of attack as they move through the sheared flow. However, as pointed out, indeed the induced velocities for a rotor with a constant rpm will be the same on either side of the rotor and will only change the induction in the vertical direction. This is the effect that we wanted to describe with the work of Frosting et al [9] who show the variation of induction in the rotor plane for strongly sheared flow. We also described the mechanism of momentum transfer between the lower and upper parts of the rotor as an additional explanation for the asymmetry. The authors thank the reviewer for pointing us towards the work of Madsen et al [8] who provided similar reasoning of the wake rotation and its impact on the induced velocities at the rotor plane.

However, as noted by Reviewer 1, we had not fully characterised the test site in terms of terrain which could provide additional explanation for the measured asymmetry. The induction zone behaviour can be mischaracterised with observations if the effects of nonuniform terrain on the flow are not carefully considered. This non-uniformity could be a result of the changing terrain upstream of the turbine or the presence of a forest or a tree line. For instance, Mikkelsen et al., [10] in their dual-WindScanner lidar measurements at the DTU Riso site measured a vertical velocity of 1 m/s in the induction zone which was attributed to the upward sloping terrain upstream of the turbine.

We first discuss the layout of the site more in detail, as illustrated in Fig. 3, with a spatial resolution of 200 m obtained from [3]. While the elevations at WT1 and WT2 are similar, abrupt changes in elevation are seen upstream notably the presence of a small hill with an elevation of 105 m 22 D upstream of WT1 along the predominant wind direction, creating a slope of 1.09° towards the two turbines. We also note the presence of a village, tree lines and small clumps of forested terrain along the predominant wind direction. Furthermore, a treeline exists between WT1 and WT2 and therefore in the induction region of WT2. The treeline extended towards the met mast with a height of approximately 15 m-20 m estimated from pictures taken during the installation campaign. Further analysis of the measurements from the same site was done by Hulsman et al. [7] who showed a clear effect of the same tree line by comparing the met mast and the VAD lidar data at 100 m AGL. However, the comparison is not directly transferable in our sector of interest due to the orientation of the VAD lidar and the met mast

[Figure]

Figure 3: The wind park and measurement layout at Kirch Mulsow overlayed with elevation contours. A zoomed-out image of the site is shown in the top right corner illustrating the hills present upstream of the wind park. Here WT1, and WT2 refer to the upstream and downstream turbines, MM and VAD refer to the met mast and the inflow lidar while WS1 and WS2 refer to the two WindScanners.

but provides enough evidence of the perturbation of the flow by the treeline. This is consistent with literature where trees acting as windbreaks have been shown to perturb the vertical flow profile high above the treeline [6, 14].

We believe that the effect of terrain and in particular treeline could have influenced the flow behaviour in the induction zone, which we did not address previously. Furthermore, these effects could have also impacted the WindScanner measurements, in particular, the assumption of vanishing vertical wind speed for the dual-Doppler reconstruction which could have influenced the measurement results.

In our present setup, we could not quantify the magnitude of these terrain effects on the flow and the lidar measurements. We could not acquire measurements when both the turbines were non-operational, which would have provided insights into the flow behaviour due to terrain and the tree line. Using a high-resolution LES with a terrain map could have been used to isolate the terrain influences. However, the LES runs in our study were intended to study the lidar measurement accuracy and therefore were initialised with a roughness length as a proxy for the terrain complexity and hence not fully representative of the site. This could could explain why the asymmetry exists on the other side of the rotor compared to field measurements.

We hope that these changes to the manuscript will provide a suitable explanation for the measurements.

We have updated the test site description with the terrain map and described the treeline in between WT1 and WT2 and added the terrain and treeline effects as a possible additional effect for the flow perturbation in all the sections. We have further updated the discussion section with the effect of the terrain on the flow and the dual-Doppler reconstruction and discussed the limitation of our simulation set-up where the terrain and hence the lidar error due to terrain could not be modelled.

**Specific Comments**

**Comment 1** — L 18, "...due to the extraction of kinetic energy by the rotor": this may be a subtlety but technically induction zones are present also around bodies that do not necessarily extract significant kinetic energy from the flow (e.g. in front of an airfoil). It is suggested rephrasing as "...due to the rotor thrust".

**Reply**: The sentence has been rephrased as suggested.

**Comment 2** — L 57: please clarify what "assumptions of the global flow field" refers to.

**Reply**: The assumption of global flow field refers to the assumption of a constant vertical wind component required to perform the dual-Doppler reconstruction. We have replaced the phrasing in the paper. However, owing to the lidar measurement principle and scanning limitations, such as the volume averaging effect, assumptions of vertical velocity for dual-Doppler reconstruction, scanning speeds, beam pointing, and intersection accuracies, a thorough error and uncertainty assessment is required before interpreting the measurements

**Comment 3** — Table 2: A few improvements are suggested:

- The source of the dual-Doppler error is not necessarily "Non-ideal lidar placement impacts the beam-intersection angles" as an ideal placement leading to 0 error does not exists. It is indeed an "Amplification of single-Doppler uncertainty due to dual-Doppler reconstruction". We have changed the text in the revised manuscript.

- The "averaging period error" could be renamed "statistical uncertainty". We have changed the text in the revised manuscript.

- It would be better to remove the "unavoidable" word in the description of the error due to neglecting $w$. In fact, one could use for instance continuity or other techniques to estimate $w$. Also, please explain SUP in the caption. We have changed the text in the revised manuscript.

**Reply**: We have changed the descriptions of the errors to reflect the reviewer comments and defined the abbreviations of LES and SUP on the caption.

**Comment 4** — L 213: Please add reference for the dual-Doppler error (e.g. Stawiarski 2013). Also, the $\Delta\chi$ which indicates the angle between the intersecting beams which is the driving factor for the error is not the difference of the azimuth angles $\chi_1 - \chi_2$. Please clarify this point and possibly use a symbol other that $\Delta\chi$ to describe the intersection angle as $\chi$ was already used for azimuth.

**Reply**: We have added references to Stawiarski et al[13], van Dooren et al [16] and Pena and Mann [12] as appropriate references here. Indeed, we refer here to the intersection angle of the beams which is scales the dual-Doppler reconstruction error as described in Stawiarski et al [13]. We have represented the term as $R_{\text{int}}$ in the revised manuscript.

**Comment 5** — Table 3: A few improvements are suggested:

- Please explain all symbols in the caption.

- The stability classes indicated here do not match the three given later (stable, neutral, unstable)

- Add $z/L$ in a new column

**Reply**:

- We have expanded the caption to the following:
  Summary of the measurement cases. Each case is characterised by its freestream wind speed $u_\infty$, turbulence intensity (TI), mean wind direction ($\theta_{\mathrm{wdir}}$), stability parameter ($z/L$), stability, wind veer ($\gamma$), vertical wind shear ($\alpha_{\mathrm{shear}}$) and the yaw offset of the turbines ($\gamma_{\mathrm{WT}}$).

- The table has been expanded to contain $z/L$ values and the stability classes has been checked and made consistent with the results section.

**Comment 6** — L 254-259: Please add justification or reference for the choice the stability classes based on $L$.

**Reply**: For justification of the choice of $L$ for stability classification, we have added the following references to the text: .
The atmospheric stability of the boundary layer can be characterised by the Monin-Obhukov similarity theory [11, 1]. The stability parameter $z/L$ was measured by the eddy covariance station at a height of 6 m above the ground following Bromm et al,[5].

**Comment 7** — L 325-327: The details of the SUP calculation should be moved at L 314 when first introducing the figures.

**Reply**: We have moved the details of the SUP calculation before the figures are introduced.

**Comment 8** — L 331-332: Please expand further why a high elevation leads to a high uncertainty associated with elevation. Is this just due to the structure of $\frac{\partial u}{\partial \delta}$ being proportional to $\delta$?

**Reply**: The general form of the uncertainty propagation involves the product of the partial derivative and the uncertainty of the input variable, in this case, line-of-sights, beam pointing angles and vertical wind component. For the case of the elevation angle, the partial derivative $\frac{\partial u}{\partial \delta}$ was positively correlated to the elevation angle $\delta$. Therefore, an increase in $\delta$ would result in a more substantial impact on the overall uncertainty in the wind component $e_{\mathrm{u}}$. We have modified the text to the following:

At P1, P2 and P3, $e_{\delta_{\mathrm{i}}}$ is the largest contributor due to the severe elevation angles required to scan at these points and the positive correlation between $\frac{\partial u}{\partial \delta}$ and $\delta$.

**Comment 9** — L 332-333: It is also not immediately clear why the error due to neglecting $w$ is larger for the lidar more aligned with the wind. Please clarify.

**Reply**: We can explain this from the contribution of the vertical velocity to the individual line-of-sight of the two lidars. For the un-aligned lidar, the lidar measures with an angle to the measurement point

and therefore the line-of-sight component will contain contributions from both the lateral and vertical velocity components. Neglecting the vertical velocity might have a smaller impact on this lidar since it already incorporates both components in its measurements. For the aligned lidar, the measurements are more sensitive to the changes in the $w$ component. For a non-zero $w$ component, the aligned lidar will contain a larger contribution of the $w$ component projected onto its line-of-sight compared to the un-aligned case. We have modified the text to the following:

> The varying contributions of $e_{\mathrm{w,i}}$ at the points of interest can be explained by the relative alignment of the lidar with the wind direction. For a non-zero $w$ component, an aligned lidar will contain a larger contribution of the $w$ component projected onto its line-of-sight compared to the un-aligned case.

**Comment 10** — L 333-334: Not only P3 and P6, but also P5 has a preponderant error $e_{\mathrm{w,1}}$ which should be explained.

**Reply**: The error at P5 follows the same argumentation as Comment 9 and is dominated by the term $e_{\mathrm{w,i}}$. The value of $e_{\mathrm{w,1}}$ is larger than P3 and P6 due to the large local $w$ velocity close to the rotor obtained from the LES (Fig. 6 (top right)). We have modified the text to the following:

> Similarly, at P3, P5 and P6, WS1 is approximately aligned with the longitudinal wind speed component. So the errors at these points are dominated by the $e_{\mathrm{w,1}}$, which is highest at P5 due to the large local $w$ velocity in the LES field (Fig. 06 (top right)).

**Comment 11** — L 341: The location $x/D = 0.16$ is not shown in Fig 9.

**Reply**: Thank you for pointing this out. We have corrected the location in the text to $x/D = -0.08$ as depicted in Fig. 9.

**Comment 12** — L360-361: There are several ways stable stratification could impact the induction strength, please explain why it is enhancing the velocity decrease in this case.

**Reply**: Stable stratification inhibits wind vertical turbulent mixing. In the induction zone, where the rotor blades are influencing the flow, stable stratification can limit the upward transport of momentum. This reduced mixing can lead to an enhancement of the induction zone and the velocity decrease. We have modified the text to the following:

> This strong velocity deficit can be attributed to high axial induction and weakly stable stratification during the measurement period inhibiting vertical turbulent mixing.

**Comment 13** — L367-368: It is not clear why the blades would experience different relative inflow on the left and right side of the rotor. Considering hub-height for simplicity, the inflow velocity $u_{hub}$ is the same on both sides of the rotor if vertical shear only is present. Being the rotational component $\omega R$ necessarily the same, the velocity experienced by the blades is identical. Also, the cited reference by Meyer-Forsting shows a symmetrical behavior of the induction in their LES results (below) and is therefore inadequate. The explanation based on the interaction of wake rotation and shear is sounder [Madsen et al., 2014] at least in the near wake. Please clarify this point.

**Reply**: We have addressed this point in General Comment 4. Indeed the induced velocities for a rotor with a constant rpm will be the same on either side of the rotor and will only change the induction in the vertical direction. This is the effect that we wanted to describe with the work of Frosting et al [9] who show the variation of induction in the rotor plane for strongly sheared flow. We have added a further explanation of the asymmetry due to terrain effects and the presence and influence of a treeline in between WT1 and WT2.

Looking downwind, this slight asymmetry could be attributed to the presence of a tall treeline in-between WT1 and WT2 perturbing the flow and the strong vertical shear $\alpha_{\mathrm{shear}} = 0.21$ that causes a vertical wind speed gradient varying the relative wind speed and the angle of attack of the blades during a rotation. Additionally, the induced velocities at the rotor plane are influenced by the counter-rotating wake creating a momentum transfer between the lower and upper rotor regions leading to a difference in flow magnitude between $y/D > 0$ and $y/D < 0$ [8]. Hence, the blade sections would experience varying blade forces that vary the local thrust coefficient, and hence, the induction factor and corresponding deceleration.

**Comment 14** — L 380: Is the vertical velocity used only to compute uncertainty or also for the dual-Doppler reconstruction? Please explain.

**Reply**: The vertical velocity assumption of 0.2 m/s was chosen only for computing the uncertainty while the dual-Doppler reconstruction was done by assuming $w = 0$ m/s. Similarly, for the uncertainty assessment for the partial and full wake cases, a constant vertical velocity of 1 m/s was assumed following the work of van Dooren et al [15]. The implications of this assumption on the propagated uncertainty has been discussed in Major Comment 3. We have modified the text to the following:

For calculating the propagated uncertainties, a constant vertical component $w = 0.2$ m/s is assumed, as no wakes propagating from the non-operational upstream turbine and no direct measurements of the $w$ component were available in the scanned area.

**Comment 15** — L 388: Please clarify what the 1-96$\sigma$ bounds mean. Is this the uncertainty from SUP? Why is the error bar not centered on the data?

**Reply**: We clarify that the error bars presented represent the propagated uncertainties from SUP and not the statistical uncertainty. The revised version of the manuscript has been updated with a plot with centred error bars on the WindScanner data.

**Comment 16** — Figure 12: A deeper analysis on why FLORIS is underpredicting so drastically the induction is needed. The fact that even at the rotor plane the estimated induction is 1/3 that of other models sounds concerning. Please also provide the Ct for this case to allow other researchers to replicate the results.

**Reply**: We acknowledge that the systematic underprediction of the velocity deceleration's from the FLORIS model is a cause of concern and hence was not used for estimating the velocity decelerations in the partial and full wake cases. We had used the FLORIS+Induction coupling described in Branland et al [4] that downloaded from the Github repository. We also contacted the authors of the paper [4] to describe and find a solution to the problem. The most likely reason for the discrepancy is a bug in the coupling between the induction zone models and the wake models in FLORIS. This reasoning is

supported by the good agreement between the standalone induction zone models that are used in the coupling and the field measurements.
Due to an NDA with the turbine operator, we cannot disclose the turbine Ct curve for confidentiality.

**Comment 17** — L 431: Same as comment on L 380.

**Reply**: Please refer to Comment 14 for our answer.

**Comment 18** — L 440: It is unclear what "profiles at $y/D\pm0.5$" means as those specific spanwise locations correspond to a point value in the velocity deficit, not "profiles".

**Reply**: We apologise for the ambiguity. We have rephrased the sentence to:
Interestingly, the spanwise velocity profiles at various upstream positions exhibits a slight asymmetry.

**Comment 19** — L 442: The concept of "asymmetric induction" here is repeated and not explained. Again, the cited references do not mention any asymmetric induction but focus on the effects of stability and veer on the wake morphology.

**Reply**: We have removed the term "asymmetric induction zone " as the reference stated are primarily discussing the wake shape under wind veer and shear.

**Comment 20** — L 453-456: You could cite Fleming et al, 2018 to support the fact that wake rotation and counter-rotating vortices sum up in the positive yaw case and cancel out in the negative one.

**Reply**: We have added reference to Fleming et al.,2018 and modified the text to the following:

The findings correspond to Fleming et al.,2018, where a stronger wake deflection for positive yaw case is seen due to the aggregated effect of the wake rotation and counter-rotating vortices in comparison to the negative deflection case.

**Comment 21** — L 457: Which velocity? The lateral one?

**Reply**: Here we refer to the longitudinal velocity and have modified the text accordingly.

**Comment 22** — Figures 16 and 17: please swap the plots to show first the positive yaw as in Figure 15.

**Reply**: We have swapped the plots to maintain consistency with Fig. 15.

**Comment 23** — L 474-475: It is hard to see the claimed stronger induction for negative y. Maybe provide a quantitative parameter like the velocity ratio between inflow and location closer to rotor plane.

**Reply**: We estimated the velocity reduction between the most upstream scan location ($0.8 \ D$) and very close to the rotor plane ($0D$) by calculating the velocity ratio between the two locations at various spanwise locations given by the following equation:

$$v_{\mathrm{ratio}} = \frac{v_{0.8\mathrm{D}}}{v_{0\mathrm{D}}} \tag{1}$$

For brevity, we show the velocity reduction ratio at spanwise positions $y/D = 0.19$ and $y/D = -0.19$ as the trend remained similar towards the rotor tips. For the positive yaw case, the value of $v_{\mathrm{ratio}}$ was 1.2 and 1.14 at $y/D = -0.19$ and $y/D = 0.19$ respectively. Similarly, for the negative yaw case, the value of $v_{\mathrm{ratio}}$ was 1.23 and 1.04 at $y/D = -0.19$ and $y/D = 0.19$ respectively. We have added the following text to the revised manuscript:

The induction strength between the left and right sides of the rotor is quantified using the ratio of the longitudinal wind speed far upstream of the scan at 0.8 $D$ and the rotor plane at 0 $D$ as $v_{\mathrm{ratio}} = \frac{v_{0.8D}}{v_{0D}}$. For the positive yaw case, the value of $v_{\mathrm{ratio}}$ was 1.2 and 1.14 at $y/D = -0.19$ and $y/D = 0.19$ respectively. Similarly for the negative yaw case, the value of $v_{\mathrm{ratio}}$ was 1.23 and 1.04 at $y/D = -0.19$ and $y/D = 0.19$ respectively with similar trends observed at other spanwise positions.

**Comment 24** — L 477-478: The wake recovery followed by the deceleration is only visible for the positive yaw case.

**Reply**: We regret that our statement on the paper has been misinterpreted. We have not noted that the wake recovery followed by deceleration was visible in the positive yaw case. We simply wanted to convey that the induction effect is stronger at $y/D < 0$ which was exposed to the wake for the negative yaw case. We have added the following text to the revised manuscript:

For the negative offset case, similar effects of the induction are seen where the wake at $y/D < 0$ decelerated faster compared to the freestream at $y/D > 0$.

**Comment 25** — L 496: why is the probe averaging not included as possible source of errors?

**Reply**: Probe volume averaging is indeed a source of error and we have added this in the revised version.

**Comment 26** — L511 - The explanation of the angle of attack as the cause of non-symmetrical induction is not sound (see above). The cited Bastankhah paper shows a symmetrical induction for 0-yaw misalignment (below):

**Reply**: Bastankah et al [2], in their measurements of a turbine operating with zero yaw noted "a slight lateral asymmetry with respect to the rotor axis" visualised with the iso-velocity contours at far upstream positions $(x/D \approx 1)$ with the asymmetry vanishing towards the rotor plane, similar to our measurements.

We have updated the explanation of the non-symmetrical induction to also include the interaction of wind shear with wake rotation and the effect of terrain and the tall tree line.

**Comment 27** — L 524-525: the non-symmetrical induction one is hard to see (see above).

**Reply**: The non-symmetrical induction for the aligned inflow case is visible through the iso-velocity contours at far upstream locations $(x/D \approx 1)$ and disappears towards the rotor plane.

**Comment 28** — L 550: the explanation of the shear as the cause of non-symmetrical induction is not sound (see above).

**Reply**: We have updated the explanation of the non-symmetrical induction to also include the interaction of wind shear with wake rotation and the effect of terrain and the tall tree line. We hope that these changes to the manuscript will provide a suitable explanation for the measurements.

**References**

[1] R. J. Barthelmie. The effects of atmospheric stability on coastal wind climates. *Meteorological Applications*, 6(1):39–47, 3 1999.

[2] M. Bastankhah and F. Porte-Agel. Wind tunnel study of the wind turbine interaction with a boundary-layer flow: Upwind region, turbine performance, and wake region. *Physics of Fluids*, 29(6), 2017.

[3] BKG. Digitales Geländemodell Gitterweite 200 m, 2013.

[4] E. Branlard and A. R. Meyer Forsting. Assessing the blockage effect of wind turbines and wind farms using an analytical vortex model. *Wind Energy*, 23(11):2068–2086, 11 2020.

[5] M. Bromm, A. Rott, H. Beck, L. Vollmer, G. Steinfeld, and M. Kühn. Field investigation on the influence of yaw misalignment on the propagation of wind turbine wakes. *Wind Energy*, 21(11):1011–1028, 11 2018.

[6] J. Counihan, J. C. Hunt, and P. S. Jackson. Wakes behind two-dimensional surface obstacles in turbulent boundary layers. *Journal of Fluid Mechanics*, 64(3):529–564, 1974.

[7] P. Hulsman, C. Sucameli, V. Petrović, A. Rott, A. Gerds, and M. Kühn. Turbine power loss during yaw-misaligned free field tests at different atmospheric conditions. *Journal of Physics: Conference Series*, 2265(3):032074, 5 2022.

[8] H. A. Madsen, V. Riziotis, F. Zahle, M. O. Hansen, H. Snel, F. Grasso, T. J. Larsen, E. Politis, and F. Rasmussen. Blade element momentum modeling of inflow with shear in comparison with advanced model results. In *Wind Energy*, volume 15, pages 63–81. John Wiley and Sons Ltd, 1 2012.

[9] A. R. Meyer Forsting, M. P. Van Der Laan, and N. Troldborg. The induction zone/factor and sheared inflow: A linear connection? In *Journal of Physics: Conference Series*, volume 1037, page 072031. IOP Publishing, 6 2018.

[10] T. Mikkelsen, M. Sjöholm, P. Astrup, A. Peña, G. Larsen, M. F. van Dooren, and A. P. Kidambi Sekar. Lidar Scanning of Induction Zone Wind Fields over Sloping Terrain. *Journal of Physics: Conference Series*, 1452(1):012081, 1 2020.

[11] A. S. Monin and A. M. Obukhov. Basic laws of turbulent mixing in the surface layer of the atmosphere. *Originally published in Tr. Akad. Nauk SSSR Geophiz. Inst*, 24(151):163–187, 1954.

[12] A. Peña and J. Mann. Turbulence Measurements with Dual-Doppler Scanning Lidars. *Remote Sensing*, 11(20):2444, 10 2019.

[13] C. Stawiarski, K. Traumner, C. Knigge, and R. Calhoun. Scopes and challenges of dual-doppler lidar wind measurements-an error analysis. *Journal of Atmospheric and Oceanic Technology*, 30(9):2044–2062, 2013.

[14] N. Tobin, A. M. Hamed, and L. P. Chamorro. Fractional Flow Speed-Up from Porous Wind-breaks for Enhanced Wind-Turbine Power. *Boundary-Layer Meteorology*, 163(2):253–271, 5 2017.

[15] M. F. van Dooren, F. Campagnolo, M. Sjöholm, N. Angelou, T. Mikkelsen, and M. Kühn. Demonstration and uncertainty analysis of synchronised scanning lidar measurements of 2-D velocity fields in a boundary-layer wind tunnel. *Wind Energy Science*, 2(1):329–341, 6 2017.

[16] M. F. van Dooren, D. Trabucchi, and M. Kühn. A methodology for the reconstruction of 2D horizontal wind fields of wind turbinewakes based on dual-Doppler lidar measurements. *Remote Sensing*, 8(10):809, 9 2016.

---

## Referee Report (RR1)

**Review of the revised manuscript « Synchronised WindScanner Field Measurements of the Induction Zone Between Two Closely Spaced Wind Turbines"**

**Summary:** The study presents measurements of the flow between two closely spaced wind turbines using a ground-based dual Doppler lidar setup. Four data sets for a fully waked inflow (with wake steering), a partially waked inflow, and an undisturbed inflow are described and discussed in detail. Additionally, a detailed error study is presented including a LES simulation of the setup to investigate the dual-Doppler lidar reconstruction errors. The case study is quite interesting, because the near wake and the induction zone have seen much less attention in literature than the far wake – especially from field measurements.

**General comments**

I have not detected any major flaws with the manuscript. My specific comments below mostly concern a more precise description of the methods. I also noticed several instances of missing words, wrong sentence structures, or one-sentence paragraphs and I recommend that the authors iterate the manuscript more to improve this aspect.

The replies of the authors to the comments of the reviewers are mostly satisfactorily in my opinion.

**Specific Comments**

Line 50-51: A sentence introducing Doppler lidars could be added here.

Line 172: If the x-axis is the connecting line between the two turbines and only wind directions approximately parallel to it are considered, then the y-axis should be pointing in the lateral direction and not the longitudinal direction.

Line 187: "met mast hub height" is not quite clear.

Line 259: The filtering method for hard targets is not described in the manuscript, but data at the rotor plane is used later (e.g. Figure 12). How where the measurements affected by hard-targets identified and discarded?

Table 3: Caption does not mention the +/- term for the wind direction. I assume it is the standard deviation?

Line 305: Is it the gradient of the temperature or the potential temperature?

Figure 7 / Reply to Reviewer 2, Comment 2: Normalizing the statistical uncertainty of the v-component with the longitudinal wind speed might be misleading, because the resulting percentage values are not useful as a relative error of the lateral velocity. Showing the non-normalized uncertainty should be considered for the lateral velocity.

Line 442: Please specify to which fit is referred here.

Line 492: For consistency of language, using "lateral velocity" instead of "spanwise velocity" would be better (I believe all previous instances used lateral velocity for v).

Line 502-506: Aside from the yaw difference between Case 3 and Case 4, there is also a 10° wind direction difference. For Case 3, the average wind direction is 217° and a wake that is offset to the left from the WT1-WT2 line would be expected from this as well. Is the found difference in wake

deflection between the two cases larger than what would be expected for "straight wakes" for a 10° difference in wind direction alone?

Line 506-507: Along the same line as above, I wonder what the impact of different wind directions is on lateral velocity component. Because the coordinate system was defined along the WT1-WT2 line and not parallel to the wind direction, the wind direction offset from the x-axis will be projected into the lateral velocity. Can this be quantified and is it smaller than the observed difference in lateral velocity between the two cases?

The reply to Reviewer 1, Comment 4 seems to support that an assumed line-of-sight velocity accuracy of 0.1% is too low. Would the results hold for the more realistic 2% error? Or can a threshold be provided until which the results hold?

**Technical Comments**

Line 127: I believe "scanner head" is more common than "scan head".

Eq. (1): Comma instead of full stop.

Line 145: The u and v variables were already introduced in line 140.

Line 204: Insert "it" in "as is".

Line 211-212: Sentence structure (remove first and).

Line 220: Remove "a" in "until a multiple scans".

Line 290:  Either plural for performance or replace "were" with "was".

Line 360: No new paragraph here.

Figure 9: Longitudinal error should $e_u$ not $e_v$ in the caption.

Line 500: No new paragraph.

---

## Referee Report (RR2)

**Comments to the first revision of WES-114-2023**

Thanks for implementing the suggested changes. The statistical uncertainty part needs additional work, as shown below.

The main criticality, however, remains the fact the calculated uncertainty bounds are far larger than some difference observed in the mean velocity fields. The justification that the error due to the vertical velocity may be overestimated is not compelling. Generally, one provides a maximum error value to estimate a cap to the uncertainty and show that "despite the error being overestimated the uncertainty is below the acceptable value etc…" This is not the case for this work where a lot of speculations are made to interpret differences in the mean flow way smaller than the uncertainty bands. Please remove all those parts discussing effects that do not pass the uncertainty test such as the non-symmetric induction zone. The paper is already quite long and the readers will benefit from the improved conciseness.

**New comments**

Figure 7: please use either always percentage errors (e.g 5%) or non-dimensional error (e.g 0.05).

L 570: "A deeper analysis into the propagated uncertainties indicated that the error in the u,v component estimation was primarily because of the volume averaging effect, beam-intersection angles and beam-pointing errors while the w = 0 m/s assumption was the most dominant source of error. ":  so are the volume averaging, beam angles and pointing accuracy the main sources of error, or is it the vertical velocity assumption?

L 665: please remove that this paper shows how one "should analyse" field as it sounds too bold.

**Main comments**

2: Thanks for adding some statistical uncertainty quantification. This is however not enough for two reasons:

- The use of $1.96\,\sigma/\sqrt{N}$ is not justified. This simple formula relies on the assumptions of large $N$ (central limit theorem and use of estimator instead of true variance) and independent samples [1]. While the 71 scan repetition can be considered a large enough sample size, in turbulent flows the samples are never independent due to presence of a non-0 integral time scale. Please ensure that the scan repetition time is slow enough to assume that the samples are separated in time by several integral time scales.

- In the conclusion, it is stated that the statistical uncertainty is smaller than the uncertainty of dual Doppler. Statistical uncertainty indeed seems to be ignored after section. Statistical uncertainty being smaller does not mean negligible however, as its magnitude is still a few percent of the freestream velocity. Please add the statistical uncertainty also to the experimental results. It is usually ok to consider it independent from the instrumental uncertainty to that it can be squared-sum to the propagated uncertainty already present in the paper.

**Specific comments**

6: The comment was more intended to point out the missing justification for the values of $z/L$ used to define the stability classes. More specifically, why is $\frac{z}{|L|} < 0.4$ chosen as the neutral class?

11: the error on the location was not corrected in the new version, there is still $\frac{x}{D} = -0.16$.

12: please add that stable stratification also suppressed vertical displacement of air parcels thus enhancing flow blockage as possible explanation of the stronger induction.

13: Thanks for pointing out Bastankhah's explanation for this asymmetry. In the text, it should be made clearer that this is a dynamic aerodynamics or stall effect, but that the angles of attack are perfectly symmetric in a quasi-static sense.

15: every time the "uncertainty bounds" are cited in the text, the confidence level should be mentioned too. E.g. $1.96\,\sigma$ would correspond to 95% confidence level if Gaussianity is assumed, and so forth.

24: it is still confusing. It is first stated that the $y < 0$ sides shows faster deceleration. This implies for positive yaw offset, more deceleration in the freestream, for the negative yaw offset more deceleration in the wake. Then is it stated that for the negative offset "**similar** effects of the induction are seen where the wake at y/D < 0 decelerated faster compared to the freestream" which is in contradiction with what happens for the positive offset where the freestream part has stronger deceleration.

**References**

[1] V. Bayley and J. M. Hammersley. The "Effective" Number of Independent Observations in an Autocorrelated Time Series. *Supplement to the Journal of the Royal Statistical Society*, Vol. 8, No. 2 (1946), pp. 184-197.

---

## Referee Report (RR3)

**Comments to second revision of WES-2024-113**

Thanks to the authors for the vast revision implemented. The manuscript can be accepted provided that the two additional comments below are addressed. Please, next time provide a marked up version including only the changes done at the last iteration.

**New comments**

L 520: "This interaction of vertical shear 520 with the wake can lead to an asymmetric velocity distribution as the wake rotation due to difference in wake convection speeds between the upper and lower rotor halves enhances mixing between the low and high momentum regions of the wake": this explanation is not clear. The wake rotation is not "due to differences in wake convection" but is the to the third Newton law applied to the turbine torque. Are the authors trying to say that the wake rotation in case of high shear the wake rotation mixes up layers of fluid with different speeds, as reported at line 609? Please rephrase.

Fig 10: After adding the statistical uncertainty to the propagated uncertainty, the errors seem to be smaller, while they should be bigger by definition (see below). Please provide an explanation and fix it.

[Figure]

*Old figure*                                    *New figure*

---

## Author Response (AR2)

**Synchronised WindScanner Field Measurements of the Induction Zone Between Two Closely Spaced Wind Turbines**

**Anantha Padmanabhan Kidambi Sekar, Paul Hulsman, Marijn Floris van Dooren and Martin Kühn**

April 11, 2024
* * *
**Reviewer 1**

I would like to thank the authors for providing a thorough answer to my questions and comments. The authors added a description of the terrain and a discussion about the potential impact of the ground heterogeneities on the measured wind speed in the induction zone of the WT2 wind turbine.

*Thank you for the feedback and the thorough evaluation of our manuscript. Comments to the reviewer points are made in blue while modifications to the manuscript are shown in red. We hope these changes will positively benefit the manuscript.*
* * *
**Main Comments**

**Comment 1**: In Fig. 12 the authors present measurements of the induction zone of the WT2 wind turbine when the upstream wind turbine was not operating. Based on a visual inspection of that figure, one can observe differences of the inflow wind speed of up to 10% between the y/D>0 and y/D<0 sides. Similar differences are visually observed in Fig. 15 and 16. Furthermore, the wake center in Fig. 15 is not aligned to the center of the rotor, which also introduces an asymmetry to the spanwise profile. For this reason, I do not see how the data presented here reveal a horizontal asymmetry of the longitudinal profile that is a result of the interaction between the strong shear and the wake as the authors suggest. I mention this since the observed asymmetry is highlighted as one of the findings of this study in the abstract and in the conclusions section. I suggest that the authors should comment in the manuscript why they think that the impact of the heterogeneity of the terrain on the flow, which is presented in Fig. 12, is not sufficient to explain the differences presented in Fig. 15.

**Reply**:
Indeed, the presence of the treeline is expected to cause a perturbation of the flow. While measurements of the undisturbed induction zone showed indications of the impact of the terrain heterogeneity, the additional perturbations induced due to the WT1 wake will additionally influence the flow evolution in the WT2 induction zone. Further differences in the wake profiles can be expected, not only from the impact of the treeline and the WT1 wake but also from the differences in the inflow conditions between the undisturbed and fully waked cases. For the full wake case, the inflow is characterised by

high shear ($\alpha = 0.38$) and 19° veer between the top and bottom rotor blade tips. This is in contrast to the undisturbed inflow where a relatively weak shear of $\alpha = 0.21$ was present with a 2° wind veer. Therefore, for the fully waked inflow case, the interaction of the strongly sheared inflow with veering wind into WT2 with the WT1 wake can affect the velocity distribution in the wake profiles, for instance, by promoting mixing between the low and high momentum regions of the rotor [1, 11]. However, due to the large measurement uncertainty, the asymmetric profiles lie within the uncertainty bounds, complicating the interpretation to determine significant flow features, especially in waked measurements.

We have made the following changes in the revised manuscript:

- We have removed the mention of asymmetric induction zone in the abstract and the conclusions.

- In the revised manuscript, we report the asymmetrical velocity distribution in the undisturbed induction zone in Section 3.2.1 and Section 3.2.2, report the possible causes and discuss that the magnitude of the velocity deviations in the horizontal velocity profiles lie within the calculated uncertainty bounds.

- The following text has been added to Section 3.2.2 L506-L514: "The lateral velocity profiles at different upstream positions exhibit a slight asymmetry. While terrain heterogeneity could explain some of the measured features, further differences with the undisturbed inflow case is expected due to the WT1 wake and differences in inflow conditions. For Case 2, the inflow is characterised by high shear and veer between the top and bottom rotor blade tips, in contrast to Case 1. This interaction of vertical shear with the wake can lead to an asymmetric velocity distribution as the wake rotation due to difference in wake convection speeds between the upper and lower rotor halves enhances mixing between the low and high momentum regions of the wake [6, 11, 1]."

- The discussion section (Section 4) is updated to refer to literature which addresses similar observations. Further, the limitations of error analysis for the field measurements is discussed along with recommendations for further measurement campaigns to mitigate limitations of the present study.

**Specific Comments**
* * *
**Comment 1** — Lines $451 - 453$. Is it possible to add the induction factor value used in Fig. 14?

**Reply**: L459: We have added the value of the induction factor ($a = 0.23$) used for the estimation of the induction zone deceleration along the rotor axis in Fig. 13 of the revised manuscript.

**Technical Corrections**
* * *
**Comment 1** — Lines 329 - 330: "A good qualitative agreement between ... is noted ... "

**Reply**: We have changed this in the revised manuscript.

**Comment 2** — Lines 511 – 513. Please add the degree symbol after the values of the yaw misalignment

**Reply**: We have gone through the revised manuscript and added any missing units to variables.

**Reviewer 2**

Thanks for implementing the suggested changes. The statistical uncertainty part needs additional work, as shown below.

The main criticality, however, remains the fact the calculated uncertainty bounds are far larger than some difference observed in the mean velocity fields. The justification that the error due to the vertical velocity may be overestimated is not compelling. Generally, one provides a maximum error value to estimate a cap to the uncertainty and show that "despite the error being over estimated the uncertainty is below the acceptable value etc. . . " This is not the case for this work where a lot of speculations are made to interpret differences in the mean flow way smaller than the uncertainty bands. Please remove all those parts discussing effects that do not pass the uncertainty test such as the non-symmetric induction zone. The paper is already quite long and the readers will benefit from the improved conciseness.

Thank you for the feedback and the thorough evaluation of our manuscript. We acknowledge that the combination of the measurement challenges at the test site and instrumentation limitations led to a large uncertainty in measuring complex flow interactions between the closely spaced turbines. This required a detailed analysis of the associated lidar errors and uncertainties for interpretation of the measurements which was addressed through the high-fidelity simulation approach. The large uncertainty bands resulting from the uncertainty analysis greatly impact how the measurements are interpreted. Therefore, we recognise the validity of the reviewers comments on the interpretation of the field measurements and have reworked the paper to exclude interpretations that lie within the uncertainty bands of the measurements.

The following changes are made:

- References to the symmetrical induction zone:

  - We have removed the description of the asymmetrical induction zone from the abstract and conclusions

  - Section 3.2.1 (L435-455): We report the asymmetrical velocity distribution in the undisturbed induction zone measurements and discuss that the magnitude of the velocity deviations in the horizontal velocity profiles lie within the calculated uncertainty bounds.

  - Section 3.2.2 (L506-L514): We report a asymmetrical velocity distribution in the fully waked induction zone measurements and discuss the difference in the inflow conditions (wind veer and wind shear) that can possibly skew the structure in the spatial distribution of the wake profiles with reference to literature.

  - Section 3.2.3, including Figs. 18, 19 (of the previous revision), the analysis of the induction profiles at 6 spanwise positions for the wake steering cases has been removed as the described effects lie within the uncertainty bounds of the measurements.

  - In the discussion section, only literature is referred which addresses similar observations.The limitations of error analysis for the field measurements is discussed along with recommendations for further measurement campaigns to mitigate limitations of the present study.

- To reduce the length of the paper, we have removed Section 3.1.3 describing the dual-Doppler propagated uncertainty in field measurement and summarised it in Section 3.2.

- An extended statistical uncertainty analysis is presented in Section 3.1.1 based on the requested main corrections.

With these changes, we hope to have addressed the main concern of the reviewer on the interpretation of the measurements and condensed the paper sufficiently to benefit the reader.

**New Comments**

**Comment 1** — Figure 7: please use either always percentage errors (e.g 5%) or non-dimensional error (e.g 0.05).

**Reply**: As suggested by Reviewer 3, we now show the absolute magnitude of the errors for both the longitudinal and lateral component.

**Comment 2** — L 570: "A deeper analysis into the propagated uncertainties indicated that the error in the u,v component estimation was primarily because of the volume averaging effect, beam-intersection angles and beam-pointing errors while the w = 0 m/s assumption was the most dominant source of error. ": so are the volume averaging, beam angles and pointing accuracy the main sources of error, or is it the vertical velocity assumption?

**Reply**: We have changed this in the revised manuscript.
"A deeper analysis of the propagated uncertainties indicated that the main contribution to the uncertainty to estimate the $u, v$ component was the $w = 0$ m/s assumption. Other important sources of the uncertainty were the probe volume averaging effect, the inaccuracy of the beam-intersection angles and the beam-pointing errors. "

**Comment 3** — L 665: please remove that this paper shows how one "should analyse" field as it sounds too bold

**Reply**: We have removed this in the revised manuscript. It now reads:
"The study further highlights the challenges in conducting field measurements, and the additional considerations needed to characterise the induction zone behaviour."

**Main Corrections**

**Comment 1**:
Thanks for adding some statistical uncertainty quantification. This is however not enough for two reasons:

- The use of $\frac{1.96\sigma}{\sqrt{N}}$ is not justified. This simple formula relies on the assumptions of large N (central limit theorem and use of estimator instead of true variance) and independent samples. While the 71 scan repetition can be considered a large enough sample size, in turbulent flows the samples are never independent due to presence of a non-0 integral time scale. Please ensure that the scan repetition time is slow enough to assume that the samples are separated in time by several integral time scales.

- In the conclusion, it is stated that the statistical uncertainty is smaller than the uncertainty of dual Doppler. Statistical uncertainty indeed seems to be ignored after section. Statistical uncertainty being smaller does not mean negligible however, as its magnitude is still a few percent of the freestream velocity. Please add the statistical uncertainty also to the experimental results. It is usually ok to consider it independent from the instrumental uncertainty to that it can be squared-sum to the propagated uncertainty already present in the paper

**Reply**:

- We acknowledge that we have not provided any justification for the usage of the statistical uncertainty formula. Indeed the utilisation of $\frac{1.96\sigma}{\sqrt{N}}$ requires a large sample size and that the measurement samples are independent of each other. To analyse the independence of the measurement samples, we first calculate the integral timescale ($T_i$) based on the auto-correlation of the wind speed time series for all presented cases following Cheynet et al [4]. The $T_i$ for the field measurements is calculated from the longitudinal velocity measurements from the hub height anemometer mounted on the met mast. Figure 1 illustrates the auto-correlation function for the measurements and simulations.
    The $T_i$ is obtained by integrating the auto-correlation function till the first-zero crossing and

[Figure]

Figure 1: Plot of the autocorrelation function of the five cases. The LES data was extracted at the location $x/D = -0.75, y/D = 0, z = 136$ m.

tabulated in Tab 1. It should be noted that the time scales for the field cases could differ from the time scales present in the scanning area due to the separation between the met mast and the scan area. Furthermore, the presence of the WT1 wake in Cases 2, 3, and 4 will reduce the time scales present in the scanning area due to increased turbulence in the near wake depending on the evolution of the turbulent eddies in the wake.

For all cases, the integral length scale is smaller than the scan repetition rate by a factor of at least 2. Henk and Lumey [5] describe that, for statistical independence, sampling the wind once every two integral time scales is adequate. Therefore, while the measurements may not be entirely independent due to the relatively short integral time scale compared to the scanning time, they may still be treated as approximately independent.

We now calculate the statistical uncertainty using $\frac{1.96\sigma}{\sqrt{N_s}}$ to estimate the statistical uncertainty where $N_s$ is the number of independent sample size. This effective sample size accounts for correlations in the turbulent flow, leading to a more accurate estimate of the error of the mean in comparison to the number of measured samples and is calculated as:

$$N_s = N\langle\frac{1-r}{1+r}\rangle \tag{1}$$

Where $r$ is the lag-1 auto correlation [9]. The effective sample sizes are presented in Tab 1 and subsequently used for calculating the statistical uncertainty.

We have added the following information to the revised manuscript:

– We have updated Table 3 describing the inflow conditions with the integral time scales provided in Table 1 .

– Revised Section 3.1.1 to include the method for estimation of independent sample size that is thereafter used for statistical uncertainty estimation.

| Case | $T_i$ (s) | Scan time multiples | N | $N_s$ |
|------|-----------|---------------------|-----|-------|
| 1 | 9.4 | 3.1 | 107.4 | 95.4 |
| 2 | 10.5 | 2.8 | 52.7 | 41.1 |
| 3 | 11.1 | 2.6 | 70.9 | 25.5 |
| 4 | 9.10 | 3.2 | 64.8 | 44.7 |
| LES | 4.5 | 6.5 | 71.0 | 64.7 |

Table 1: Estimations of the integral time scale ($T_i$), number of measured samples ($N$) and the number of independent samples $N_s$ for the field measurements and the LES.

• Assuming a perfectly calibrated lidar system with no measurement bias and uncorrelated errors, we have added the statistical uncertainty to the propagated uncertainty through the squared sum approach to estimate the total combined uncertainty. The total combined uncertainty accounts for the statistical variability in the measured flow in addition to the variability due to the lidar limitations.

– Revised Section 3.1.2 on how the total combined uncertainty was estimated. "The shaded region illustrates the total measurement uncertainty where the statistical uncertainty and the

propagated uncertainty are summed in quadrature assuming a perfectly calibrated lidar with no measurement bias and uncorrelated errors. The total combined uncertainty accounts for the statistical variability in the measured flow in addition to the variability due to the lidar limitations."

  – Figures 10, 12, 13, 15, 17 of the revised manuscript are now updated to show the total combined measurement uncertainty.

**Technical Corrections**
* * *
**Comment 1** — 6: The comment was more intended to point out the missing justification for the

values of $z/L$ used to define the stability classes. More specifically, why is $z/L < 0.4$ chosen as the neutral class?

**Reply**: Thank you for the clarification. The stability classification was provided in the paper following the free field lidar experiments of Simley et al., [7] who had also used the $z/L$ parameter to characterise stability for their WindScanner measurements. We have further corrected a typo whereby neutral conditions is selected for $-0.04 \leq z/L \leq 0.04$.
"The stability classification of the Obhukov parameter $z/L$ is performed for 30-minute averages based on Wyngaard [10] and further used in Simley et al [7], where negative values indicate the presence of unstable conditions ($z/L \leq -0.04$), positive values ($z/L \geq 0.4$) correspond to stable conditions, and values close to zero ($-0.04 \leq z/L \leq 0.04$) are related to neutral conditions. "

**Comment 2** — 11: The error on the location was not corrected in the new version, there is still $x/D = 0.16$

**Reply**: In the revised manuscript we have indicated that the profiles are extracted at $x/D$ = -0.08 in alignment with the horizontal profiles plotted in Fig. 10.

**Comment 3** — 12: please add that stable stratification also suppressed vertical displacement of air parcels thus enhancing flow blockage as possible explanation of the stronger induction

**Reply**: We have changed this in the revised manuscript to the following:
"This strong velocity deficit can be attributed to high axial induction and weakly stable stratification during the measurement period inhibiting vertical displacement of air particles further enhancing the blockage. "

**Comment 4** — 3: Thanks for pointing out Bastankhah's explanation for this asymmetry. In the text, it should be made clearer that this is a dynamic aerodynamics or stall effect, but that the angles of attack are perfectly symmetric in a quasi-static sense

**Reply**: We have changed this in the revised manuscript to the following:
"This asymmetry can potentially be attributed to the dynamic interaction between the vertical shear and the rotating blades, which was noted by [2] using wind tunnel measurements."

**Comment 5** — 15: every time the "uncertainty bounds" are cited in the text, the confidence level should be mentioned too. E.g. 1.96 $\sigma$ would correspond to 95% confidence level if Gaussianity is assumed, and so forth

**Reply**: We have added the following text to Section 2.3.2 of the revised manuscript:
"All the uncertainty terms in the paper are the 1.96 $\sigma$ values of the corresponding error distributions; i.e. they are expected to include 95 % of all values."

**Comment 6** — 24: it is still confusing. It is first stated that the $y/D < 0$ sides shows faster deceleration. This implies for positive yaw offset, more deceleration in the freestream, for the negative yaw offset more deceleration in the wake. Then is it stated that for the negative offset "similar effects of the induction are seen where the wake at $y/D < 0$ decelerated faster compared to the freestream" which is in contradiction with what happens for the positive offset where the freestream part has stronger deceleration

**Reply**: We have removed the spanwise induction figures and explanation in the revised manuscript as described earlier as the measured flow features are within the uncertainty bounds of the measurements.

**Reviewer 3**

**Summary:** The study presents measurements of the flow between two closely spaced wind turbines using a ground-based dual Doppler lidar setup. Four data sets for a fully waked inflow (with wake steering), a partially waked inflow, and an undisturbed inflow are described and discussed in detail. Additionally, a detailed error study is presented including a LES simulation of the setup to investigate the dual-Doppler lidar reconstruction errors. The case study is quite interesting, because the near wake and the induction zone have seen much less attention in literature than the far wake – especially from field measurements.

**General comments** I have not detected any major flaws with the manuscript. My specific comments below mostly concern a more precise description of the methods. I also noticed several instances of missing words, wrong sentence structures, or one-sentence paragraphs and I recommend that the authors iterate the manuscript more to improve this aspect.
The replies of the authors to the comments of the reviewers are mostly satisfactorily in my opinion.

*We thank the reviewer for their critical assessment of our work with general and specific comments. The revised manuscript has been carefully proofread for spelling and grammar errors. In the following section, we address the specific comments point by point. Comments to the reviewer points are made in blue while modifications to the manuscript are shown in red. We hope these changes will positively benefit the manuscript.*

**Specific Comments**

**Comment 1** — Line 50-51: A sentence introducing Doppler lidars could be added here.

**Reply**: We have changed this in the revised manuscript to:
"Lidars are capable of measuring the velocity through the Doppler shift remotely and provide a way to measure the flow around wind turbines in the field [8]."

**Comment 2** — Line 172: If the x-axis is the connecting line between the two turbines and only wind directions approximately parallel to it are considered, then the y-axis should be pointing in the lateral direction and not the longitudinal direction.

**Reply**: We have changed this in the revised manuscript:
"The measurements are visualised in a global fixed reference frame centred at the bottom of WT2, where the $x$-axis is the connecting line between the two turbines, and the $y-$ and $z-$ axes are positive to the right looking towards WT2 and in an upward direction."

**Comment 3** — Line 187: "met mast hub height" is not quite clear.

**Reply**: We have changed this in the revised manuscript:
"We rotated all measurements in the global reference frame into the main wind direction measured at the met mast at 1 m below WT1 hub height"

**Comment 4** — Line 259: The filtering method for hard targets is not described in the manuscript, but data at the rotor plane is used later (e.g. Figure 12). How where the measurements affected by hard-targets identified and discarded?

**Reply**: The dynamic filtering method from Beck and Kühn [3] filters for the line-of-sight velocity and the signal quality in a bi-variate manner based upon the assumption of self-similarity of valid data. Plotting the LOS and signal quality (SNR) together, clusters of data points corresponding to the measurements and hard targets such as moving blades, clustered at different levels of signal quality can be identified owing to their differing signal quality and hence are removed. It is noted that the filtering is applied on the LOS measurements collected over the entire scanning area for the duration of the measurements. Hard targets such as the nacelle could be easily identified from the LOS-SNR distribution while the filtering of the blade interference is dependent on the blade azimuthal angle.
We could plot the data at the rotor plane as the measurements are presented as averages for the total measurement duration and valid measurements could be recorded at the rotor plane when devoid of blade interference.
We have added the following text in Section 3 L291-L297:
"Data filtering for the field measurements was performed using a kernel density-based filter based on [3] to identify and remove low-quality measurements. The method filters for the line-of-sight velocity and the Signal-Noise-Ratio in a bi-variate manner based upon the assumption of self-similarity of valid data. The method is applied on all the collected $v_{\mathrm{vlos}}$ measurements on the measurement plane and is capable of identifying hard targets such as the nacelle and blades through the clusters in the $v_{\mathrm{vlos}}$-SNR space. The measurements are discretized and grouped into bins based on their $v_{\mathrm{vlos}}$- SNR values. The frequency distribution of data points within each bin was then determined. Bins with frequencies exceeding 20 % of the most populated bin were retained for further analysis. "

**Comment 5** — Table 3: Caption does not mention the ± term for the wind direction. I assume it is the standard deviation?

**Reply**: The ± term indicates standard deviation of the wind direction. The table caption has been updated.

**Comment 6** — Is it the gradient of the temperature or the potential temperature?

**Reply**: The gradient refers to the potential temperature gradient. We have updated the revised manuscript with this information.

**Comment 7** — Figure 7 / Reply to Reviewer 2, Comment 2: Normalizing the statistical uncertainty of the v-component with the longitudinal wind speed might be misleading, because the resulting percentage values are not useful as a relative error of the lateral velocity. Showing the non-normalized uncertainty should be considered for the lateral velocity

**Reply**: Thank you for this comment. We now show the un-normalised longitudinal and lateral velocity error in Figures 7, 8.

**Comment 8** — Line 442: Please specify to which fit is referred here

**Reply**: Thank you for pointing this out. We wanted to refer to the fact that due to low data return, the velocity along the rotor axis was not plotted behind the rotor.
We have rephrased the sentence to:
"Data availability between $0 \leq x/D \leq 0.2$ is reduced due to the presence of the nacelle and therefore excluded."

**Comment 9** — Line 492: For consistency of language, using "lateral velocity" instead of "spanwise velocity" would be better (I believe all previous instances used lateral velocity for v)

**Reply**: We have replaced the term "spanwise velocity" with "lateral velocity" throughout the revised manuscript.

**Comment 10** — Line 502-506: Aside from the yaw difference between Case 3 and Case 4, there is also a 10° wind direction difference. For Case 3, the average wind direction is 217° and a wake that is offset to the left from the WT1-WT2 line would be expected from this as well. Is the found difference in wake deflection between the two cases larger than what would be expected for "straight wakes" for a 10° difference in wind direction alone?

**Reply**: The 10° difference from the WT1-WT2 line for Case 3 would influence the wake from WT1 and move it further left of WT2 looking downstream. The additional positive yaw offset at WT1 will further deflect the wake in this direction. This is supported by the location of the wake for the positive yaw case at $y/D = 0.32$ in comparison to the wake being present at $y/D = $ -0.20 in the negative yaw case.
In a scenario where the two turbines operate with a 10° difference in wind direction from the WT1-WT2 line without any wake steering applied on WT1, it is expected that the wake will not be deflected further to the left in comparison to when wake steering is applied. However, we cannot provide a definitive answer as the wake behaviour is highly dependant on the inflow and turbine conditions and such a wake case with a 10° offset between the 228° line and the wind direction while the turbines were operating in a straight wake case was not measured.
We have made the following changes to the manuscript:
For clarity, Figure 16 in the revised manuscript has been updated to show the average wind direction during the positive and negative wake steering cases.

**Comment 11** — Line 506-507: Along the same line as above, I wonder what the impact of different wind directions is on lateral velocity component. Because the coordinate system was defined along the WT1-WT2 line and not parallel to the wind direction, the wind direction offset from the x-axis will be projected into the lateral velocity. Can this be quantified and is it smaller than the observed difference in lateral velocity between the two cases?

**Reply**: Yes, due to our definition of the coordinate system where the x-axis is aligned to the WT1-WT2 line at a heading of 228°, any changes in wind direction would be projected onto the velocities on the scanning area. This would be in addition to the lateral velocity magnitude measured in the defined coordinate system in comparison to the aligned case. This can be already seen in Fig 17 top right, where the $v$ component magnitudes are much larger in comparison to the relatively more aligned negative steering case. A preliminary quantification can be provided by calculating the addition to the

lateral velocity component as $u_\infty \sin(\Phi)$ where $u_\infty$ is the wind velocity and $\Phi$ is the offset from the WT1-WT2 line. Therefore, for the 10° offset case, approximately 17% of the wind magnitude will be projected into the lateral component.

For cases 3 and 4, this value is still slightly smaller than the observed differences in the lateral velocities between the two cases across the scanning area.

We have made the following changes to the manuscript at L525-L528:

"In both cases, the maximum magnitude of the lateral velocity inside the deflected wake is approximately $0.2\ u_\infty$ to $0.25\ u_\infty$. The positive yaw offset case exhibits a comparatively more substantial lateral flow component compared to the negative yaw offset due to the 10° misalignment between the turbine orientation and the wind direction as the lateral velocity would be increased by the projection of misaligned inflow into the defined coordinate system."

**Comment 12** — The reply to Reviewer 1, Comment 4 seems to support that an assumed line-of-sight velocity accuracy of 0.1% is too low. Would the results hold for the more realistic 2% error? Or can a threshold be provided until which the results hold?

**Reply**: To estimate the error in the 2D velocity estimation for a range of line-of-sight errors $(e_{\mathrm{vlos},1}, e_{\mathrm{vlos},2})$ we use the SUP methodology described in Eqns 7, 8 in the revised manuscript applied on the LES wind field.

[Figure]

Figure 2: Variation of $e_{\mathrm{u}}$ and $e_{\mathrm{v}}$ to $e_{\mathrm{vlos}}$ at locations P1 to P6 in the LES wind fields.

Figure 2 shows the variation of $v_{\mathrm{los}}$ contribution to the u and v component error for a range of $v_{\mathrm{los}}$ errors from 0.05 % to 25 % assuming same $e_{\mathrm{vlos}}$ for both lidar systems. The results are shown for 6 spatial locations P1...P6 distributed over the scanning area as described in Section 3.1.2 of the revised manuscript. The spatial variations of $e_{\mathrm{u,v}}$ at the different locations is due to the variations in the $w$ components and beam scanning angles contributing to the total error.

At each spatial location, $e_{\mathrm{u}}, e_{\mathrm{v}}$ does not show large variations between $e_{\mathrm{vlos}}$ of 0.1 % utilised in the paper and a realistic 2 % error requested by the reviewer. Both $e_{\mathrm{u}}, e_{\mathrm{v}}$ show larger sensitivity to higher $e_{\mathrm{vlos}}$ simply due to the magnitude of the line-of-sight error becoming larger than the other error sources $(e_{\mathrm{w}}, e_\chi, e_\delta)$. Therefore, the presented results will hold for a more realistic 2% $v_{\mathrm{los}}$ error.

We have added the following in Section 4 Discussion of the revised manuscript:

Further WindScanner simulations indicated that the total propagated error was insensitive to a higher and more realistic 2 % line-of-sight error.

**Technical Comments**

**Comment 1** — Line 127: I believe "scanner head" is more common than "scan head".

**Reply**: We have changed this in the revised manuscript.

**Comment 2** — Eq. (1): Comma instead of full stop.

**Reply**: We have changed this in the revised manuscript.

**Comment 3** — Line 145: The u and v variables were already introduced in line 140.

**Reply**: We have changed this in the revised manuscript to the following:
"The $u, v$ velocity components can be resolved by an additional assumption of the vertical flow component and combining the two $v_{\mathrm{los}}$ measurements by dual-Doppler wind field reconstruction by solving Eq. (2)."

**Comment 4** — Line 204: Insert "it" in "as is"

**Reply**: We have changed this in the revised manuscript.

**Comment 5** — Line 211-212: Sentence structure (remove first and).

**Reply**: We have changed this in the revised manuscript.

**Comment 6** — Line 220: Remove "a" in "until a multiple scans".

**Reply**: We have changed this in the revised manuscript.

**Comment 7** — Line 290: Either plural for performance or replace "were" with "was".

**Reply**: We have changed this in the revised manuscript.

**Comment 8** — Line 360: No new paragraph here.

**Reply**: We have removed the single line paragraph and added it to the next paragraph.

**Comment 9** — Line 500: No new paragraph.

**Reply**: We have removed the single line paragraph and added it to the next paragraph..

**References**

[1] M. Abkar, J. N. Sørensen, and F. Porté-Agel. An Analytical Model for the Effect of Vertical Wind Veer on Wind Turbine Wakes. *Energies 2018, Vol. 11, Page 1838*, 11(7):1838, 7 2018.

[2] M. Bastankhah and F. Porte-Agel. Wind tunnel study of the wind turbine interaction with a boundary-layer flow: Upwind region, turbine performance, and wake region. *Physics of Fluids*, 29(6), 2017.

[3] H. Beck and M. Kühn. Dynamic Data Filtering of Long-Range Doppler LiDAR Wind Speed Measurements. *Remote Sensing*, 9(6):561, 6 2017.

[4] E. Cheynet, J. B. Jakobsen, J. Snæbjörnsson, T. Mikkelsen, M. Sjöholm, J. Mann, P. Hansen, N. Angelou, and B. Svardal. Application of short-range dual-Doppler lidars to evaluate the coherence of turbulence. *Experiments in Fluids*, 57(12):1–17, 12 2016.

[5] Henk Tennekes and John L. Lumley. *A First Course in Turbulence*. The MIT Press, 2018.

[6] N. Sezer-Uzol and O. Uzol. Effect of steady and transient wind shear on the wake structure and performance of a horizontal axis wind turbine rotor. *Wind Energy*, 16(1):1–17, 1 2013.

[7] E. Simley, N. Angelou, T. Mikkelsen, M. Sjöholm, J. Mann, and L. Y. Pao. Characterization of wind velocities in the upstream induction zone of a wind turbine using scanning continuous-wave lidars. *Journal of Renewable and Sustainable Energy*, 8(1):013301, 1 2016.

[8] C. Werner and J. Streicher. Lidar: Range-Resolved Optical Remote Sensing of the Atmosphere. 102, 2005.

[9] D. S. Wilks. Statistical Methods in the Atmospheric Sciences, Fourth Edition. *Statistical Methods in the Atmospheric Sciences, Fourth Edition*, pages 1–818, 1 2019.

[10] J. C. Wyngaard. *Turbulence in the Atmosphere*. Cambridge University Press, Cambridge, 2010.

[11] S. Xie and C. L. Archer. A Numerical Study of Wind-Turbine Wakes for Three Atmospheric Stability Conditions. *Boundary-Layer Meteorology*, 165:87–112, 2017.

---

## Author Response (AR3)

**Synchronised WindScanner Field Measurements of the Induction Zone Between Two Closely Spaced Wind Turbines**

**Anantha Padmanabhan Kidambi Sekar, Paul Hulsman, Marijn Floris van Dooren and Martin Kühn**

May 5, 2024
* * *
**Reviewer 1**

No further comments from Reviewer 1
* * *
**Reviewer 2**

Thanks to the authors for the vast revision implemented. The manuscript can be accepted provided that the two additional comments below are addressed. Please, next time provide a marked up version including only the changes done at the last iteration

*We thank the reviewer for the additional comments. The responses to the reviewer are written in blue while the modifications to the manuscript are shown in red. We further attach a marked up version of the manuscript tracking changes from the previous iteration. We hope that these changes positively benefit the manuscript.*

**New Comments**

**Comment 1** — L 520: "This interaction of vertical shear 520 with the wake can lead to an asymmetric velocity distribution as the wake rotation due to difference in wake convection speeds between the upper and lower rotor halves enhances mixing between the low and high momentum regions of the wake": this explanation is not clear. The wake rotation is not "due to differences in wake convection" but is the to the third Newton law applied to the turbine torque. Are the authors trying to say that the wake rotation in case of high shear the wake rotation mixes up layers of fluid with different speeds, as reported at line 609? Please rephrase

**Reply**:  We are indeed referring to the explanation that the rotation of the wake mixes up vertical layers of fluid moving at different velocity due to the vertical shear that can lead to an asymmetric wake velocity distribution. We have rephrased the sentence to the following:
"This interaction of vertical shear with the wake can lead to an asymmetric velocity distribution as the wake rotation mixes the different layers of fluid in the vertically sheared flow."

**Comment 2** — Fig 10: After adding the statistical uncertainty to the propagated uncertainty, the errors seem to be smaller, while they should be bigger by definition (see below). Please provide an explanation and fix it

**Reply**:  Thank you for pointing this out. We have corrected our plotting routines and have updated Figure 10 to ensure that the statistical uncertainty and propagated uncertainty are added in quadrature at every location on the spatial grid. The differences in the wake profiles is shown in Fig. 1 between the revised manuscript (red shaded areas) and the original manuscript (green shaded areas). The updated figure now illustrates that the LES wake profiles have larger error bars due to the addition of the statistical uncertainty to the propagated uncertainty. Due to the normalisation of the wake profiles with mean wind speed $u_\infty = 7.7$ m/s, the differences between the wake profiles with and without the addition of the statistical uncertainty are small, however as expected the total uncertainty is larger than the propagated uncertainty.

[Figure]

Figure 1: Updated WindScanner longitudinal velocity wake profiles presented in the revised manuscript (red shaded areas) and the original manuscript (green shaded areas).

**Reviewer 3**

I am satisfied with author's response to my comments and the resulting changes to the manuscript. I only have a minor and a technical comment on the new additions:

*We thank the reviewer for their further review of the revised manuscript. The responses to the reviewer are written in* blue *while the modifications to the manuscript are shown in* red. *With these changes, we hope to sufficiently addresses the minor and technical comment from the reviewer.*

**Specific Comments**

**Comment 1** — Line 192: The abbreviation SNR should be introduced.

**Reply**: We have added an abbreviation for the term SNR

"The method filters for the line-of-sight velocity and the Signal-Noise-Ratio (SNR) in a bi-variate manner based upon the assumption of self-similarity of valid data. The method is applied on all the collected $v_{\mathrm{vlos}}$ measurements on the measurement plane and is capable of identifying hard targets such as the

nacelle and blades through the clusters in the $v_{\mathrm{vlos}}$- SNR space. The measurements are discretized and grouped into bins based on their $v_{\mathrm{vlos}}$- SNR values."

**Comment 2** — Line 515-523: The gist of the author's reply to Comment 10 of Reviewer 3 should be included in the in the first paragraph of Section 3.2.3. Currently, the text reads like the difference in wake position is explained by the yaw offset alone. It should mention that both, the differences in wind direction and the differences in the yaw offset, contribute to the wake displacement.

**Reply**:  We have added the following sentence to the revised manuscript.
"For both cases, the partially waked inflow into WT2 is caused due to a combination of the yaw offset applied on WT1 and the misalignment of the wind direction with the orientation of the WT1-WT2 axis."